# Intestinal IL-22RA1 signaling regulates intrinsic and systemic lipid and glucose metabolism to alleviate obesity-associated disorders

Stephen J. Gaudino[1], Ankita Singh[1], Huakang Huang[1], Jyothi Padiadpu [2], Makheni Jean-Pierre[1], Cody Kempen[1], Tej Bahadur[1], Kiyoshi Shiomitsu [1], Richard Blumberg [3], Kenneth R. Shroyer[4], Semir Beyaz[5], Natalia Shulzhenko [6], Andrey Morgun [2] & Pawan Kumar[1] ✉

IL-22 is critical for ameliorating obesity-induced metabolic disorders. However, it is unknown where IL-22 acts to mediate these outcomes. Here we examine the importance of tissue-specific IL-22RA1 signaling in mediating long-term high fat diet (HFD) driven metabolic disorders. To do so, we generated intestinal epithelium-, liver-, and white adipose tissue (WAT)-specific *Il22ra1* knockout and littermate control mice. Intestinal epithelium- and liver-specific IL-22RA1 signaling upregulated systemic glucose metabolism. Intestinal IL-22RA1 signaling also mediated liver and WAT metabolism in a microbiota-dependent manner. We identified an association between *Oscillibacter* and elevated WAT inflammation, likely induced by *Mmp12* expressing macrophages. Mechanistically, transcription of intestinal lipid metabolism genes is regulated by IL-22 and potentially IL-22-induced IL-18. Lastly, we show that Paneth cell-specific IL-22RA1 signaling, in part, mediates systemic glucose metabolism after HFD. Overall, these results elucidate a key role of intestinal epithelium-specific IL-22RA1 signaling in regulating intestinal metabolism and alleviating systemic obesity-associated disorders.

Interleukin (IL)−22 is an IL-10 family cytokine that is predominantly produced by Th17 cells and type 3 innate lymphoid cells. IL-22 signals through its receptor complex (IL-22RA1/IL-10Rβ) which is expressed throughout the body including by the intestines, liver, and adipose tissues[1,2]. Previous studies have shown that IL-22 is protective against inflammatory and metabolic disorders[3–5]. IL-22 particularly plays a critical role in maintaining homeostatic properties in the intestine by signaling to an assortment of unique cell types. The intestinal epithelium consists of crypt/villus structures that are comprised of absorptive and secretory cell lineages. Leucine-rich repeat-containing G protein coupled-receptor 5 (Lgr5)[+] intestinal stem cells (ISCs) reside at the bottom of the crypt where they replicate. As ISCs move toward the villus, they develop into progenitor cells which can differentiate into either absorptive enterocytes or secretory (Paneth, goblet, endocrine, tuft) cell types. Paneth cells particularly play a key role in secreting antimicrobial proteins (including α-defensins and lysozyme) that maintain a homeostatic microbiota as well as growth factors that maintain the Lgr5[+] ISC niche[6,7]. IL-22 acts on Lgr5[+] ISCs to regulate

[1]Department of Microbiology and Immunology, Renaissance School of Medicine, Stony Brook University, Stony Brook, NY, USA. [2]College of Pharmacy, Oregon State University, Corvallis, OR, USA. [3]Department of Medicine, Brigham and Women's Hospital, Harvard Medical School, Boston, MA 02115, USA. [4]Department of Pathology, Renaissance School of Medicine, Stony Brook University, Stony Brook, NY, USA. [5]Cold Spring Harbor Laboratory, Cold Spring Harbor, NY 11724, USA. [6]Carlson College of Veterinary Medicine, Oregon State University, Corvallis, OR, USA. ✉e-mail: pawan.kumar@stonybrook.edu

epithelial regeneration, goblet cells to mediate mucous secretion, and Paneth cells and other enterocytes to regulate antimicrobial peptide production[1,3,8–11].

The rise of obesity and its associated metabolic disorders has become an issue of great concern in recent years and can result in debilitating health disorders[12]. IL-22 plays a beneficial role in ameliorating various metabolic disorders[4,13]. After a prolonged high-fat diet (HFD), Il22RA1[−/−] mice displayed increased weight gain and impaired clearance of glucose from their blood[4]. Furthermore, wildtype mice fed a HFD and intraperitoneally injected with IL-22-Fc, a fusion protein of human IL-22 and mIgG2a, displayed improved glucose clearance and decreased weight gain, adiposity, and hepatic lipid levels compared to their IgG-injected counterparts[4]. However, studies utilizing Il22[−/−] mice or models that regulate endogenous IL-22 levels did not observe differences in glucose clearance when compared to their controls[4,14]. This suggests that IL-22 may act in a context-dependent manner and highlights the importance of tissue-specific IL-22RA1 signaling in regulating metabolism. In addition, IL-22 may differentially mediate systemic inflammatory responses. IL-22 reduces liver damage during murine injury models but increases inflammation in adipose tissue from patients with obesity and type II diabetes[15–17].

While studies have indicated a beneficial role for IL-22 in ameliorating the onset of metabolic disorders, key questions remain. Primarily, it remains unknown which tissue types (liver, intestines, or adipose) IL-22 specifically acts on to mediate these outcomes. It is important to note that previous studies evaluating the role of IL-22 in mediating metabolic disorders have relied solely on the use of global knockouts of the cytokine or its receptor or systemic treatment and/or depletion of IL-22. While these models help determine the global effects that IL-22 has on mediating metabolism, it is difficult to dissect the tissue-specific functions regulated by IL-22. Elucidating the tissue-specific roles of IL-22 is vital to understand how this cytokine mediates key metabolic and inflammatory functions that influence disease etiology as well as for the development of more targeted therapeutics.

To address this question, we generated tissue-specific Il22ra1 knockout and littermate control mice of the intestinal epithelium (Il22ra1[fl/fl];Villin-cre), liver (Il22ra1[fl/fl];Albumin-cre), and white adipose tissue (WAT; Il22ra1[fl/fl];Adiponectin-cre). Overall, we elucidate an important regulatory role of intestinal IL-22RA1 signaling in mediating systemic glucose and lipid metabolism after diet-induced obesity.

## Results

### Systemic IL-22 differentially regulates lipid metabolism in the WAT, liver, and small intestine

We first examined what general metabolic effects may be regulated by IL-22. Importantly, we recapitulated data showing that Il22[−/−] mice display increased weight gain and similar glucose clearance when compared to wildtype (C57BL/6J) control mice after long-term HFD (Fig. 1A, B).

Next, we wanted to observe what effects IL-22 solely exerts under homeostatic conditions across different metabolic tissues. To do so, C57BL/6 wildtype mice (Jackson Laboratory) were fed a chow diet to recapitulate homeostatic conditions and minimize the effects that more inflammatory diets may have on gene expression. We then intraperitoneally injected mice with either 0 or 80 µg of IL-22.Fc and harvested tissues after 24 h. This dose was used based on our previous publication and is high enough to broadly induce the expression of IL-22-regulated genes to assess its general effects on systemic tissues under homeostatic conditions[8]. IL-22 has been shown to influence functions of the WAT, liver, and intestines. We confirmed that these tissues all express Il22ra1 and that IL-22 treatment did not regulate the expression of its receptor (Fig. 1C). After IL-22 treatment, we did not observe altered expression of lipid metabolism- or inflammation-associated genes in the WAT (Supplementary Fig. 1A). We did, however, observe significantly elevated expression of Pnpla2 and Lipe in

the liver of mice treated with IL-22.Fc (Fig. 1D). In addition, we observed increased staining by Oil Red O (ORO), a dye that specifically stains lipid droplets, of HepG2 cells that were treated with 4% lipid mixture (LM) and 80 ng IL-22.Fc, as opposed to pure LM-treated groups (Fig. 1E). Altogether, this suggests that IL-22 increases hepatic lipid metabolism. Upon analyzing the effects of IL-22 in the WAT and liver, we evaluated how IL-22.Fc treatment regulated intestinal function. As expected, we observed that the expression of IL-22 inducible antimicrobial peptide genes, Reg3g and Reg3b, were upregulated in the ileum (Fig. 1F). Opposite to what we observed in the liver, the expression of Pnpla2 and Lipe were significantly downregulated in the ileum after IL-22.Fc treatment (Fig. 1G). While we did not detect a difference in the expression of peroxisome proliferator-activated receptor genes, Ppara and Pparg (key transcription factors that aid peroxisomal fatty acid metabolism), in the liver after IL-22.Fc treatment, we observed that IL-22.Fc treatment resulted in decreased expression of these genes in the ileum (Fig. 1D, G). This decrease in Ppara coincides with previous studies that display that Pnpla2 activates transcription of Ppara and its target genes[18,19]. To examine the ex vivo effects of IL-22, small intestinal organoids from wildtype (C57BL/6) mice were cultured in the presence or absence of 4% LM and 5 ng rIL-22 and were stained with ORO. We observed decreased ORO staining in the LM and rIL-22-treated organoids as opposed to the pure LM-treated groups (Fig. 1H). This impaired lipid accumulation in rIL-22-treated organoids correlates with the reduced expression of ileal lipid metabolism genes after IL-22.Fc treatment and implies that IL-22 acts to downregulate intestinal lipid metabolism. To assess whether rIL-22 acts directly on intestinal epithelial cells to mediate the expression of these genes, we stimulated C57BL6/J mice small intestinal organoids with rIL-22 (Fig. 1I). Expression of Pnpla2, Lipe, Ppara, and Pparg were all reduced after rIL-22 stimulation which supports this conclusion.

Our data indicate that systemic IL-22 induces opposing metabolic effects depending on its tissue-specific target. However, the tissue-specific effects of IL-22RA1 signaling remain unclear.

### Deficiency of intestinal epithelium-specific IL-22RA1 signaling mediates the onset of systemic metabolic disorders

We next examined how IL-22RA1 signaling to specific tissues regulates metabolism. Since the intestines regulate the absorption of nutrients, communicate with other metabolic tissues, and highly express Il22ra1 (Fig. 1C), we assessed the intestinal epithelium-specific role of IL-22RA1 signaling in mediating systemic metabolism. To do so, we used Il22ra1[fl/fl]; Villin-cre mice. Knockdown of Il22ra1 in Il22ra1[fl/fl];Villin-cre+ mice was confirmed by reverse transcriptase-polymerase chain reaction (RT-PCR) (Fig. 2A). No difference was observed in weight gain or systemic glucose clearance between control diet-fed Il22ra1[fl/fl];Villin-cre+ and cre- mice (Fig. 2B; Supplementary Fig. 2A).

To assess the effects of intestinal IL-22RA1 signaling in mediating the onset of diet-induced obesity metabolic disorders, we placed Il22ra1[fl/fl];Villin-cre mice on a long-term (16 weeks) HFD. The HFD used contains approximately 29% greater fat than the control diet, and HFD-fed mice receive 60% of their dietary calories from fat (Supplementary Table 1). After HFD, Il22ra1[fl/fl];Villin-cre mice displayed greater weight gain than control diet-fed mice, but no difference in weight was observed between HFD-fed cre- and cre+ mice (Fig. 2B). However, upon conducting glucose tolerance tests (GTTs), Il22ra1[fl/fl];Villin-cre+ mice displayed impaired glucose clearance when compared to their littermate controls (Fig. 2C), indicating that intestinal-specific IL-22RA1 signaling is important for increasing the clearance of systemic glucose. No differences in insulin resistance or serum c-peptide and insulin levels were detected between HFD-fed Il22ra1[fl/fl];Villin-cre mice to account for their differences in glucose clearance (Supplementary Fig. 2B, C).

To assess whether intestinal IL-22RA1 signaling mediates extra-intestinal metabolism, we first examined the liver of Il22ra1[fl/fl];Villin-cre

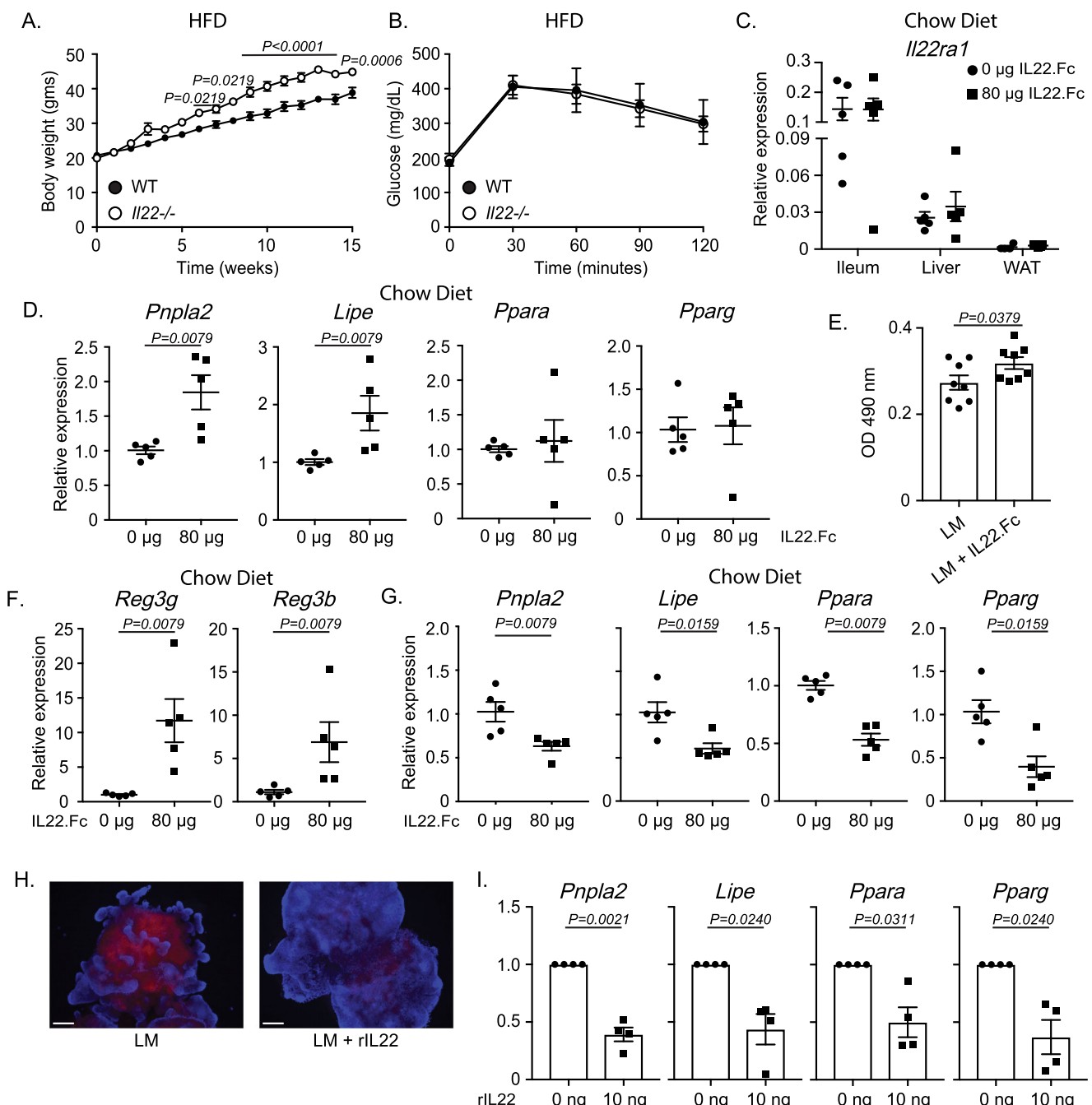

**Fig. 1 | IL-22 differentially regulates systemic lipid metabolism. A** Weight gain from a high-fat diet (HFD) fed wildtype (WT) and *Il22⁻/⁻* mice. **B** Glucose tolerance test (GTT) from HFD-fed *Il22⁻/⁻* mice. **C** RT-PCR analysis of *Il22ra1* expression from wildtype mice injected with either 0 or 80 µg IL-22.Fc. **D** Expression of lipid metabolism genes from liver tissues of wildtype mice fed a chow diet and treated with or without IL-22.Fc. **E** Quantification of ORO staining from HepG2 cells treated with either 4% lipid mixture (LM) or 4% LM and 80 ng IL-22.Fc. Data depict OD 490 nm values. **F** Expression of antimicrobial genes from ileal tissues of wildtype mice fed a chow diet and treated with or without IL-22.Fc. **G** Expression of lipid metabolism genes from ileal tissues of wildtype mice treated with or without IL-

22.Fc. **H** Representative image of small intestinal organoids treated with 4% LM or 4% LM and 5 ng rIL-22. **I** RT-PCR analysis of lipid metabolism genes from small intestinal organoids treated with 0 or 10 ng rIL-22. Figure 1A–I was generated from 2–3 independent experiments. *N* = 6 WT and 5 knockout mice for Fig. 1A. *N* = 7 WT and 5 knockout mice for Fig. 1B. Figure 1H is representative of 3 mice. *N* = 5 mice in each group for Fig. 1C. *N* = 5 mice in each group for Figs. 1D, F, and G. *N* = 8 replicates in Fig. 1E. *N* = 4 mice in Fig. 1I. Data are presented as mean ± SEM in all graphs. Mann–Whitney test, two-tailed in 1**D**–**G**, 1**I**; 2-way ANOVA with Sidak multiple comparisons in 1**A**–**C**. Scale bar = 100 µm.

---

mice. *Il22ra1*ᶠˡ/ᶠˡ;*Villin-cre* mice fed a control diet displayed similar hepatic histology (Supplementary Fig. 2D). However, HFD-fed *Il22ra1*ᶠˡ/ᶠˡ; *Villin-cre+* mice displayed increased hepatocyte ballooning and lipid droplet deposition (Fig. 2D, E). Lipid deposition was quantified via hepatic ORO staining (Fig. 2E [right]). RT-PCR analysis of key glucose (*Foxo1*, *G6PC*) and lipid (*Acc*, *Ppara*) metabolism genes revealed

decreased transcripts of *Acc* and *Ppara* in *Il22ra1*ᶠˡ/ᶠˡ;*Villin-cre+* mice after HFD but not control diet (Fig. 2F). No changes were observed in the expression of additional metabolic genes after HFD feeding (Supplementary Fig. 2E). Correlating with these findings, gas chromatography analysis of liver tissues from HFD-fed *Il22ra1*ᶠˡ/ᶠˡ;*Villin-cre+* mice also displayed elevated levels of total ceramides (significant) and cholesterol

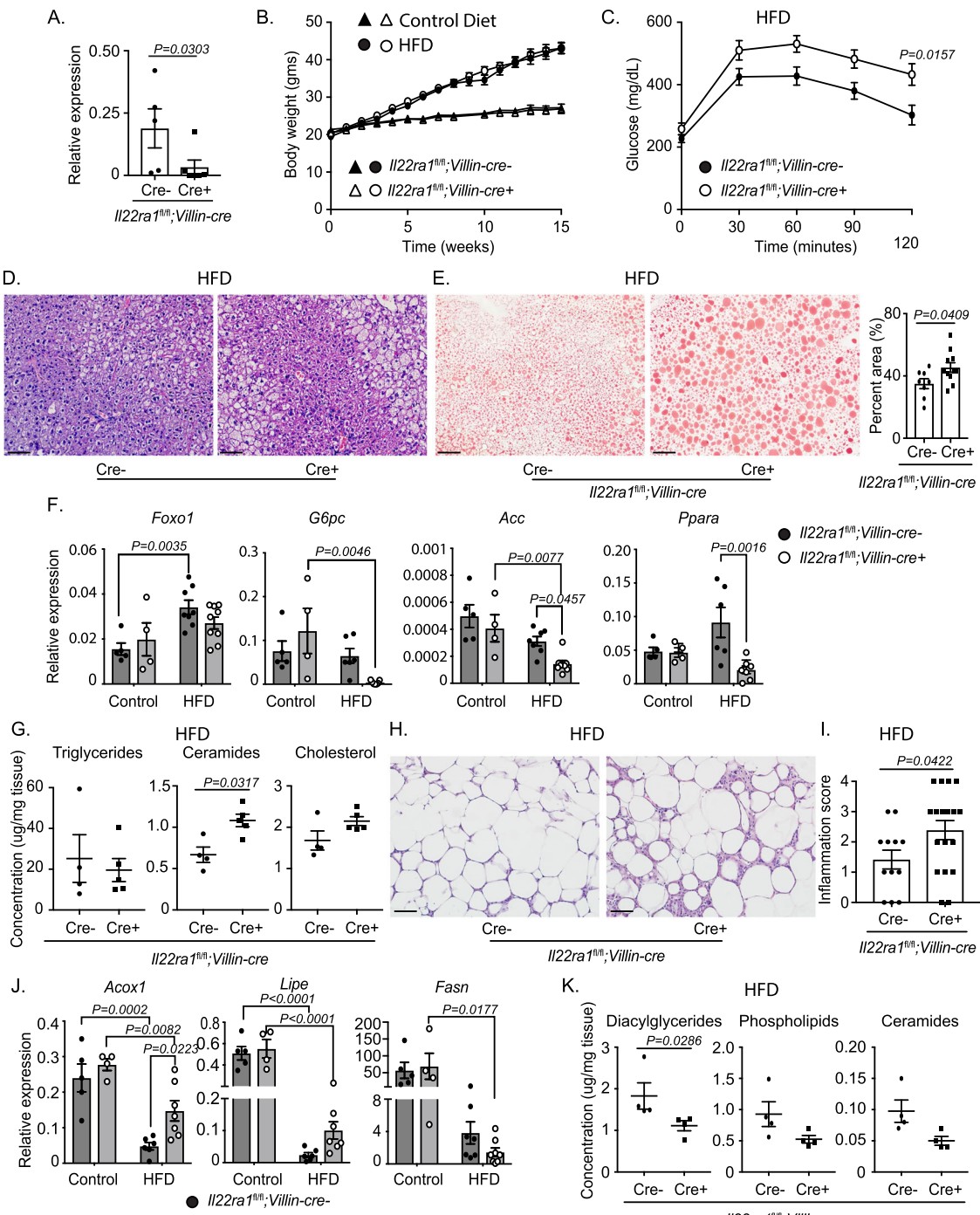

**Fig. 2 | Systemic metabolism (but not weight gain) is regulated by intestinal-epithelium- and liver-specific IL-22RA1 signaling. A** RT-PCR analysis of ileal *Il22ra1* expression from *Il22ra1*[fl/fl]*;Villin-cre* mice. **B** Weight gain from *Il22ra1*[fl/fl]*;Villin-cre* mice. **C** GTT from HFD-fed *Il22ra1*[fl/fl]*;Villin-cre* mice. **D** Representative H&E images of liver tissue from HFD-fed *Il22ra1*[fl/fl]*;Villin-cre* mice. **E** Representative Oil Red O (ORO) images with quantification from HFD-fed *Il22ra1*[fl/fl]*;Villin-cre* mice. **F** RT-PCR analysis of hepatic glucose and lipid metabolism genes from HFD-fed *Il22ra1*[fl/fl]*;Villin-cre* mice. **G** Gas chromatography of total hepatic triglycerides, ceramides, and cholesterol from HFD-fed *Il22ra1*[fl/fl]*;Villin-cre* mice. **H** Representative H&E images of epididymal WAT from HFD-fed *Il22ra1*[fl/fl]*;Villin-cre* mice. **I** Scoring of tissues from Fig. 2H. **J** RT-PCR analysis of lipid metabolism genes from the epididymal WAT of *Il22ra1*[fl/fl]*;Villin-cre* mice. **K** Gas chromatography of diacylglycerides, phospholipids, and ceramides from epididymal WAT of HFD-fed *Il22ra1*[fl/fl]*;Villin-cre* mice. Figure 2A represents 5 cre- and 6 cre+ mice. Figure 2B triangles represent

control diet and circles represent HFD. Figure 2B represents at least 2 independent experiments for HFD ($N = 6$ cre- and 4 cre+) and control diet ($N = 9$ cre- and 8 cre+) mice. Figure 2C represents $N = 4$ cre- and 6 cre+ mice from 2 independent experiments. Figure 2D represents $N = 4$ cre- and 6 cre+ mice from 2 independent experiments. Figure 2E represents $N = 8$ cre- and 11 cre+ mice from 5 independent experiments. Figures 2F and J represent $N = 5$ (control, cre-), 4 (control, cre+), 6–8 (HFD, cre-), and 7–9 (HFD, cre+) mice per group and 2 (control diet) or 3 (HFD) independent experiments. Figure 2G ($N = 4$ cre- and 5 cre+ mice) and 2**K** ($N = 4$ cre- and 4 cre+ mice) are representative of 2 independent experiments. Figure 2H–I represent $N = 12$ cre- and 18 cre+ mice and 5 independent experiments. Data are presented as mean ± SEM in all graphs. 2-way ANOVA with Sidak multiple comparisons in 2**B**, **C**, 2**F**, 2**J**; Mann–Whitney test, two-tailed in 2**A**, 2**E** [right], 2**G**, 2**I**, and 2**K**. Scale bar = 50 µm.

(Fig. 2G). However, significant effects were not detected when analyzing individual groups of lipids (Supplementary Fig. 2F).

Next, we assessed if the absence of intestinal IL-22RA1 signaling affects WAT functions. While control diet-fed *Il22ra1*<sup>fl/fl</sup>;*Villin-cre* mice displayed similar WAT histology (Supplementary Fig. 3A), WAT from HFD-fed *Il22ra1*<sup>fl/fl</sup>;*Villin-cre+* mice displayed significantly greater inflammation (i.e., accumulation of crown-like structures) than their littermate controls (Fig. 2H, I). Macrophage accumulation was confirmed via immunofluorescence microscopy (Supplementary Fig. 3B, C). IL-22 also has been associated with elevated adipose inflammation in patients with obesity and after HFD[15,20,21]. No difference in epididymal WAT fat pad mass was observed between HFD-fed *Il22ra1*<sup>fl/fl</sup>;*Villin-cre-* and *cre+* mice (Supplementary Fig. 3D). RT-PCR analysis of lipid catabolizing (*Acox1* and *Lipe*) and synthesizing (*Fasn*) genes revealed that *Il22ra1*<sup>fl/fl</sup>;*Villin-cre+* mice displayed greater transcript levels of *Acox1* after HFD but not control diet (Fig. 2J). A trend for elevated *Lipe* and decreased *Fasn* expression was present (Fig. 2J). We further examined changes in WAT lipid metabolism by gas chromatography. WAT from *Il22ra1*<sup>fl/fl</sup>;*Villin-cre+* mice displayed lower levels of diacylglycerides (significant), phospholipids, and ceramides (Fig. 2K). However, significant effects were not detected when analyzing individual groups of lipids (Supplementary Fig. 3E). No difference in WAT triglyceride levels was observed (Supplementary Fig. 3F).

We next assessed whether altered levels of systemic IL-22 between *Il22ra1*<sup>fl/fl</sup>;*Villin-cre-* and *cre+* mice may mediate the onset of these HFD-induced phenotypes. However, no differences in ileal *Il22* expression as well as liver-specific or serum-specific IL-22 levels were detected (Supplementary Fig. 3G–I).

Altogether, our data suggest that intestinal IL-22RA1 signaling plays an important role in regulating extra-intestinal gene expression and lipid metabolism and accumulation.

## Deficiency of liver-specific IL-22RA1 signaling impairs systemic glucose metabolism

To determine whether impaired systemic glucose and lipid metabolism are primary consequences of intestinal IL-22RA1 signaling, we assessed the liver-specific role of IL-22RA1 signaling. We generated *Il22ra1*<sup>fl/fl</sup>;*Albumin-cre* mice and confirmed *Il22ra1* knockdown by RT-PCR (Fig. 3A). No differences in weight gain were observed between *Il22ra1*<sup>fl/fl</sup>;*Albumin-cre-* and *cre+* mice after feeding either a control diet or HFD, although increased weight gain was observed after HFD (Fig. 3B). However, we observed that *Il22ra1*<sup>fl/fl</sup>;*Albumin-cre+* mice displayed significantly impaired glucose clearance compared to their cre- counterparts after HFD but not control diet (Fig. 3C; Supplementary Fig. 4A). Thus, in addition to intestinal signaling, hepatic IL-22RA1 signaling regulates systemic glucose metabolism.

No major differences in hepatic histology or lipid deposition were observed between *Il22ra1*<sup>fl/fl</sup>;*Albumin-cre* mice after HFD (Fig. 3D, E). Except for *G6pc*, no significant changes were observed in glucose or lipid metabolism genes that were downregulated in *Il22ra1*<sup>fl/fl</sup>;*Villin-cre+* mice (Figs. 2F and 3F; Supplementary Fig. 4B). Notably, no difference in *Ppara* expression was observed (Fig. 3F). This coincides with our data from Fig. 1D and additionally suggests that signaling outside the liver (including the intestine) is necessary for mediating hepatic expression of *Ppara*. To further assess changes in lipid metabolism, we performed gas chromatography analysis of liver tissues from HFD-fed *Il22ra1*<sup>fl/fl</sup>;*Albumin-cre* mice. Unlike *Il22ra1*<sup>fl/fl</sup>;*Villin-cre* mice, *Il22ra1*<sup>fl/fl</sup>;*Albumin-cre+* mice displayed significantly decreased levels of triglycerides but similar levels of ceramides and cholesterol (Figs. 2H and 3G; Supplementary Fig. 4C). Upon examining WAT histology, epididymal adiposity, and gene expression (*Lipe, Acox1, Fasn*), no differences were observed between *Il22ra1*<sup>fl/fl</sup>;*Albumin-cre* mice (Fig. 3H–J). Altogether, this data further highlights the tissue-specific role that IL-22RA1 signaling induces in regulating metabolism.

## Intestinal IL-22RA1 signaling mediates extra-intestinal metabolism in a microbiota-dependent manner

We next assessed what intestinal factors may be regulated by IL-22 to mediate systemic metabolism and inflammation. Notably, the intestinal microbiota mediates key host functions including the absorption of nutrients and lipids[22,23]. However, during obesity and after HFD, the microbiota undergoes a dysbiosis[24,25]. Microbial dysbiosis has been associated with altered lipid absorption, metabolism, and adiposity[22,23,26,27]. We and others have shown that IL-22 induces various antimicrobial proteins including Reg3 family proteins and Lyz1[8,11,28]. Likewise, an interconnected nature exists between the intestinal microbiota and IL-22[29,30]. Therefore, we evaluated if IL-22-mediated regulation of the microbiota affects systemic metabolism after HFD.

As expected, 16S rRNA sequencing revealed differences in alpha and beta diversity when comparing the genus-level microbial composition of cre- and cre+ mice before (day 0) and after (day 105) HFD; however, no differences were observed between the two genotypes in the same period (Supplementary Fig. 5A, B). We then assessed differences among specific bacterial genera. Thirteen genera were significantly differentially regulated in a two-factor (diet, genotype) analysis with the predominant effect of IL-22 deficiency (Fig. 4A). Notably, among these microbes, we detected the genus *Oscillibacter* (Fig. 4A, B), which has been previously reported to worsen glucose metabolism[31]. Furthermore, while *Il22ra1*<sup>fl/fl</sup>;*Villin-cre+* mice displayed only a trend of increased levels of *Oscillibacter* before HFD (Supplementary Fig. 5C, D), cre+ mice displayed a significantly greater abundance after HFD (Fig. 4B; Supplementary Fig. 5D). During obesity, *Oscillibacter* stimulates *Mmp12* expression by WAT-associated macrophages which results in impaired glucose metabolism[31]. Reduction of this microbe in the gut is associated with decreased signatures of insulin-resistance-associated macrophages in the WAT and improved systemic glucose metabolism[31]. To verify if this mechanism may be operating in our system, we evaluated *Mmp12* expression in epididymal WAT and found that *Il22ra1*<sup>fl/fl</sup>;*Villin-cre+* mice displayed elevated transcripts of *Mmp12* as well as MMP12 immunofluorescence staining after HFD (Fig. 4C, D). These results indicate a potential contribution of *Oscillibacter* for driving WAT inflammation and, in turn, dysregulation of glucose metabolism in *Il22ra1*<sup>fl/fl</sup>;*Villin-cre+* mice.

To further assess whether differences in microbiota composition mediate extra-intestinal metabolic and inflammatory responses after HFD, we treated *Il22ra1*<sup>fl/fl</sup>;*Villin-cre* mice with broad-spectrum antibiotics during their last 4 weeks of HFD to deplete their microbiota. After HFD and antibiotics treatment, we did not observe any differences in weight gain between cre- and cre+ mice (Fig. 4E). In addition, *Il22ra1*<sup>fl/fl</sup>;*Villin-cre-* and *cre+* mice displayed similar clearance of glucose from their blood (Fig. 4F). This is notable since non-antibiotics-treated *Il22ra1*<sup>fl/fl</sup>;*Villin-cre+* mice displayed impaired clearance of blood glucose when compared to cre- mice (Fig. 2C).

To additionally verify if microbiota-induced changes influence systemic glucose metabolism, we gavaged cecal content homogenate collected from *Il22ra1*<sup>fl/fl</sup>;*Villin-cre* mice after 16 weeks of HFD into control diet-fed germ-free mice. We observed that mice colonized with *Il22ra1*<sup>fl/fl</sup>;*Villin-cre+* cecal contents displayed significantly impaired glucose clearance 30 min after glucose injection (Supplementary Fig. 5E). We also cohoused *Il22ra1*<sup>fl/fl</sup>;*Villin-cre-* and *cre+* mice over a 16 week HFD period. Unlike our antibiotics and microbiota transplantation data, cohoused *Il22ra1*<sup>fl/fl</sup>;*Villin-cre-* and *cre+* mice displayed similar trends in glucose metabolism as non-cohoused mice (Figs. 2C and 4F; Supplementary Fig. 5F). This indicates that the microbiota is indeed one of the key factors that regulate systemic metabolism, but other factors, such as tissue-specific IL-22RA1 signaling or HFD consumption, play additional roles.

To further assess the importance of IL-22 mediation of microbiota composition in regulating the onset of HFD-induced disorders, we

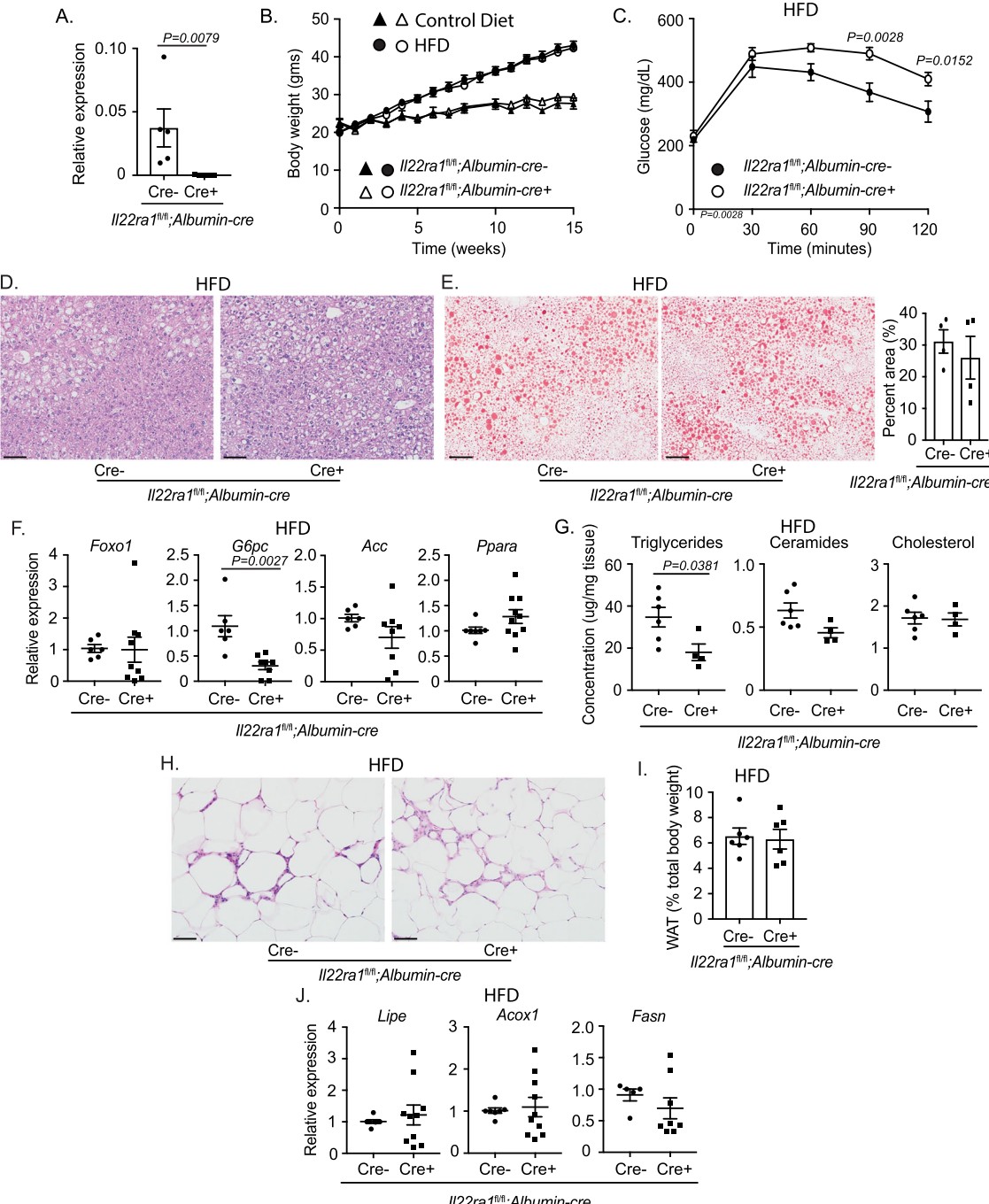

**Fig. 3 | Liver-specific IL-22RA1 signaling regulates systemic glucose metabolism. A** RT-PCR analysis of *Il22ra1* expression from liver tissues of *Il22ra1*fl/fl;*Albumin-cre* mice. **B** Weight gain from HFD-fed *Il22ra1*fl/fl;*Albumin-cre* mice. **C** GTT from HFD-fed *Il22ra1*fl/fl;*Albumin-cre* mice. **D** Representative H&E images of liver tissue from HFD-fed *Il22ra1*fl/fl;*Albumin-cre* mice. **E** Representative ORO images (left) and quantification (right) of liver tissue from HFD-fed *Il22ra1*fl/fl;*Albumin-cre* mice. **F** RT-PCR analysis of glucose (*Foxo1*, *G6pc*) and lipid (*Acc*, *Ppara*) metabolism genes from the liver of HFD-fed *Il22ra1*fl/fl;*Albumin-cre* mice. **G** Concentration of total triglycerides, ceramides, and cholesterol from liver tissues of HFD-fed *Il22ra1*fl/fl;*Albumin-cre* mice determined by gas chromatography. **H** Representative H&E images of epididymal WAT from HFD-fed *Il22ra1*fl/fl;*Albumin-cre* mice. **I** Epidydimal WAT fat pad mass after HFD. **J** RT-PCR analysis of lipid metabolism genes from the epididymal WAT of HFD-fed *Il22ra1*fl/fl;*Albumin-cre* mice. Figure 3A is generated from 5

cre- and 5 cre+ mice. Figure 3B triangles represent control diet and circles represent HFD. Figure 3B is representative of at least 3 independent experiments for HFD (*N* = 7 cre- and 7 cre+) and control diet (*N* = 5 cre- and 6 cre+) mice. Figure 3C is generated from *N* = 7 cre- and 7 cre+ mice and 3 independent experiments. Figure 3D is representative of 6 cre- and 10 cre+ mice. Figure 3E is representative of *N* = 4 cre- and 4 cre+ mice from 2 independent experiments. Figures 3F and 3J are generated from *N* = 5–7 cre- and 8–10 cre+ mice per group. Figure 3H is representative of *N* = 6 cre- and 8 cre+ mice. Figures 3C, D, F, H, J are representative of 4 independent experiments. Figure 3G, I are generated from 3 independent experiments. Data are presented as mean ± SEM in all graphs. 2-way ANOVA with Sidak multiple comparisons in 3**B**, **C**, 3**F**, 3**J**; Mann–Whitney test, two-tailed in 3**A**, 3**E** (right)–**G**, 3**I**, 3**K**. Scale bar = 50 μm.

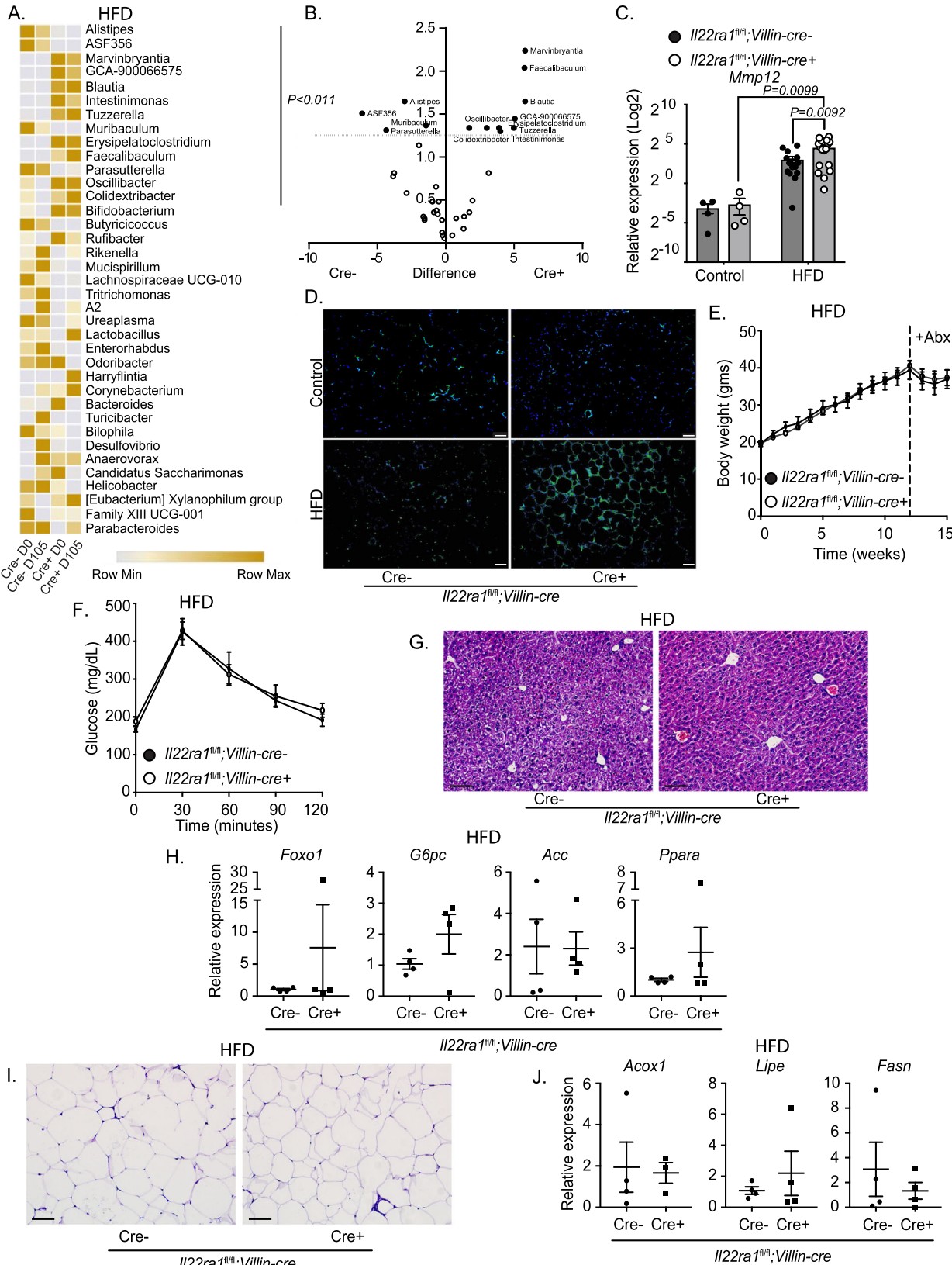

analyzed the liver and WAT of antibiotics-treated mice. Respectively, H&E and ORO staining of liver tissues revealed that *Il22ra1*<sup>fl/fl</sup>*;Villin-cre-* and cre+ mice displayed similar histology and lipid accumulation after antibiotics treatment (Fig. 4G; Supplementary Fig. 5G). While RT-PCR analysis revealed impaired expression of *Foxo1, G6PC, Acc*, and *Ppara* from liver tissues of non-antibiotics-treated *Il22ra1*<sup>fl/fl</sup>*;Villin-cre+* mice,

no notable differences in the expression of glucose and lipid metabolism genes were observed among mice after antibiotics treatment (Fig. 2F and 4H). Additionally, H&E analysis of WAT post antibiotics revealed that *Il22ra1*<sup>fl/fl</sup>*;Villin-cre-* and cre+ mice displayed similar histology (Fig. 4I). While greater trends for *Acox1* and *Lipe* transcripts were detected via RT-PCR in non-antibiotics-treated *Il22ra1*<sup>fl/fl</sup>*;*

**Fig. 4 | Intestinal IL-22RA1-mediated alterations in microbiota composition regulate systemic metabolism and inflammation. A** Genus-level heatmap based on Euclidian distance with average linkage using Hierarchical clustering of 16S rRNA sequencing data derived from fecal DNA of *Il22ra1*<sup>fl/fl</sup>;*Villin-cre* mice before (day 0) and after (day 105) HFD. **B** Volcano plot displaying a differential expression of bacterial genera from *Il22ra1*<sup>fl/fl</sup>;*Villin-cre* mice after HFD. **C** RT-PCR analysis of epididymal WAT *Mmp12* expression from *Il22ra1*<sup>fl/fl</sup>;*Villin-cre* mice fed a control diet or HFD for 16 weeks. **D** Representative immunofluorescence image of MMP12 staining of epididymal WAT from HFD-fed *Il22ra1*<sup>fl/fl</sup>;*Villin-cre* mice. **E** Weight gain of HFD-fed *Il22ra1*<sup>fl/fl</sup>;*Villin-cre* mice after 4 weeks of antibiotics treatment starting at week 12 (indicated by dotted line). **F** GTT from HFD-fed *Il22ra1*<sup>fl/fl</sup>;*Villin-cre* mice after antibiotics treatment. **G** Representative H&E images of liver tissue from antibiotics treated and HFD-fed *Il22ra1*<sup>fl/fl</sup>;*Villin-cre* mice. **H** RT-PCR analysis of glucose (*Foxo1*, *G6pc*) and lipid (*Acc*, *Ppara*) metabolism genes from the liver of

antibiotics treated and HFD-fed *Il22ra1*<sup>fl/fl</sup>;*Villin-cre* mice. **I** Representative H&E images of epididymal WAT from antibiotics treated and HFD-fed *Il22ra1*<sup>fl/fl</sup>;*Villin-cre* mice. **J** RT-PCR analysis of lipid metabolism genes from the epididymal WAT of antibiotics-treated and HFD-fed *Il22ra1*<sup>fl/fl</sup>;*Villin-cre* mice. Figures 4A–B, 4E–J are representative of 2 independent experiments. Figure 4A, B are representative of $N = 6$ cre- and 5 cre+ mice. Figure 4C is representative of $N = 4$ (control, cre-), 4 (control, cre+), 14 (HFD, cre-), and 17 (HFD, cre+) mice from 2 (control) or 9 (HFD) independent experiments. Figure 4D is representative of $N = 4$ cre- and 3 cre+ control diet-fed mice, $N = 4$ cre- and 5 cre+ HFD-fed mice, and 2 (control diet) and 3 (HFD) independent experiments. Figure 4E–J are representative of $N = 4$ cre- and 4 cre+ mice. Data are presented as mean ± SEM in all graphs. 2-way ANOVA with Sidak multiple comparisons in 4**C**, 4**E**, **F**; Unpaired *t*-test, one-tailed in **B**; Mann–Whitney test, two-tailed in **H**, **J**. Scale bar = 50 μm.

*Villin-cre+* mice, this difference was ablated after antibiotics treatment (Figs. 2J and 4J).

Our data indicate that intestinal IL-22RA1 signaling is important for maintaining a microbiota that aids systemic metabolism. Furthermore, we identify an association between impaired intestinal IL-22RA1 signaling in elevating *Oscillibacter* levels and increasing levels of WAT-associated *Mmp12*-expressing macrophages.

## WAT-specific IL-22RA1 signaling does not protect against HFD-driven metabolic disorders

WAT is important for regulating lipid and glucose metabolism[32,33]. Furthermore, IL-22 decreases fat pad mass and regulates the expression of lipid metabolism genes[4]. Our observations that epididymal WAT of *Il22ra1*<sup>fl/fl</sup>;*Villin-cre+* display elevated macrophage accumulation and *Mmp12* expression also suggest the importance of IL-22 in mediating WAT-specific functions. Therefore, we investigated the WAT-specific role of IL-22RA1 signaling in mediating systemic metabolism by generating and validating *Il22ra1*<sup>fl/fl</sup>;*Adiponectin-cre+* and cre- mice (Fig. 5A).

We did not observe any differences in weight gain or systemic glucose clearance between *Il22ra1*<sup>fl/fl</sup>;*Adiponectin-cre-* and cre+ mice after placement on long-term HFD (Fig. 5B, C). Likewise, no notable differences in hepatic morphology, lipid deposition, and metabolic transcript levels were observed between *Il22ra1*<sup>fl/fl</sup>;*Adiponectin-cre-* and cre+ mice (Fig. 5D–F). These mice also displayed no differences in WAT morphology, epididymal fat pad weight, and metabolic transcript levels (Fig. 5G–I; Supplementary Fig. 6A). Similar transcript levels of *Il22* were also detected (Supplementary Fig. 6B). These results indicate that WAT-specific IL-22RA1 signaling may not play a critical role in directly regulating HFD-associated metabolic disorders.

## IL-22RA1 signaling directly regulates intestinal lipid metabolism

While we showed that intestinal IL-22RA1 signaling regulates extra-intestinal metabolism after HFD (Fig. 2), we next assessed how IL-22RA1 signaling affects intestinal metabolic pathways. To do so, we performed RNA sequencing on ileal tissues of HFD-fed *Il22ra1*<sup>fl/fl</sup>;*Villin-cre* mice (Fig. 6A). Many genes associated with lipid storage and metabolism were upregulated in the absence of IL-22RA1 signaling (Fig. 6A [left]). Furthermore, many peroxisome-associated genes, some of which are also important for lipid metabolism, were upregulated in *Il22ra1*<sup>fl/fl</sup>;*Villin-cre+* mice (Fig. 6A [right]). This is notable since peroxisomes are key organelles for coordinating lipid metabolism. Specifically, we noticed elevated expression of *Ppara* and *Pparg* (Fig. 6A [right]). To determine whether these phenotypes were diet-induced, we performed RT-PCR analysis of *Ppara* and *Lipe* on ileal tissues from *Il22ra1*<sup>fl/fl</sup>;*Villin-cre-/+* mice fed either an HFD or chow diet. We observed significant upregulation of *Ppara* and *Lipe* transcripts in *Il22ra1*<sup>fl/fl</sup>;*Villin-cre+* mice after HFD but chow diet (Fig. 6B). This suggests that IL-22RA1 signaling is important for regulating intestinal lipid metabolism pathways after HFD.

The importance of intestinal IL-22RA1 in mediating lipid metabolism may explain why *Il22ra1*<sup>fl/fl</sup>;*Albumin-cre+* mice but not *Il22ra1*<sup>fl/fl</sup>;*Villin-cre* mice displayed significantly decreased levels of triglycerides. It is possible that IL-22-mediated inhibition of intestinal lipid metabolism enhances liver lipid accumulation. We show that systemic IL-22 regulates the expression of *Pnpla2*, a triglyceride hydrolyzer, in an opposing manner in the small intestine (reduced) and liver (increased; Fig. 1D, G). It has been shown that intestinal Pnpla2 (ATGL) deficiency has no impact on plasma and liver triglyceride accumulation, but the small intestinal tissues of these mice possess an increased accumulation of triglycerides[18]. Furthermore, crosstalk between the liver and intestines can occur via circulation, thereby opening the possibility that liver-specific IL-22RA1 signaling may release soluble mediators that influence intestinal metabolism. To test this, we placed *Il22ra1*<sup>fl/fl</sup>;*Villin-cre* and *Il22ra1*<sup>fl/fl</sup>;*Albumin-cre* mice on 6 weeks of HFD and injected them with 80 μg IL-22.Fc. In addition, HFD-fed control mice were intraperitoneally injected with either 0 or 80 μg IL-22.Fc. Control mice injected with IL-22.Fc lost significantly greater weight than those treated with pure PBS (Supplementary Fig. 7A). In addition, we observed that *Il22ra1*<sup>fl/fl</sup>;*Villin-cre+* mice were injected with IL-22.Fc lost significantly less weight when compared to their cre- counterparts (Supplementary Fig. 7B); however no significant difference in weight loss was observed between treated *Il22ra1*<sup>fl/fl</sup>;*Albumin-cre* mice (Supplementary Fig. 7C). This indicates that intestinal IL-22RA1 signaling may partially mediate weight loss under conditions with elevated systemic IL-22. However, it is also possible that weight gain relies on IL-22RA1 signaling to tissues besides the intestines, liver, and WAT and/or coordinated IL-22RA1 signaling to different tissue types. Next, we examined the tissue-specific effects of IL-22RA1 signaling on the expression of intestinal lipid metabolism genes. Ileal expression of *Ppara*, *Lipe*, and *Pnpla2* were downregulated by intestinal-specific but not liver-specific IL-22RA1 signaling (Supplementary Fig. 7D), highlighting the importance of tissue-specific IL-22RA1 signaling in mediating intestinal metabolism.

IL-22 exerts its effects by activating downstream transcription factors including Stat3[34]. We observed Stat3 binding sites upstream of the start site for *Ppara* (Supplementary Fig. 7E), indicating that its expression may be regulated in a Stat3-dependent manner. We observed a modest reduction in *Ppara* expression after treatment of organoids with S3I-201 (Stat3 inhibitor) and a significant reduction after treatment with rIL-22 (Supplementary Fig. 7F). However, treatment of S3I-201 did not reverse or enhance the inhibitory effects of rIL-22 on the expression of *Ppara*, indicating that IL-22 may not be responsible for mediating *Ppara* transcription in a Stat3-dependent manner (Supplementary Fig. 7F).

While we show that IL-22 mediates the expression of intestinal lipid metabolism genes (Fig. 6A, B), it is unclear whether IL-22 is solely responsible for promoting these effects. IL-22 has also been shown to induce the production of epithelial IL-18 to provide protection against parasitic and bacterial infections[34,35]. Therefore, we evaluated the role

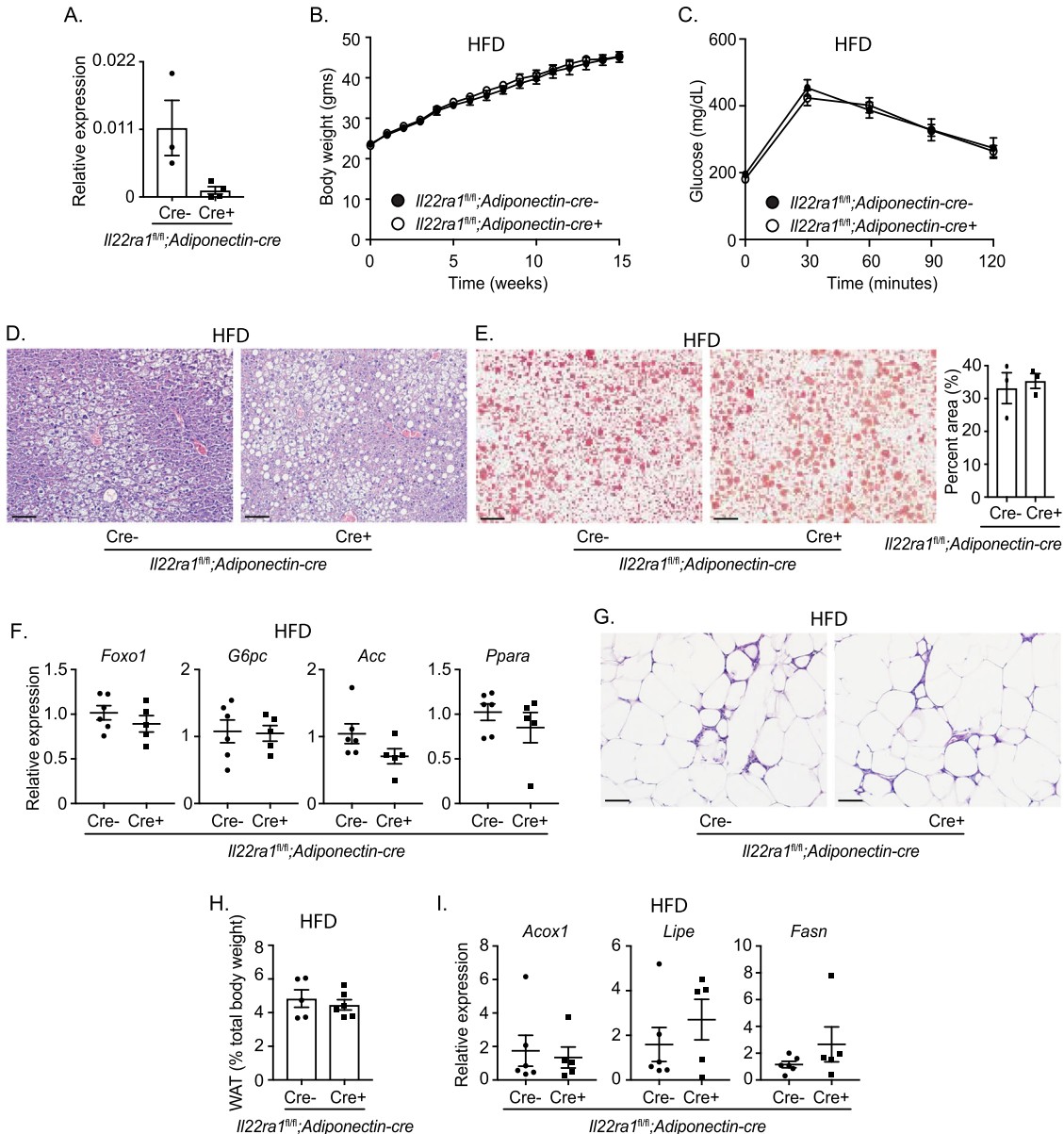

**Fig. 5 | WAT-specific IL-22RA1 signaling does not ameliorate HFD-induced metabolic disorders. A** RT-PCR analysis of *Il22ra1* expression from epididymal WAT of *Il22ra1*^fl/fl^;*Adiponectin-cre* mice. **B** Weight gain from HFD-fed *Il22ra1*^fl/fl^;*Adiponectin-cre* mice. **C** GTT from HFD-fed *Il22ra1*^fl/fl^;*Adiponectin-cre* mice. **D** Representative H&E images of liver tissue from HFD-fed *Il22ra1*^fl/fl^;*Adiponectin-cre* mice. **E** Representative ORO images (left) and quantification (right) of liver tissue from HFD-fed *Il22ra1*^fl/fl^;*Adiponectin-cre* mice. **F** RT-PCR analysis of glucose and lipid metabolism genes from the liver of HFD-fed *Il22ra1*^fl/fl^;*Adiponectin-cre* mice. **G** Representative H&E images of epididymal WAT from HFD-fed *Il22ra1*^fl/fl^; *Adiponectin-cre* mice. **H** Epidydimal WAT fat pad mass after long-term HFD.

**I** RT-PCR analysis of lipid metabolism genes from the epididymal WAT of HFD-fed *Il22ra1*^fl/fl^;*Adiponectin-cre* mice. Figure 5A is generated from 3 cre- and 4 cre+ mice. Figure 5B, C are generated from $N = 8$ cre- and 8 cre+ mice from 4 independent experiments. Figure 5D, G are generated from $N = 6$ cre- and 5 cre+ mice from 2 independent experiments. Figure 5E is generated from $N = 3$ cre- and 3 cre+ mice. Figure 5F, I are generated from $N = 6$ cre- and 5 cre+ mice from 2 independent experiments. Figure 5H is generated from $N = 5$ cre- and 6 cre+ mice from 3 independent experiments. Mann–Whitney test, two-tailed in **5A**, **5E** (right), **5F**, **5H**, **I**; 2-way ANOVA with Sidak multiple comparisons in **5B**, **C**. Scale bar = 50 µm.

of IL-18 in mediating the expression of intestinal lipid metabolism genes. Whereas IL-22RA1 deficiency solely reflects a lack of IL-20/IL-22/IL-24-mediated effects, IL-22 supplementation induces additional proteins/pathways such as IL-18. Upon intraperitoneal injection, we reproduced data indicating that IL-22 results in elevated ileal *Il18* transcript levels (Fig. 6C). We did not observe any significant difference in *Il18r1* transcripts after injection (Fig. 6C). Furthermore, we did not observe significant differences in *Il18* or *Il18r1* transcripts in the liver or WAT after IL-22.Fc injection (Supplementary Fig. 7G), indicating that intestinal IL-18 responses are more strongly regulated by IL-22.

Upon stimulating C57BL/6 small intestinal organoids with rIL-18, we observed significantly downregulated expression of *Lipe* and *Pnpla2* while *Ppara* showed a trend in the same direction (Fig. 6D). This implies that IL-22 may mediate some of its metabolic effects in conjunction with or by inducing IL-18, possibly explaining why systemic IL-22.Fc injection opposingly mediated gut and liver expression of *Lipe* and *Pnpla2*. These results may partially explain why we observed impaired hepatic functions in HFD-fed *Il22ra1*^fl/fl^;*Villin-cre*+ mice, but prior studies show that systemic IL-22 treatment mediates cytoprotective effects in the liver.

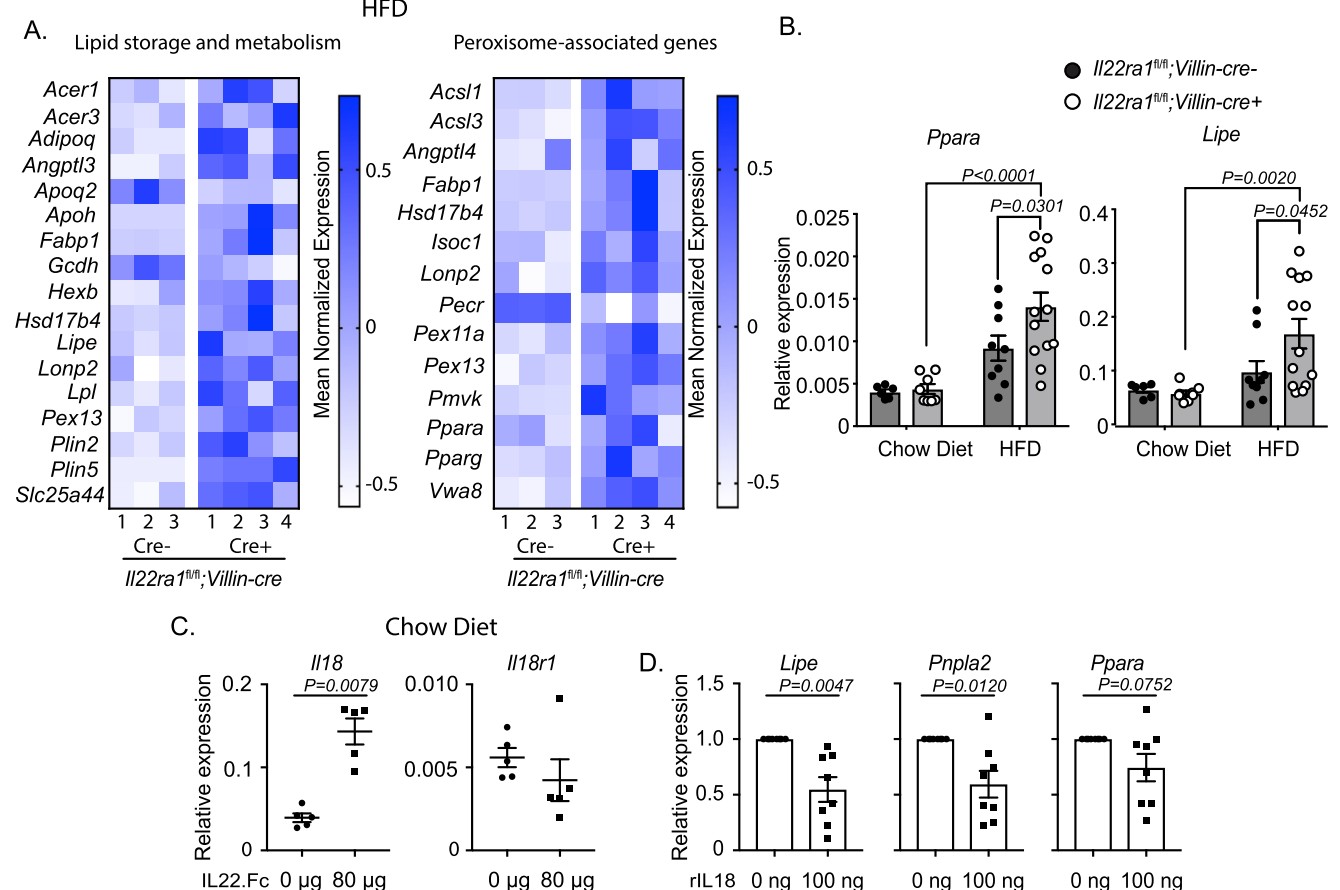

**Fig. 6 | Intestinal IL-22RA1 signaling downregulates the expression of lipid metabolism but not ISC- or progenitor cell-associated markers after HFD.**
**A** RNA-seq analysis of ileal tissue from *Il22ra1*[fl/fl];*Villin-cre* mice fed 16 weeks of HFD. Heatmaps of genes associated with lipid storage and metabolism (left) and peroxisome-associated genes (right) are shown. **B** RT-PCR analysis of lipid metabolism genes from ileal tissue of *Il22ra1*[fl/fl];*Villin-cre* mice fed 16 weeks of chow diet or HFD. **C** RT-PCR analysis of *Il18* and *Il18r1* from the ileum of wildtype mice treated with or without IL22.Fc. **D** RT-PCR analysis of lipid metabolism genes from small

intestinal C57BL6/J organoids treated with or without rIL18. Figure 6A is generated from *N* = 3 cre- and 4 cre+ mice from 3 independent experiments. Figure 6B is generated from *N* = 6 (chow diet, cre-), 8 (chow diet, cre+), 9 (HFD, cre-), and 13 (HFD, cre+) mice from 2 (chow) or 4 (HFD) independent experiments. Figure 6C is generated from *N* = 5 cre- and 5 cre+ mice from 2 independent experiments. Figure 6D is generated from *N* = 8 mice from 3 independent experiments. Data are presented as mean ± SEM in all graphs. Mann–Whitney test, two-tailed in 6 **C**, **D**; 2-way ANOVA with Sidak multiple comparisons in 6**B**.

## IL-22RA1 signaling is not critical for regulating Lgr5[+] ISC and progenitor cell-associated genes and epithelial proliferation during HFD

Thus far, we observed that IL-22RA1 signaling regulates the expression of intestinal lipid metabolism genes after HFD (Fig. 6A, B). Next, we examined how specific intestinal cell types may be affected by IL-22RA1 signaling to mediate these HFD-driven effects. HFD enhances crypt regeneration, reduces the dependence of Lgr5[+] ISCs on Paneth cell-derived growth factors, and enhances PPARδ/α-mediated programs that induce ISC and progenitor cell proliferation[36,37]. Unlike Lgr5[+] ISCs and progenitor cells, which display increased functionality after HFD, Paneth cells display impaired number and function[36,38]. To assess the effects of HFD on Lgr5[+] ISCs (Lgr5[+] high), progenitor cells (Lgr5[+] low), and Paneth cells, we analyzed publicly available RNA-seq data (GSE67324) from small intestinal sorted cells of *Lgr5-EGFP-IRES-CreERT2* mice fed a long-term (9–14 months) chow diet or HFD (Supplementary Fig. 8A)[36]. We assessed transcript levels of lipid storage and metabolism genes as well as peroxisome-associated genes based on our data from Fig. 6A. As expected, HFD resulted in upregulated function of many genes associated with lipid metabolism. Compared to Lgr5[+] ISCs and Paneth cells, progenitor cells generally displayed higher overall expression of many lipid metabolism- and peroxisome-associated genes including *Ppara*, *Ppard*, and *Pparg*

(Supplementary Fig. 8A). However, we did observe that the expression of several of these genes (including *Ppara* and *Ppard*) was upregulated in Paneth cells after HFD. Furthermore, expression of IL-22-associated genes (*Il10rb* and *Il22ra1*) was highest in Paneth cells (Supplementary Fig. 8A), which is consistent with our prior work[8]. We also observed that Paneth cell transcripts of *Il22ra1* increased after HFD, while no striking difference in expression was observed in progenitor cells and a modest decrease in expression was observed in ISCs (Supplementary Fig. 8A). Our analysis suggests that HFD regulates the transcriptional profile of lipid metabolism- and peroxisome-associated genes in Paneth cells and that IL-22RA1 signaling to Lgr5[+] ISCs, progenitor cells, and/or Paneth cells may regulate intestinal metabolism.

We next investigated whether IL-22 regulates Lgr5[+] ISC function after exposure to fatty acids ex vivo. Ileal organoids were grown from C57BL/6 mice over 6 days during which they were treated with or without 10 ng/mL rIL-22 or 30 μM palmitic acid (PA), a fatty acid found in HFD. Fewer numbers of organoids were observed after treatment with rIL-22 (Supplementary Fig. 8B), which has previously been reported[39]. RT-PCR analysis revealed no difference in *Lgr5* or *Olfm4* transcripts after stimulation with pure media, IL-22, PA, or PA + rIL-22 (Supplementary Fig. 8C). This suggests that rIL-22 stimulation either in the absence or presence of PA may not affect the ex vivo presence of Lgr5[+] ISCs. In line with these observations, we did not detect any

differences in transcripts of progenitor cell- (*Dll1*, *Dll4*, *Prom1*, *Zpf652*) or Lgr5+ ISC- (*Lgr5*, *Olfm4*, *Sox9*) associated genes in the ileum of HFD-fed *Il22ra1*<sup>fl/fl</sup>;*Villin-cre* mice (Supplementary Fig. 6D, E). No difference in ISC number was observed after staining ileal tissues from the control diet or HFD-fed *Il22ra1*<sup>fl/fl</sup>;*Villin-cre* mice for OLFM4 (Supplementary Fig. 8F). Likewise, no striking difference in cellular proliferation was observed upon staining ileal tissues for PCNA and examining *Mki67* transcript levels (Supplementary Fig. 8G). This suggests that IL-22RA1 signaling does not mediate epithelial proliferation, Lgr5+ cell stemness, or induction of progenitor cells in vivo during HFD. However, it is possible that IL-22RA1 signaling regulates other functions of Lgr5+ ISCs and progenitor cells after HFD or that IL-22RA1 signaling to other enterocytes such as Paneth cells plays an important role in regulating HFD-driven changes in metabolism.

### Intestinal IL-22RA1 signaling upregulates Paneth cell number and function to mediate systemic glucose metabolism during HFD

Although transcript levels of some Paneth cell lipid metabolism genes were upregulated after HFD (Supplementary Fig. 8A), the primary role of Paneth cells is to regulate the intestinal microbiota by producing shared antimicrobial peptides, including Reg3 proteins, and unique proteins such as Lyz1, α-defensins, and MMP7. Ileal tissues from HFD-fed *Il22ra1*<sup>fl/fl</sup>;*Villin-cre*+ mice displayed decreased expression of Paneth cell-associated antimicrobial genes after HFD (Fig. 7A). In addition, HFD-fed cre+ mice displayed decreased numbers of LYZ1+ cells (Fig. 7B, C). This indicates that intestinal IL-22RA1 signaling mediates Paneth cell antimicrobial programs and numbers after HFD.

Since Paneth cell-related programs are not maintained in *Il22ra1*<sup>fl/fl</sup>;*Villin-cre*+ mice after HFD (Fig. 7A–C), it is possible that IL-22RA1 signaling is important in mitigating the downregulatory effects that HFD has on the Paneth cell differentiation/transcriptional program of Lgr5+ ISCs and progenitor cells. To study this, we placed non-tamoxifen injected Lgr5 reporter mice (see methods) on HFD and control diet, intraperitoneally injected them with 0 or 80 μg IL-22.Fc, and sorted single Lgr5-GFP low (progenitor) and high (Lgr5+ ISCs) cells to assess their expression of Paneth cell-associated genes. The following gating strategy was used (Supplementary Fig. 9A). We determined the purity of Paneth cells (CD24+, UEA+) isolated from all sorted cells. Our sorted CD24+, UEA+ population was enriched for Paneth cells (*Lyz1*) upon RT-PCR analysis (Supplementary Fig. 9B). Fairly higher (but not significant) Paneth cell numbers were detected from mice injected with 80 μg IL-22.Fc (Supplementary Fig. 9A, C). We observed that IL-22.Fc treatment significantly upregulated the expression of *Mmp7* in Lgr5+ ISCs after HFD (Fig. 7D). However, IL-22.Fc treatment did not significantly upregulate the expression of *Mmp7* in progenitor cells or the expression of *Lyz1* in Lgr5+ ISCs or progenitor cells (Fig. 7D). Altogether, this suggests that IL-22 may not facilitate the differentiation of Lgr5+ ISCs or progenitor cells into Paneth cells.

Using primary organoid culture, we evaluated whether rIL-22 and PA treatment alter the expression of Paneth cell antimicrobial genes from C57BL/6 mice-derived ileal organoids. As expected, we observed a significant increase in *Lyz1* expression in the rIL-22 treated group when compared to the control group (Fig. 7E [left]). While no difference in *Lyz1* expression was noted between the control or PA-treated groups, we observed that the rIL-22-treated group possessed significantly greater *Lyz1* expression than the PA-treated group. While PA + rIL-22 treatment resulted in decreased *Lyz1* expression than the group treated only with rIL-22, this decrease was not as significantly low as the group treated only with PA (Fig. 7E [left]). Altogether, this indicates that while IL-22 induces Paneth cell antimicrobial gene expression, fatty acids (such as PA) can act to dampen this regulation. To assess whether this downregulation is a general phenomenon or is more specific to Paneth cells, we also examined the expression of *Reg3g*. As expected, transcript levels of *Reg3g* were elevated in the presence of rIL-22 (Fig. 7E [right]). Treatment with PA + rIL-22, however, still resulted in elevated expression of *Reg3g* comparable to that of the group treated only with rIL-22 (Fig. 7E [right]). This indicates that PA does not indeterminately reduce antimicrobial gene expression of enterocytes and that Paneth cell transcriptional functions may be more specifically regulated by the presence of fatty acids.

To determine whether Paneth cell number directly mediates systemic metabolism, we generated diphtheria toxin-depleted Paneth cell knockout (*Defa6-cre*+, *ROSA26*<sup>DTA</sup>) and littermate control (*Defa6-cre*-, *ROSA26*<sup>DTA</sup>) mice. Validation of knockout was confirmed (Supplementary Fig. 9D). Upon performing GTTs, we observed that Paneth cell-specific knockout mice fed a chow diet displayed impaired systemic glucose metabolism (Fig. 7F). To further elucidate the role of IL-22RA1 signaling to Paneth cells, we generated Paneth cell-specific *Il22ra1* knockout (*Il22ra1*<sup>fl/fl</sup>;*Defa6-cre*+) and littermate control mice. These mice were previously validated[8]. *Il22ra1*<sup>fl/fl</sup>;*Defa6-cre*- and *cre*+ mice displayed similar numbers of LYZ1+ cells after long-term control diet and HFD (Fig. 7G, H). We next assessed if Paneth cell-specific IL-22RA1 signaling is important for regulating systemic metabolism. As expected, *Il22ra1*<sup>fl/fl</sup>;*Defa6-cre*- and cre+ mice displayed similar weight gain when placed either on control diet or HFD (Fig. 7I; Supplementary Fig. 9E). Upon conducting GTTs, we did not observe any differences in glucose clearance or fasting glucose levels between control diet-fed *Il22ra1*<sup>fl/fl</sup>;*Defa6-cre* mice (Supplementary Fig. 9F, G). After HFD, *Il22ra1*<sup>fl/fl</sup>;*Defa6-cre*+ mice displayed significantly greater levels of blood glucose directly after fasting (Fig. 7J). However, glucose was similarly cleared between groups within 30 min after injection (Supplementary Fig. 9H). No differences in hepatic or WAT gene expression were observed in these mice after HFD (Supplementary Fig. 9I, J). Overall, our data suggest that IL-22 may act on Paneth cells to regulate their function and, in part, maintain a baseline glucose metabolism during HFD.

## Discussion

In this study, we elucidate the tissue-specific role of IL-22RA1 signaling in mediating the onset of HFD-driven metabolic disorders. We observed that IL-22RA1 signaling differentially mediates the metabolic functions of intestinal, liver, and adipose tissues. Furthermore, our work highlights a key role of intestinal IL-22RA1 signaling in regulating intestinal and extra-intestinal metabolism along with the underlying immune and microbial mechanisms that shape these outcomes.

Our study indicates that IL-22-mediated regulation of the intestinal microbiota mediates systemic metabolism during HFD. Furthermore, we observed that the absence of intestinal IL-22RA1 signaling during HFD is associated with a microbial dysbiosis marked by an increased abundance of *Oscillibacter* which promotes the accumulation of metabolically damaging macrophages in the WAT[31]. This coincided with our observation of elevated macrophage infiltration and Mmp12 levels in the WAT of *Il22ra1*<sup>fl/fl</sup>;*Villin-cre*+ mice, thereby supporting a protective role of intestinal IL-22RA1 signaling in mediating the microbiota to limit systemic WAT inflammation. Mechanistically, we also show that IL-22 may elicit its effects by activating IL-18. IL-22 protects against parasitic and bacterial infections by inducing epithelial IL-18[34,35]. IL-18 also protects against obesity and its associated systemic metabolic disorders[40,41]. While studies indicate that IL-18 can act on various sites, including skeletal muscle, adipose tissue, and the brain, to mediate metabolism and food intake, the direct metabolic role of intestinal IL-18 remained unclear[42]. Our study shows that IL-22 can also induce IL-18 to upregulate the expression of intestinal epithelial metabolic genes.

In addition, our study highlights a key regulatory role of Paneth cell-specific IL-22RA1 signaling. Prior studies illustrate the key role of IL-22 in mediating the differentiation of Paneth cells as well as their antimicrobial functions to regulate the intestinal microbiota and protect against infection[8,34,43]. However, we did not observe any

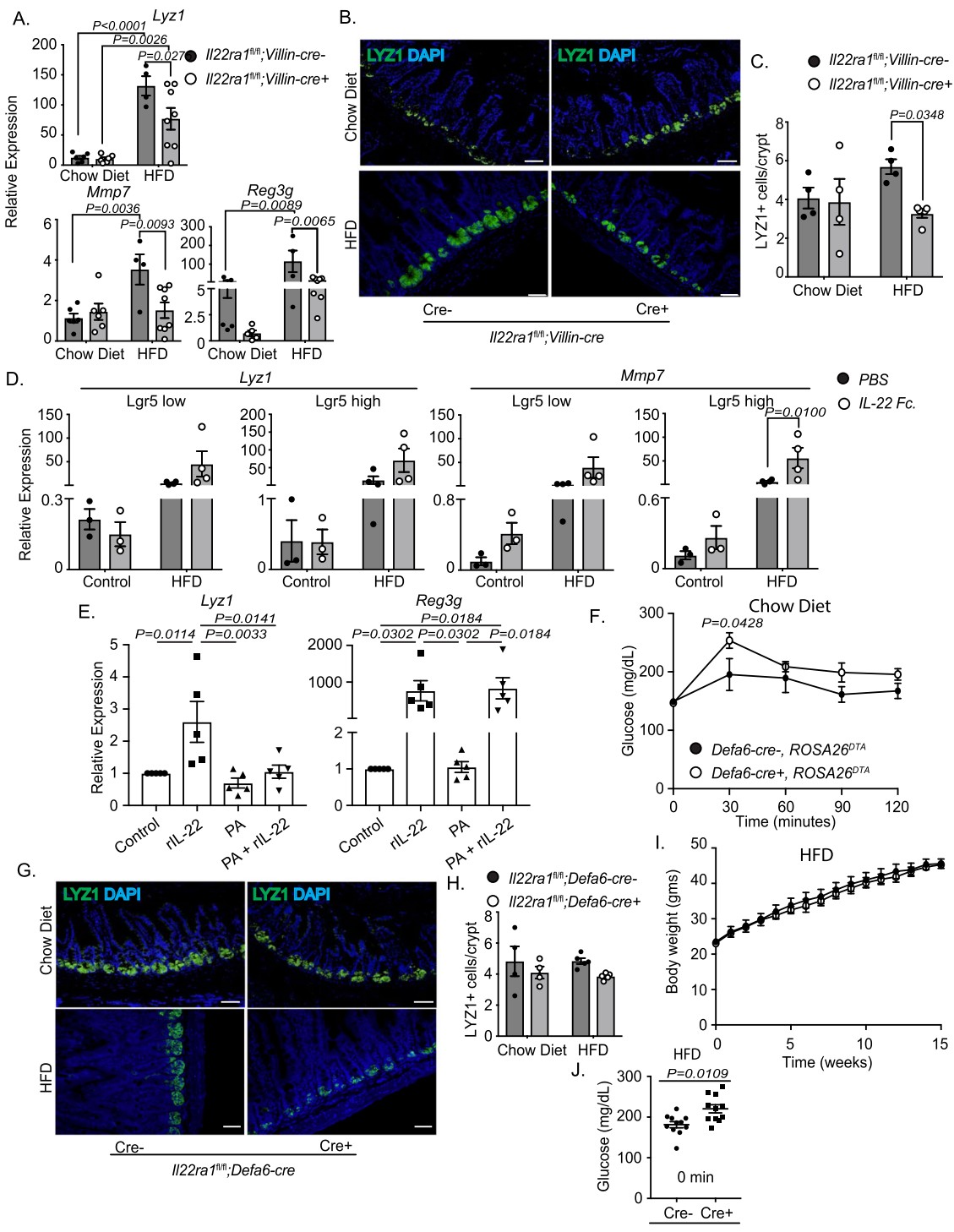

**Fig. 7 | Paneth cell-specific IL-22RA1 signaling mediates their antimicrobial functions and systemic fasting glucose metabolism after HFD. A** RT-PCR analysis of Paneth cell-associated antimicrobial genes from ileal tissues of *Il22ra1*^fl/fl*; Villin-cre* mice. **B** Representative immunofluorescence images of LYZ1 staining from *Il22ra1*^fl/fl*;Villin-cre* mice. **C** Average number of LYZ1+ cells/crypt for each mouse from Fig. 7B. **D** RT-PCR analysis of Paneth cell-associated genes from sorted Lgr5-GFP^high^ (ISC) and Lgr5-GFP^low^ (progenitor) cells from Lgr5 reporter mice. **E** RT-PCR analysis of organoids cultured with pure media, rIL-22, palmitic acid (PA), or PA and rIL-22. Data depict the fold change of the treated groups. **F** GTTs from chow diet-fed *Defa6-cre-/+, ROSA26*^DTA^ mice. **G** Representative immunofluorescence images of LYZ1 staining from *Il22ra1*^fl/fl*;Defa6-cre* mice. **H** Average number of LYZ1+ cells/crypt from Fig. 7G. **I** Weight gain of HFD-fed *Il22ra1*^fl/fl*;Defa6-cre* mice. **J** Glucose values from *Il22ra1*^fl/fl*;Defa6-cre* mice after fasting. Figure 7A represents *N* = 5–6 (chow diet, cre-), 5–6 (chow diet, cre+), 4 (HFD, cre-), and 8 (HFD, cre+) mice per group and 3 independent experiments. Figure 7B, C represent *N* = 4 (chow diet, cre-), 4 (chow diet, cre+), 4 (HFD, cre-), and 5 (HFD, cre+) mice from 2 independent experiments. Figure 7D represents *N* = 3 control diet-fed mice and 4 HFD-fed mice from 2 independent experiments. Figure 7E represents *N* = 5 mice from 2 independent experiments. Figure 7F represents *N* = 3 cre- and 4 cre+ mice from 2 independent experiments. Figure 7G, H represent *N* = 4 (chow diet, cre-), 4 (chow diet, cre+), 5 (HFD, cre-), and 5 (HFD, cre+) mice from 1 (chow) or 3 (HFD) independent experiments. Figure 7I represents *N* = 5 cre- and 7 cre+ mice from 4 independent experiments. Figure 7J represents *N* = 11 cre- and cre+ mice from 4 independent experiments. Data are presented as mean ± SEM in all graphs. Mann–Whitney test, two-tailed in 7**J**; 1-way ANOVA with Tukey multiple comparisons in 7**E**; 2-way ANOVA with Sidak multiple comparisons in 7**A**, 7**C**, 7**D**, 7**F**, 7**H**, 7**I**. Scale bar = 20 μm.

differences in LYZ1+ cell number in *Il22ra1*fl/fl;*Defa6-cre*+ and littermate control mice. One reason for this could be the difference in age between these mice. The previously published study by our group used 6–8-week-old mice, whereas the present study uses mice fed a long-term diet (~23 weeks old at euthanasia)[8]. With an increase in age, there could be a compensatory mechanism by which Paneth cell numbers are maintained. Therefore, the role of age-related changes in cytokine levels and their effect on Paneth cell numbers should be investigated. Our study further explores the regulatory role of IL-22 on Paneth cells and provides evidence that Paneth cell-specific IL-22RA1 signaling is important for mediating baseline systemic glucose metabolism post-HFD. It should be noted that IL-22RA1 signaling may directly mediate glucose metabolism by Paneth cells since Paneth cells are glycolytic cell types and IL-22 has been shown to upregulate glycolysis[4,44–46]. Furthermore, IL-22RA1 signaling to other enterocytes likely plays a critical role in facilitating systemic glucose metabolism. It is possible that intestinal IL-22RA1 signaling regulates systemic metabolism by mediating the production of certain metabolic hormones by enteroendocrine cells. For instance, IL-22 regulates the production of peptide tyrosine tyrosine (PYY), a hormone important for increasing insulin secretion and glucose metabolism[4,47,48]. While PYY is produced by enteroendocrine cells, it has also been shown to be produced by Paneth cells[48,49]. Therefore, IL-22RA1 signaling may directly induce PYY produced from these cells to regulate systemic glucose metabolism.

It should be noted that we analyzed publicly available RNA-sequencing data from the small intestine (Supplementary Fig. 8A) and, therefore, discrepancies between different facilities and experimental protocols (diet duration/composition, housing conditions, microbiota composition) may exist to influence sequencing results. In addition, while our study evaluated the role of IL-22RA1 signaling to WAT, we did not assess how IL-22 may specifically mediate the function of brown adipose tissue, a key mediator of energy expenditure and lipid metabolism. Regarding further examination of intestinal lipid and peroxisome pathways that may be mediated by IL-22RA1 signaling, it should be noted that the WAT and liver also express these genes and associated pathways. Therefore, blockade experiments to study the effects of a given pathway in *Il22ra1*fl/fl;*Villin-cre*-/+ mice will also impact liver/ WAT functions. An ideal approach would be to use intestinal epithelium-specific knockouts of a suspected IL-22 target such as Ppara. Lastly, how tissue-specific IL-22RA1 signaling regulates weight gain remains an open question.

Overall, the crosstalk between immune and metabolic functions in the intestinal epithelium observed herein has a potential relevance for human physiology. Indeed, similar regulatory interactions have been observed in patients with common variable immunodeficiency, HIV enteropathy, and celiac disease[50–52]. Future studies would benefit from studying how IL-22RA1 signaling to additional metabolic organs, such as the pancreas, mediates obesity-associated disorders. Finally, our results elucidate how IL-22 or IL-22RA1 can be used to develop future tissue-specific treatments for obesity-associated metabolic disorders.

## Methods

### Mice
C57BL/6J (WT), Albumin-cre (C57BL/6J background), Villin-Cre (C57BL/ 6J background), Adiponectin-Cre (C57BL/6J background), and ROSA-DTA (C57BL/6J background) mice were purchased from The Jackson Laboratory. Germ-free C57BL/6Tac mice (male, 6-week-old) were ordered from Taconic Biosciences. *Il22*−/− mice received from Dr. Jay Kolls, Tulane University. WT and *Il22*−/− mice cohoused for 2 weeks and then separated for the entire duration of HFD. Lgr5-EGFP-creERT2 mice (C57BL/6 background) were provided by Dr. Vincent W. Yang, Stony Brook University. Defa6-Cre mice (C57BL/6 background) were provided by Dr. Richard Blumberg, Brigham and Women's Hospital, Harvard. Generation and characterization of IL-22-Floxed (*Il22ra1*fl/fl)

mice were performed as described[53]. *Il22ra1*fl/fl mice were bred with Albumin-cre, Villin-cre, Adiponectin-cre, Lgr5-EGFP-creERT2, or Defa6-cre mice to generate entire liver-, intestinal epithelium-, WAT-, Lgr5+ ISC-, or Paneth cell-specific IL-22RA1 knockout and littermate control mice. Defa6-cre mice were bred with ROSA-DTA mice to generate Paneth cell-specific knockout and littermate control mice. For all HFD and control diet experiments, we placed 6-week-old male mice on 16 weeks of HFD (Bio-Serv, S3282) or control diet (Bio-Serv, F4031). Littermate mice were used in all experiments and mice were separated based on cre status before placement on HFD, chow diet (PicoLab, Rodent Diet 5053), or control diet. For cohousing experiments, mice were not separated based on cre status after weaning. Mice were housed at Stony Brook University, Stony Brook, NY under specific pathogen-free conditions. For all ex vivo culture experiments, male and female mice were at least 6 weeks old when used. All mice were housed under a 12/12 light/dark cycle, 64–79 °F, and 30–70% humidity. For all experiments, mice were euthanized with $CO_2$. Animal studies were conducted with the approval and under all relevant ethical regulations of Stony Brook University's Institutional Animal Care and Use Committee (IACUC: 968871).

### Cell lines
HepG2 cells, a human hepatoma cell line, were obtained from Dr. Patrick Hearing at Stony Brook University, Stony Brook, NY. Conditioned organoid media was prepared from cultured L-WRN cells as described[54].

### Animal treatment
IL-22.Fc (F652; Evive Biotech, China) was reconstituted in sterile 1x PBS. Either 0 ug (pure PBS) or 80 µg IL-22.Fc was intraperitoneally injected in chow diet-fed wild-type mice. Mice were euthanized 24 h post injection and tissues were harvested for relevant analysis.

Wildtype, *Il22*−/−, *Il22ra1*fl/fl;*Albumin-cre*, *Il22ra1*fl/fl;*Villin-cre*, *Il22ra1*fl/fl; *Adiponectin-cre*, and *Il22ra1*fl/fl;*Defa6-cre* mice (male, 6 weeks old) were placed on 16 weeks of HFD. A control diet was used wherever indicated. Mice were weighed weekly, feces were collected before (day 0) and after (day 115) HFD treatment and GTTs were conducted after 16 weeks of treatment. To conduct GTTs, mice were weighed and then fasted for 5 h in clean cages. After this period, an initial blood glucose reading (0 min) was recorded by obtaining a blood sample from the tail. A Contour next glucose monitor and test strips (Bayer) were used to obtain blood glucose readings. Mice were then intraperitoneally injected with a body weight-specific volume (equivalent to half of the body weight) of glucose (200 mg/mL, MP Biomedicals) as previously published. Blood glucose readings were obtained every 30 min over a 2 h period. GTTs for *Defa6-cre*-/+, *ROSA26*DTA mice were conducted on male and female chow diet-fed mice (10-16 weeks old). Insulin tolerance tests were conducted using a modified protocol described for GTTs. Mice were fasted for 5 h in a clean cage and then intraperitoneally injected with 0.75 IU/kg body weight insulin (Lilly). Glucose readings were obtained at 0 (fasting), 15, 30, 45, 60, and 90 min post-injection.

Mice were treated with antibiotics using methods similar to those previously described[55]. Briefly, drinking water was substituted with an antibiotics cocktail containing vancomycin (0.5 g/L), ampicillin (0.5 g/L), neomycin (1 g/L), and metronidazole (1 g/L) on week 12 of HFD. Mice received antibiotics-supplemented drinking water for 4 weeks. For cecal content transfer experiments, 6-week-old male germ-free C57BL/6NTac mice were purchased from Taconic Biosciences. Cecal contents from either *Il22ra1*fl/fl;*Villin-cre*- or cre+ mice fed 16 weeks of HFD were pooled and homogenized. Upon arrival, germ-free mice were housed in freshly autoclaved cages, fed a sterilized control diet, and gavaged with 200 µL 0.1 g/mL cecal contents (day 0). Contents were additionally given on days 2 and 4. GTTs were conducted on day 7.

*Lgr5-EGFP-IRES-CreERT2*, *Il22ra1*fl/fl;*Villin-cre-/+*, and *Il22ra1*fl/fl;*Albumin-cre-/+* mice were placed on HFD for 6 weeks. *Lgr5-EGFP-IRES-CreERT2* mice were intraperitoneally injected with either 0 or 80 µg IL-22.Fc every other day starting on week 5 (4 injections total). *Il22ra1*fl/fl;*Villin-cre-/+* and *Il22ra1*fl/fl;*Albumin-cre-/+* mice were intraperitoneally injected with 80 µg IL-22.Fc every other day starting on week 5 (4 injections total). Body weight was recorded at the indicated time points. Mice were euthanized and tissues were harvested for transcriptional analyses.

## Reverse transcriptase-polymerase chain reaction (RT-PCR)

Total RNA was extracted from the liver, WAT, and terminal ileum tissues using Trizol-based isolation. Total RNA was isolated from organoids using an RNeasy Mini Kit (QIAGEN). Following the manufacturer's instructions, cDNA was synthesized using Bio-Rad iScript kits. RT-PCR assays were performed by mixing 5 µL of SsoAdvanced™ Universal Probes Supermix (Bio-Rad) or SsoAdvanced™ Universal SYBR Green Supermix (Bio-Rad), 4.5 µL of cDNA, and 0.5 µL of primers/probes per well. Expression of genes of interest was calculated relative to *Hprt* or *Gapdh*. Fold change (derived from ΔΔC$_T$ analysis) is calculated for individual litters. Control groups for organoid experiments are set to a baseline expression of 1.0 to clearly assess the relative fold change (derived from ΔΔC$_T$ analysis) of each treatment on a biological sample. Refer to Supplementary Table 2 for the full list of primers.

## Histopathology

For H&E staining, 5 µm paraffin-embedded tissue sections were cut, deparaffinized using xylene (2 washes), and rehydrated through a descending ethanol gradient (100%, 95%, 70%, 2x pure dH$_2$O). Hematoxylin (VWR) was added to all slides for 1 min. Slides were then washed in running tap water for approximately 15 min, incubated in dH$_2$O for 30 s, treated with 95% ethanol for 1 min, and stained with eosin (VWR) for 1 min. Staining was briefly evaluated under a microscope and slides were transferred to 95% and 100% ethanol for 2 min each followed by xylene for 2 min. Slides were mounted with toluene and images were acquired. Crypt/villus length was quantified using ImageJ.

For alcian blue staining, tissue sections were prepared as described above. Sections were covered with 3% acetic acid for 3 min. Acetic acid was removed and sections were stained with alcian blue for 30 min. Sections were washed with 3% acetic acid for 15 s. Slides were rinsed in running tap water for 10 min. Sections were stained with Nuclear Fast Red solution (Electron Microscopy Sciences) for 6 min. Slides were rinsed in running tap water for 2 min. Sections were dehydrated with an ascending ethanol wash (70%, 95%, and 100%) followed by xylene for 2 min each. Slides were mounted with toluene and images were acquired.

ORO tissue staining was performed as previously described[55]. Briefly, liver tissues were harvested and frozen in an OCT compound medium (Sakura). Cryosections (5 µm) were cut. Before staining, tissues were dried at room temperature for 10 min. A 3:2 ratio solution of ORO Isopropanol Solution (Electron Microscopy Sciences):dH$_2$O was applied to the slides for 5 min. Slides were washed in running tap water for 30 min. Slides were dried and mounted with glycerol/gelatin-based medium. Images were acquired. Percent area coverage of ORO staining per image was quantified using ImageJ.

Scoring of H&E-stained WAT was performed by Dr. Kenneth R. Shroyer. A scale from 0–4 was used to subjectively score the degree of inflammation (i.e., the presence of crown-like structures). Scale: 0, no or minimal inflammation, 1 indicates low inflammation; 2, moderate inflammation; 3, high inflammation; and 4, severe inflammation.

## Culture of murine intestinal organoids

The murine organoid culture was performed similarly to what we previously described[56]. Briefly, small intestinal ileal tissues were harvested from C57BL/6 mice. Peyer's patches and connective tissues were removed. Intestinal tissues were flushed with ice-cold 1x PBS, cut longitudinally, and quickly washed in 1x PBS. Tissues were cut into approximately 1 cm pieces and washed in cold 1x PBS until the liquid was predominantly clear of debris (usually 3–4 washes). To isolate intestinal crypts, tissues were incubated at 4 °C on an orbital shaker for 30 min in 1x PBS containing 3.75 mM EDTA. Tissues were washed once in ice-cold 1x PBS to remove residual EDTA. Fresh ice-cold 1x PBS was added and the tissues were shaken to promote the release of crypts into the supernatant. The supernatant was passed into a new tube over a 70 µm cell strainer. To collect more crypts, this step was repeated two more times with gradually stronger shaking. Crypts were then pelleted down at 200 x *g* and each sample's overall number was calculated.

For plating in a 24-well plate, crypts were resuspended in Matrigel Matrix (Corning) at a concentration of 80–100 crypts per 30 µL Matrigel. Plated Matrigel domes were kept at 37 °C, 5% CO2 for 10 min until they fully polymerized. Domes were overlaid with 500 µL of Organoid Growth Medium (1:1 mixture of L-WRN and DMEM/F12). The following media supplements were added: 1x penicillin-streptomycin-glutamine (Invitrogen), 1x N2 (Invitrogen), 1x B27 (Invitrogen), 50 ng/mL EGF (R&D system), 1 mM N-acetylcysteine (Sigma-Aldrich), 10 nM Gastrin-Leu15 (Sigma-Aldrich), 500 nM A83-01 (Torics), 10 µM Y-27632 (Sigma-Aldrich), and 10 µM of CHIR99021 (Tocris). The media was changed every 2 days. Y-27632 and CHIR99021 were not added to the media after day 2.

For Figs. 1I and 6D, organoids were either treated with pure media, rIL-22 (10 ng; R&D Systems), or rIL-18 (100 ng; R&D Systems) starting on day 2. On day 5, organoids were lysed in RLT buffer (QIAGEN) containing 1% β-mercaptoethanol for RNA isolation. For Fig. 7E and Supplementary Fig. 8B, C, organoids were either treated with pure media, rIL-22 (10 ng/mL; R&D Systems), PA (30 µM; Sigma), or rIL-22 and PA starting on day 0. On day 6, brightfield images were taken and organoids were lysed in RLT buffer (QIAGEN) containing 1% β-mercaptoethanol for RNA isolation.

## Single-cell isolation and sorting

*Il22ra1*fl/fl;*Lgr5-EGFP-creERT2*+ or *Il17ra*fl/fl;*Lgr5-EGFP-creERT2*+ mice were placed on control diet or HFD for 1 week. These mice were not injected with tamoxifen and, therefore, were useful for specifically sorting Lgr5$^+$ ISC (Lgr5-GFP$^{high}$) and progenitor (Lgr5-GFP$^{low}$) cells based on GFP expression. Mice were injected with IL-22.Fc or PBS 72 h and 24 h prior to being euthanized. Small intestinal crypts were isolated as discussed in the prior section one day after the last IL-22 injection. Once crypts were isolated, they were incubated in DMEM/F12 (Gibco) containing 20% FBS for 15 min at 4 °C. Crypts were dissociated into single cells upon incubation with TrypLE Express (Invitrogen) containing 10 µM Y-27632 (Sigma-Aldrich) and 10 µg/mL DNase I (Roche) for 5 min at 37 °C after which an equal volume DMEM/F12 was added. Cells were pipetted to further promote dissociation. Cells were washed with 1x PBS and pelleted by centrifugation at 4 °C at 200 xg for 5 min. Cells were stained with Aqua Dead Cell Stain Kit (1:500; Invitrogen; Ref: L34957) for 30 min to exclude dead cells, anti-CD45:APC-eFluor780 (1:200; Invitrogen; Ref: 47–0451) to exclude CD45+ cells, and anti-CD24:PE-Cyanine7 (1:200; Invitrogen; Ref: 25–0242) and UEA1:DyLight649 (1:200; Vector Laboratories; Ref: DL-1068-1) to sort Paneth cells. All antibodies were validated by their manufacturer and/or our lab. Cells were processed for fluorescence-activated cell sorting (FACS) sorting of Lgr5-GFP$^{hi}$ and Lgr5-GFP$^{low}$ cells using a BD FACSAria. Cells were sorted into RLT buffer (QIAGEN) containing 1% β-mercaptoethanol for RNA isolation.

## Immunofluorescence (tissues and organoids)

Liver, adipose, and intestinal tissues were harvested and fixed in 10% formalin for 24 h. Tissues were processed and embedded in paraffin

blocks. Sections (5 µm) were cut. Sections were deparaffinized and rehydrated using xylene (2 washes) and a descending ethanol gradient (100%, 95%, 70%, 2x pure dH2O). Antigen retrieval was performed by incubating tissues in citric acid buffer and placing them in a steamer for 40 min. Tissues were cooled to room temperature, encircled with an IHC PAP pen (ENZO Life Sciences), and permeabilized (10 min) with 1x PBS containing 0.1% Triton X-100 (Sigma-Aldrich). Tissues were washed 3x (5 min) with 1x PBS containing 0.01% Tween 20 (VWR). Tissues were blocked in 1x PBS containing 5% bovine serum albumin for 30 min at 37 °C. Tissues were incubated overnight at 4 °C in 1x PBS containing 5% bovine serum albumin and primary antibodies against LYZ1-FITC (1:200; Dako; Ref: F0372), OLFM4 (1:200; Cell Signaling Technology; Ref: 39141S), PCNA (1:200; Santa Cruz Biotechnology; Ref: Sc-56), anti-F4/80-FITC (1:50; Miltenyi Biotec; Ref: 130-117-509), MMP12 (1:50; Abcam; Ref: ab231109). Slides were washed 3x with 1x PBS. Tissues were incubated with anti-rabbit IgG Fab2 Alexa Fluor® 488 (1:200; Cell Signaling Technology; Ref: 4412). All antibodies were validated by their manufacturer and/or our lab. DAPI hard stain mounting media (Vector® Laboratories) was applied to visualize cell nuclei. Images were acquired using an Olympus CKX41 microscope.

For ORO staining, organoids were cultured in the presence or absence of 5 ng rIL-22 starting on day 2. After 24 h (day 3), organoids were treated with 4% LM. Organoids were dissociated from the Matrigel with cell recovery media (Corning). Organoids were placed on an orbital shaker at 4 °C for 5 min. Organoids were washed in DMEM, 10% FBS, spun down at 180 xg, and fixed with 4% paraformaldehyde (Electron Microscopy Sciences) for 30 min. Organoids were briefly washed with 1x PBS and then 60% isopropanol. ORO solution (similar to what we used for tissue staining) was added for 20 minutes. The solution was carefully removed, leaving organoids undisturbed at the bottom of the tube. Organoids were washed in 1x PBS. Pipette tips were coated with 5% bovine serum albumin to prevent organoids from sticking. Organoids were transferred to a slide and were encircled by an IHC PAP pen. DAPI hard stain mounting media (Vector® Laboratories) was applied to visualize cell nuclei. Images were acquired using an Olympus CKX41 microscope.

## Cell line assays

HepG2 cells were grown until 70% confluency in a T-75 flask, dissociated into single cells, washed, and plated (200 µL) at equal concentrations in a 96-well plate. At 60–70% confluency, cells were stimulated with or without 80 ng IL-22.Fc. After 24 h, cells were treated with 4% LM. After 48 h, cells were washed and fixed with 4% paraformaldehyde. Cells were washed with 60% isopropanol and stained with ORO (similar to what we previously used) for 10 min. Cells were washed with dH2O until the supernatant was clear (approximately 2x). ORO stain was extracted from cells using 250 µL 100% isopropanol. Extracted dye (200 µL) was plated in a clean well and its OD of 490 nm was measured.

## Gas chromatography

The Folch-Lees method of lipid extraction was used[57]. Extracts were filtered. Lipids were isolated in chloroform and separated via thin-layer chromatography. Separation was performed using Silica Gel 60 A plates developed in petroleum ether, ethyl ether, and acetic acid (80:20:1). Separation was visualized with rhodamine 6G. As performed by Morrison and Smith, phospholipids, diglycerides, triglycerides, and cholesteryl esters were isolated and methylated using BF3/methanol[58]. Methylated fatty acids were extracted and analyzed by gas chromatographic analyses using an Agilent 7890 A gas chromatograph with flame ionization detectors and capillary column (SP2380, 0.25 mm×30 m, 0.25 µm film, Supelco, Bellefonte, PA). Helium was used as a carrier gas and oven temperature was set from 160–230 °C at 4 °C/min. To determine fatty acid methyl esters, sample

and standard retention times were compared. Standards with odd-chain fatty acids were used to quantify sample lipids. The following standards were used: dipentadecanoyl phosphatidylcholine (C15:0), diheptadecanoin (C17:0), trieicosenoin (C20:1), and cholesteryl eicosenoate (C20:1).

## 16S rRNA microbial community analysis

Microbial community analysis was performed as previously described[8]. Feces were collected from mice before (day 0) and after (day 105) HFD treatment. Fecal DNA was isolated using a QiAMP stool DNA extraction kit (QIAGEN). Analysis utilizing PCR amplification of the V4 region of 16 S rRNA followed by sequencing on an Illumina MiSeq was performed by MR DNA (Shallowater, TX, USA). Identification of gut microbial amplicon sequence variants (ASVs) followed by chimera removal was performed with DADA2 (v1.21). Taxonomy assignment was performed using the Silva Project's version 138.1 release and formatted for use with DADA2. The downstream and statistical analysis was carried out with Phyloseq (v1.34.0), quantile normalization, and differential abundance using BRB-ArrayTools (https://brb.nci.nih.gov/BRBArrayTools/) and alpha and beta diversity using MicrobiomeAnalyst[59]. Raw sequencing data was deposited in the NCBI Gene Expression Omnibus.

## RNA sequencing (RNA-seq)

RNA-seq analysis of RNA isolated from terminal ileum tissue was performed as we previously described[56]. Ileal tissues were collected from *Il22ra1*[fl/fl];*Villin-cre* mice after 16 weeks of HFD. RNA was isolated via Trizol extraction methods and then purified for sequencing.

Small intestinal RNA-sequencing data previously deposited in the Gene Expression Omnibus (GEO) database (accession number: GSE67324) was analyzed. Methods for this sequencing were previously described[36]. Small intestinal crypts were isolated from *Lgr5-EGFP-IRES-CreERT2* mice fed an HFD or control diet for 9–14 months. Briefly, crypts were dissociated with TrypLe Express (Invitrogen) and stained with appropriate markers for sorting. ISCs were sorted as Lgr5-EGFP$^{high}$Epcam$^+$CD24$^{low/-}$CD31$^-$Ter119$^-$CD45$^-$7-AAD$^-$. Lgr5-EGFP$^{low}$ progenitor cells were sorted as EGFP$^{low}$Epcam$^+$CD24$^{low/-}$CD31$^-$Ter119$^-$CD45$^-$7-AAD$^-$. Paneth cells were sorted as CD24$^{high}$Sidescatter$^{high}$Lgr5-EGFP$^-$Epcam$^+$CD31$^-$Ter119$^-$CD45$^-$7-AAD$^-$. Viability dye 7-AAD (Life Technologies) was used to exclude dead cells[36].

## Luminex multiplex assay

Serum was isolated from HFD-fed mice and analyzed via multiplex analysis (Luminex) using the Mouse Metabolic Hormone Panel (MilliporeSigma) for c-peptide and insulin and TH17 Panel (Millipore Sigma) for IL-22. Samples were run at the Stony Brook School of Medicine Genomics Core (Bio-Plex 200) and Vanderbilt University Hormone Assay and Analytical Services Core (MagPix).

## Promoter analysis

The Eukaryotic Promoter Database was used to identify the transcriptional start site for murine *Ppara*. Sequence output was set to 2000 base pairs upstream of the transcriptional start site. The retrieved sequence was validated via NCBI. The upstream sequence was then analyzed using the JASPAR Core database to identify putative binding sites for murine Stat3.

## Statistics and reproducibility

Statistics were conducted using GraphPad Version 7.05.237 software. The following tests were used as indicated in figure legends to determine $p$-values: Mann–Whitney test (two-tailed), Mann–Whitney test (one-sided), Unpaired $t$-test (one-tailed), 2-way ANOVA with Sidak multiple comparisons, and one-way ANOVA with Tukey multiple comparisons. For all tests, $p < 0.05$ was considered significant. Data are presented as mean ± standard error or standard deviation as indicated

in the figure legends. At least 2 experimental replicates were performed whenever possible to verify reproducibility. Replicate number is indicated in the figure legends. Experimental sample size and replicates are indicated in the text. Data points were excluded if they were statistical outliers as determined by using GraphPad Prism (ROUT analysis, $Q = 1\%$). Data acquisition and analysis were performed blind whenever possible.

## Reporting summary

Further information on research design is available in the Nature Portfolio Reporting Summary linked to this article.

## Data availability

The 16S rRNA microbial sequencing data are deposited in the NCBI BioProject database (accession code: PRJNA944165). The RNA-sequencing data from small intestinal tissues of HFD-fed $Il22ra1^{fl/fl}$; $Villin\text{-}cre\text{-}/+$ mice are deposited in the Sequence Read Archive (accession code: PRJNA1055251). The previously published small intestinal RNA-sequencing data from control or HFD-fed $Lgr5\text{-}EGFP\text{-}IRES\text{-}CreERT2$ mice are deposited in GEO (accession code: GSE67324). Relevant data underlying all non-sequencing-related findings are available in the Source Data file. Any additional information is available upon request to the corresponding author (Pawan Kumar, pawan.kumar@stonybrook.edu). Source data are provided in this paper.

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

## Acknowledgements

We would like to acknowledge the Flow Cytometry Core, Research Histology Core, and Genomics Core at Stony Brook University. We would like to acknowledge the RNA-sequencing core at Cold Spring Harbor Laboratory. We would like to acknowledge Vanderbilt University Medical Center's Hormone Assay and Analytical Services Core. This work was supported by the NIHAI149257 to P.K., NRSA T32 training grant (T32AI007539) to S.J.G., and NIHDK088199 to R.B.. P.K. is supported by the Crohn's and Colitis Foundation (476637), NIH R01 DK121798-01, and NIH R21 AI146696. We thank Jay Kolls (Tulane University), Jeremy McAleer (Marshall University), and Vishal Singh (Pennsylvania State University) for critically reading the manuscript and providing expert advice.

## Author contributions

P.K. conceptualized the idea. S.J.G., H.H., and P.K. designed the experiments. S.J.G. and P.K. wrote the manuscript. S.J.G. and M.J.-P. performed H&E staining and RNA isolation. K.S. assisted with RNA isolation. S.J.G. and A.S. performed flow cytometry and RT-PCR experiments. S.J.G., A.S., and C.K. performed immunofluorescence staining. S.J.G. performed ORO staining, alcian blue staining, ELISA, crypt and single-cell isolation, DSS experiments, and cell/organoid culture. T.B. assisted with organoid culture. R.B. provided mouse lines for generating Paneth cell-specific Il22ra1 knockouts. K.R.S. evaluated H&E slides of WAT for inflammation. S.B. provided small intestinal RNA-seq data from Lgr5-EGFP-IRES-CreERT2 mice. J.P., N.S., and A.M. analyzed and interpreted the 16S rRNA sequencing data. All authors discussed data and commented on the manuscript.

## Competing interests

The authors declare no competing interests.
