## [Peer Review File · Nature Communications]

REVIEWER COMMENTS

Reviewer #1 (Remarks to the Author):

IL-22/IL-22Ra1 signaling is known to regulate lipid metabolism which is associated with metabolic disorders such as in the condition of HFD. Due to the distribution of IL-22Ra1 expression in the gut, liver, and adipose tissues, in this study, Gaudino et al. aimed to use tissue-specific IL22Ra1 conditional knockout mice to address where and how IL-22Ra1 signaling exerts its regulatory function towards lipid/glucose metabolism. To this end, the authors first found systemic IL22.Fc administration can cause differential regulation of genes related to lipid metabolism in liver vs ileum tissues (Fig.1). Then they found systemic glucose, but not weight gain, is controlled by gut- and liver-specific IL-22Ra1 signaling (Fig.2,3). Then they found the gut IL-22Ra1 signaling contributes to extra-intestinal lipid metabolism (liver and WAT) and this intestine-mediated regulation is via microbiota (Fig.3,4). Intriguingly, the authors claimed that WAT-specific IL-22Ra1 signaling controls weight gain in a combined model of HFD and DSS colitis (Fig.5). Next, by RNA seq analysis of HFD ileal tissues, the authors identified that intestinal IL-22Ra1 signaling could directly regulate gut lipid/glucose metabolism (Fig.6) and this regulation is likely acting on Paneth cells by affecting Lgr5+ stem and progenitor cells (Fig.7). Overall, the characterization of HFD models with tissue-specific conditional knockouts covers many different aspects but unfortunately they did not integrate the results into a clear and convincing outcome. The authors also did not provide mechanistic studies to explain the IL-22Ra-mediated upregulation or downregulation of lipid-associated genes in respective tissues at the molecular levels. Apart from technical issues described below, the title suggests that it is IL-22Ra1 signaling in Paneth cells to be crucial for HFD-induced metabolic disorders, but IL-22Ra1 signaling in liver or WAT also play a role in certain HFD conditions (actually the analyses with liver- and WAT-specific KO were kind of superficial). The major weakness is that the authors did not have a direct and convincing evidence to claim that IL-22 directly target Paneth cells to regulate metabolic disorders both in vitro and in vivo (Line 419-420), which I think could be a challenging task to test in their current experimental settings. While the novelty is somewhat compromised by several prior publications (J Hepatol 53:339, 2010, Nature 514:237, 2014, Nature 531:53, 2016), the concept of using tissue-specific IL22Ra1 conditional knockout to dissect the function of IL-22Ra1 signaling during HFD is unique and this area is of potential interest. However, it appears that experiment design, data quality, and indepth of analysis can not sufficiently support their major conclusion.

Major Comments

(01) The authors claimed that systemic treatment with IL-22 (Line 98-99) may not be a good way to dissect tissue-specific functions of IL-22. However, the results (Fig.1D,1E,1G,1H,7E) were still obtained from systemic i.p. injection of IL22.Fc. As tissues were harvested 24 hr post injection (Line 628), it is not clear that the effect of systemic IL-22 on the respective tissues is a direct or indirect effect and thus the conclusion can not be drawn convincingly, as was also claimed by the authors at Line 178-179. To exclude

an effect of IL22.Fc on liver or WAT, the authors should consider using Vil-Cre+Il22ra1f/f mice for i.p. injection as controls in those Exp.

(02) Fig.2 and Fig.3 are conceptually linked and should be combined. Otherwise, it is advised to present all data related to Vil-Cre+Il22ra f/f mice in Fig.2, and then all data related to Albumin-Cre+Il22ra f/f mice in Fig.3. Furthermore, results (Fig.3A-3C, 3F, 3G-I) should also be collected and compared by using HFD-fed Albumin-Cre+Il22ra1f/f mice. This is crucial since the authors aimed to dissect the tissue-specific function of IL-22Ra1 and to conclude that Albumin-Cre+Il22ra1f/f mice displayed different trends (i.e. the tissue-specific effect) in lipid metabolism compared to Vil-Cre+Il22ra1f/f mice (Line 234-235). The authors should also discuss the results (Fig.2-3) here and correlate to those findings in Fig.1 (Fig.1E vs 1H).

(03) Some in vitro culture could be performed to show the direct effect of IL-22 on lipid metabolism in respective tissues (as a gain-of-function approach, not just using KO), such as using IL-22-treated adipocytes (refer to Fig.4i-k, Nature 514:237, 2014), IL-22-treated HepG2 (such as Fig.1F), or IL-22-treated epithelial cells. Supporting this, the authors may try injecting IL22.Fc to Il22ra1f/f mice vs Adiponectin-Cre+Il22ra1f/f (for Fig.1D), Albumin-Cre+Il22ra1f/f (for Fig.1E), or Vil-Cre+Il22ra1f/f (for Fig.1H) mice, to show the tissue-specific effect of IL-22Ra1 signaling.

(04) The authors claimed that intestinal IL-22Ra1 signaling mediates microbiota composition which then regulates metabolic and inflammatory functions of extra-intestinal tissues (Line 303-304, which is based on Fig.4). As there are prior publications showing that IL-22Ra1 signaling regulates HFD-induced metabolic disorder (J Hepatol 53:339, 2010, Nature 514:237, 2014) and that microbiota contributes to NAFLD/NASH (EMBO Mol Med 11:e9302, 2019), the authors unfortunately add little novelty here by just performing antibiotics treatment in HFD-fed mice but did not provide clues about the potential “microbial identity” or microbiota-derived “products”, which might function as a crucial mediator linking the gut to the liver or to adipose tissues. Furthermore, this microbiota-dependent regulation also indicates that alterations of lipid or glucose metabolism programs (genes) in liver or WAT (Fig.4D-H) is not mediated by the respective IL-22Ra1 signaling in liver or WAT tissues, but a consequence effect of gut-derived microbiota. This somehow strays away from the original to-be-tested concept that there is a tissue-specific function of IL-22Ra1 signaling.

(05) The authors claimed that colons from DSS-treated Adiponectin-Cre+ Il22ra1f/f mice are better defined (Line 337). However, data interpretation of Fig.5G is not standardized and sufficiently convincing. Quantification of data (such as disease severity score) or alternative analytical methods (by confocal images) need to be provided. What was the time points for the H&E images shown in Fig.5G (go with Fig 4D or Fig.5H) ? In most cases, DSS colitic mice with 20% weight loss usually will cause significant damage (or even death) in the colon which could be easily identified by H&E staining.

(06) Recovery from DSS treatment greatly depends on epithelial tissue regeneration or repairing in the colon. As such, Fig.5F-G showed that regeneration/repair from DSS-induced colon injury is much better in Adiponectin-Cre+ Il22ra1f/f mice that likely leads to more resistant in percent weight loss (Fig.5D). However, the authors did not provide any clues (could be a WAT-to-gut feedback effect, which is a novel observation) about why adipose IL-22Ra1 signaling may affect colon regeneration which is a key factor why Adiponectin-Cre+ Il22ra1f/f mice are more resistant, if total epididymal fat-pad mass is not significantly increased in these mice. Data in Fig.6 appear not well-integrated (or separated from) to other data shown in Fig.2-3, and therefore it is kind of superficial analysis without any mechanistic insights.

(07) While Fig.6A-B show notable observations between WT vs KO, the authors unfortunately did not expand those observations to explore the underlying mechanisms, regarding how intestinal IL-22Ra1 signaling regulates those target genes related to lipid storage or peroxisome at the molecular levels. IL-22 was reported to regulate lipid metabolism in a previous study (refer to Fig.4 in Nature 514:237, 2014), thus, more insights is somewhat expected here not just comparing differentially expressed genes between WT/KO. For example, how does Ppara/Lipe levels contribute to weight gain, glucose clearance, lipid accumulation in the gut vs liver? Injection of IL22.Fc by i.p. downregulates Ppara/Pparg in the ileum of WT mice (Fig.1H), so how to prove that IL-22Ra1 signaling can improve glucose clearance or reduce lipid accumulation in the respective tissues via regulating these gene programs? I'll suggest to combine observations in Fig.1H and 6B, by performing new Exp with 4 groups of mice with and without IL22.Fc (or with some inhibitors within the IL-22Ra1-Ppara/Pparg signaling pathway) under HFD, to reveal some potential IL-22Ra1-mediated mechanistic pathway.

(08) As IL-22 (refer to Immunity 40:262, 2014) and intestinal IL-22Ra1 signaling (Fig.4) regulate microbiota community, care should be taken when using dataset mining from other groups (Fig.6C, 7D) where the HFD protocol or microbiota could be different due to housing conditions of the respective vivarium. The authors may need to justify whether other groups' HFD protocol is the same and appropriate. Data quality of Fig.6F is not good (refer to related data in Fig.2b-c in Nat Comm 13:874, 2022). Because of these concerns and being not a major finding, I suggest moving Fig.6 to supplementary data section.

(09) The authors claimed that intestinal IL-22Ra1 signaling is important in mediating Paneth cell function and numbers during HFD (Line 430-431, based on Fig.7A-C). However, the difference already exists between WT/KO mice at the steady state as shown by a previous paper from the authors (refer to Fig.2 in Mucosal Imm 14:389, 2020). Whether or not the difference is larger during HFD needs to be confirmed with new Exp using 4 groups of mice. To reveal a specific function of IL-22Ra1 signaling in Paneth cells during HFD, normal chow and HFD-fed Il22ra1f/f vs Defa6-Cre+ Il22ra1f/f mice should be used (Fig.7G-J), as there are already defects in Paneth cells in Defa6-Cre+Il22ra1f/f mice at the steady state (refer to Fig.4 in Mucosal Imm 14:389, 2020).

(10) The main conclusion (as noted by the title: Intestinal IL-22Ra1 upregulates Paneth cell differentiation of Lgr5+ stem and progenitor cells to ameliorate high fat diet-induced metabolic disorders) appears entirely based on the result of Fig.7E (IL-22Ra1 signaling is not triggered, therefore may not be involved, in the datasets of Fig.6C and 7D). In my opinion, Fig.7E is simply not sufficient and convincing to support the major conclusion of paper. First, while Lgr5-eGFP reporter mice were on HFD, the authors did not show the treatment with IL22.Fc indeed ameliorate metabolic disorders (with QPCR, GTT, or image data shown in Fig.1~3) under this condition, with Lgr5+ cells being sorted from the same mice. Second, QPCR data alone is not convincing evidence. Approach such as confocal images or electron microscopy should be provided to strengthen the conclusion. Third, systemic i.p. injection of IL22.Fc is not specific to the gut. How do we know IL22.Fc is affecting the gut (and Lgr5+ cells) and then causing alterations in the liver/WAT? An approach to address this could be using HFD-fed Lgr5-eGFP;Vil-Cre+IL22ra1f/f mice for i.p. injection as controls.

(11) Conceptually, I am confused about the title and final statement (IL-22Ra1 signaling to Paneth cells is important for maintaining their antimicrobial functions and mediating systemic glucose metabolism, Line 419-420). Do the authors want to conclude that a) IL-22Ra1 signaling targets Lgr5+ stem and progenitor cells to affect Paneth cell differentiation from these cells for ameliorating metabolic disorders (by title), or b) IL-22Ra1 signaling directly targets Paneth cells to regulate lipid/glucose metabolism (by the final statement)? Apparently, conclusion (a) and (b) are different things.

Minor Comments

(01) Last paragraph in the Introduction (Line 105-125) is more like a conclusion or summary of results. I suggest to simplify the statement here or move to the discussion section.

(02) Some result texts appear to be lengthy, containing too much background, rationale, methodology, or discussion. It is advised to keep concise and focused to the results.

(03) To be consistent with symbols used in Fig.2B, filled and open triangle as symbols for HFD-fed WT vs KO mice should also be used in Fig.2C, 2E, 2F, 4C, and 4D.

(04) Duplicate references (Line 1095-1097 and Line 1098-1100).

(05) The authors did not explore the possibility that IL-22Ra1 signaling may target and regulate brown adipose tissues (BAT) which also govern energy or lipid metabolism like WAT. WAT is the main site for

storing excess fuel while BAT is specific for energy dissipation (Sci Transl Med 12:eaaz8664, 2020). Do we know adiponectin also drive Cre expression in BAT? It is well-known that both BAT and WAT contribute to systemic energy homeostasis, both could be linked to weight gain in a macroscopic assessment.

Reviewer #2 (Remarks to the Author):

In this manuscript by Stephen J, et al, the authors explored IL-22 mediated signaling in different tissue including intestine, liver and adipose tissue for metabolic disorders induced by HFD using tissue specific IL-22 ko mice. Since IL-22 is an important cytokine for mediating gut homeostasis and could influence host metabolism, it will be important and interesting to define the detailed mechanisms by which IL-22 impact host metabolism, especially in HFD context. While interestingly, this manuscript was only described, it is not clear how intestinal IL-22, gut microbiota influence host metabolisms in HFD-fed mice. Major and minor comments are outlined below:

1. the authors observed that IL-22 signaling in intestine and liver are important to regulate systemic glucose metabolism in HFD, but not in control diet condition. Did HFD influence IL-22 expression in these tissues?
2. In Figure 2, Although the author did not observe a difference in weight gain between control diet and HFD, it is still important to have the fat pad mass or percentage to show whether there is difference in obesity for these mice.
3. Also in Figure 2, the authors showed that Villin-cre+ mice displayed impaired glucose clearance when compared to their littermate controls by glucose tolerance test, did these mice have difference in insulin resistance?
4. the authors mostly only compare the data in HFD condition, seems lack the control diet. For example, In Figure 3, the control diet data should be included to compare with HFD, which may be used to explain why this phenotype only be observed in HFD. Also, is Albumin-cre mice have different expression of lipid metabolism genes in Figure 3?
5. In Figure 3F, the authors showed that Villin-cre+ mice displayed significantly greater accumulation of crown-like structures than their littermate controls by HE staining of fat, while accumulation of crown-like structures are often evaluated by immunostaining of macrophages, not just HE staining.
6. It was interestingly that IL-22 upregulates hepatic lipid metabolism while downregulating intestinal lipid metabolism, could the authors discuss it?
7. According to composition of control diet, it seems the control diet is compositional defined diet rather than grain-based chow diet, but the authors mentioned several times of chow diet in the manuscript. Please clarify.

8. For microbiota analysis by 16Ssequencing, it will be helpful to include more data about alpha diversity and beta diversity.
9. For antibiotic treatment, did the mice with and without antibiotics have difference about GTT at week 12 before antibiotic treatment? Fecal microbiota transplantation may be needed to conform the conclusion that intestinal IL-22 RA1 signaling mediates extra-intestinal metabolism in a microbiota Intestinal.
10. In Figure 5, the data about Il22ra1fl/fl;Adiponectin-cre+ for DSS was confused to include in the manuscript. Did DSS or HFD induce IL-22 expression in fat?
11. In Figure 1, is any rationale that the dose of IL-22-Fc injection is 80ug?
12. Did Albumin-cre+ mice have difference in body weight and glucose tolerance test when fed control diet?
13. The methodology is mostly sound, yet some important information is missing, e.g., the mice age at which initial HFD.
14. Some statement miss references, such as, lines 63-72.
15. In Figure 5G, the colonic pathological score needs to be caculated.
16. In Figure 7J, the whole GTT data should presented rather glucose level at 0 min.
17. In line 510, the author think that weight gain may be more clearly observed under stronger inflammatory conditions (such as DSS-induced colitis), this statement may be not correct, since DSS induced colitis will cause loss of body weight.
18. In line 536, is one "reduced" should be " increased"?

Reviewer #3 (Remarks to the Author):

The systematic effects of IL-22 on metabolism have been well documented. However, how IL-22/IL-22RA signaling in different organs contribute to glucose and lipid metabolism is not clear. In this study, the authors used several lines of IL-22RA conditional KO mice to characterize the role of IL-22/IL-22RA signaling on metabolism in the liver, intestine and adipose tissue. Here are some concerns:

1. In Fig.1E, the authors observed IL-22Fc increased the expression of lipolysis related genes Lipe and Pnpla2, which was contradict with the results in Fig.1F which indicated that IL-22Fc increased lipid accumulation in HepG2 cells? Also, in Fig.1H, IL-22Fc reduced the expression of Lipe and Pnpla2 in ileum, but inhibits lipid accumulation in intestinal organoids? For the impact of IL-22 on WAT, the authors only

checked several lipid metabolism related gene expression, but how IL-22 directly influences WAT lipid metabolism is not clear. The genes selected mainly focused on lipolysis and beta oxidation, genes related to lipid synthesis and transport should be included in this study.

2. In Figure 2, the authors did not see the difference in bodyweight between control and two of the IL-22RA conditional KO mice, however, in Figure3, more lipid accumulation were observed in the liver in Il22ra1fl/fl;Villin-cre mice. More biochemical markers such as serum TG, CHOL levels need to be evaluated.

3. In Fig.3A, Cre- mice with HFD should have significant steatosis in H&E staining. Fig.3C, more genes related to lipid metabolism should be included to evaluated the impact of IL-22RA signaling on lipid metabolism.

4. Will lack of IL-22RA in WAT contribute to the size and weight change in WAT when fed on HFD?

5. Did the author observed any difference in lipid metabolism in the liver and WAT in Il22ra1fl/fl;Defa6-cre+ mice with HFD?

Reviewer 1:

Remarks to the Author: IL-22/IL-22Ra1 signaling is known to regulate lipid metabolism which is associated with metabolic disorders such as in the condition of HFD. Due to the distribution of IL-22Ra1 expression in the gut, liver, and adipose tissues, in this study, Gaudino et al. aimed to use tissue-specific IL22Ra1 conditional knockout mice to address where and how IL-22Ra1 signaling exerts its regulatory function towards lipid/glucose metabolism. To this end, the authors first found systemic IL22.Fc administration can cause differential regulation of genes related to lipid metabolism in liver vs ileum tissues (Fig.1). Then they found systemic glucose, but not weight gain, is controlled by gut- and liver-specific IL-22Ra1 signaling (Fig.2,3). Then they found the gut IL-22Ra1 signaling contributes to extra-intestinal lipid metabolism (liver and WAT) and this intestine-mediated regulation is via microbiota (Fig.3,4). Intriguingly, the authors claimed that WAT-specific IL-22Ra1 signaling controls weight gain in a combined model of HFD and DSS colitis (Fig.5). Next, by RNA seq analysis of HFD ileal tissues, the authors identified that intestinal IL-22Ra1 signaling could directly regulate gut lipid/glucose metabolism (Fig.6) and this regulation is likely acting on Paneth cells by affecting Lgr5+ stem and progenitor cells (Fig.7). Overall, the characterization of HFD models with tissue-specific conditional knockouts covers many different aspects but unfortunately they did not integrate the results into a clear and convincing outcome. The authors also did not provide mechanistic studies to explain the IL-22Ra-mediated upregulation or downregulation of lipid-associated genes in respective tissues at the molecular levels. Apart from technical issues described below, the title suggests that it is IL-22Ra1 signaling in Paneth cells to be crucial for HFD-induced metabolic disorders, but IL-22Ra1 signaling in liver or WAT also play a role in certain HFD conditions (actually the analyses with liver- and WAT-specific KO were kind of superficial). The major weakness is that the authors did not have a direct and convincing evidence to claim that IL-22 directly target Paneth cells to regulate metabolic disorders both in vitro and in vivo (Line 419-420), which I think could be a challenging task to test in their current experimental settings. While the novelty is somewhat compromised by several prior publications (J Hepatol 53:339, 2010, Nature 514:237, 2014, Nature 531:53, 2016), the concept of using tissue-specific IL22Ra1 conditional knockout to dissect the function of IL-22Ra1 signaling during HFD is unique and this area is of potential interest. However, it appears that experiment design, data quality, and in depth of analysis can not sufficiently support their major conclusion.

Major Comments

- 1. The authors claimed that systemic treatment with IL-22 (Line 98-99) may not be a good way to dissect tissue-specific functions of IL-22. However, the results (Fig.1D,1E,1G,1H,7E) were still obtained from systemic i.p. injection of IL22.Fc. As tissues were harvested 24 hr post injection (Line 628), it is not clear that the effect of systemic IL-22 on the respective tissues is a direct or indirect effect and thus the conclusion can not be drawn convincingly, as was also claimed by the authors at Line 178-179. To exclude an effect of IL22.Fc on liver or WAT, the authors should consider using Vil-Cre+IL22ra1f/f mice for i.p. injection as controls in those Exp.**

We agree with the reviewer's comment and have included additional data assessing a direct role of IL-22 in specifically mediating intestinal metabolism. In figure 1I, we stimulated wildtype (C57BL/6) small intestinal organoids with 0 or 10 ng rIL-22. We observed that organoids stimulated with rIL-22 displayed significantly decreased expression of the same genes (*Pnpla2*,

Lipe, *Ppara*, *Pparg*) that were downregulated when we intraperitoneally injected wildtype mice with IL22.Fc (lines 169-172). Our ex vivo results suggest that IL-22 acts directly on the intestinal epithelium to regulate the expression of these genes. To further assess the tissue-specific role of IL-22 signaling, we placed control (*Lgr5-cre^{Egfp}*), *Il22ra1^{fl/fl};Villin-cre*, and *Il22ra1^{fl/fl};Albumin-cre* mice on 6 weeks of high fat diet. Starting week 5, *Il22ra1^{fl/fl};Villin-cre* and *Il22ra1^{fl/fl};Albumin-cre* mice were intraperitoneally injected with 80 µg IL22.Fc every other day. In addition, control (*Lgr5-cre^{Egfp}*) mice were placed on high fat diet and were injected with either 0 or 80 µg IL22.Fc every other day (Supplementary Fig. 7B-E). Since no difference in the expression of WAT metabolic genes was observed upon injecting wildtype mice with IL22.Fc, we did not inject *Il22ra1^{fl/fl};Adiponectin-cre* mice. We observed that ileal expression of *Ppara*, *Lipe*, and *Pnpla2* was dependent on intestinal epithelium-specific (but not liver-specific) IL-22RA1 signaling (lines 387-401). Overall, these results implicate a direct role of intestinal IL-22RA1 signaling in mediating the expression of intestinal lipid metabolism genes.

2. **Fig.2 and Fig.3 are conceptually linked and should be combined. Otherwise, it is advised to present all data related to Vil-Cre+Il22ra f/f mice in Fig.2, and then all data related to Albumin-Cre+Il22ra f/f mice in Fig.3. Furthermore, results (Fig.3A-3C, 3F, 3G-I) should also be collected and compared by using HFD-fed Albumin-Cre+Il22ra1f/f mice. This is crucial since the authors aimed to dissect the tissue-specific function of IL-22Ra1 and to conclude that Albumin-Cre+Il22ra1f/f mice displayed different trends (i.e. the tissue-specific effect) in lipid metabolism compared to Vil-Cre+Il22ra1f/f mice (Line 234-235). The authors should also discuss the results (Fig.2-3) here and correlate to those findings in Fig.1 (Fig.1E vs 1H).**

All data related to systemic changes in *Il22ra1^{fl/fl};Villin-cre* mice were placed in Fig. 2 and Supplementary Fig. 2-3. Likewise, all data related to systemic changes in *Il22ra1^{fl/fl};Albumin-cre* mice were placed in Fig. 3 and Supplementary Fig. 4. In addition, we further explored the systemic metabolic effects occurring in *Il22ra1^{fl/fl};Albumin-cre* mice. This includes performing H&E staining (Fig. 3D, G) and RT-PCR analysis (Fig. 3E, I; Supplementary Fig. 4B) for liver and WAT samples. Our findings in Fig. 3 correlated with those from figure 1, namely regarding the expression of *Ppara* (lines 251-253).

3. **Some in vitro culture could be performed to show the direct effect of IL-22 on lipid metabolism in respective tissues (as a gain-of-function approach, not just using KO), such as using IL-22-treated adipocytes (refer to Fig.4i-k, Nature 514:237, 2014), IL-22-treated HepG2 (such as Fig.1F), or IL-22-treated epithelial cells. Supporting this, the authors may try injecting IL22.Fc to Il22ra1f/f mice vs Adiponectin-Cre+Il22ra1f/f (for Fig.1D), Albumin-Cre+Il22ra1f/f (for Fig.1E), or Vil-Cre+Il22ra1f/f (for Fig.1H) mice, to show the tissue-specific effect of IL-22Ra1 signaling.**

We agree with the reviewer's comment and performed additional experiments to test the direct effects of IL-22 on lipid metabolism. As described in comment 1, we cultured small intestinal organoids and treated them with or without rIL-22 (lines 169-172). Also described in comment 1, we intraperitoneally injected *Il22ra1^{fl/fl};Villin-cre* and *Il22ra1^{fl/fl};Albumin-cre* mice with 80 µg IL22.Fc and control (*Lgr5-cre^{Egfp}*) mice with either 0 or 80 µg IL22.Fc (Supplementary Fig. 7B-E; lines 387-401).

4. The authors claimed that intestinal IL-22Ra1 signaling mediates microbiota composition which then regulates metabolic and inflammatory functions of extra-intestinal tissues (Line 303-304, which is based on Fig.4). As there are prior publications showing that IL-22Ra1 signaling regulates HFD-induced metabolic disorder (J Hepatol 53:339, 2010, Nature 514:237, 2014) and that microbiota contributes to NAFLD/NASH (EMBO Mol Med 11:e9302, 2019), the authors unfortunately add little novelty here by just performing antibiotics treatment in HFD-fed mice but did not provide clues about the potential “microbial identity” or microbiota-derived “products”, which might function as a crucial mediator linking the gut to the liver or to adipose tissues. Furthermore, this microbiota-dependent regulation also indicates that alterations of lipid or glucose metabolism programs (genes) in liver or WAT (Fig.4D-H) is not mediated by the respective IL-22Ra1 signaling in liver or WAT tissues, but a consequence effect of gut-derived microbiota. This somehow strays away from the original to-be-tested concept that there is a tissue-specific function of IL-22Ra1 signaling.

We performed additional experiments to examine how IL-22-mediated changes in microbiota composition may influence systemic metabolism as well as what changes in microbial identity may specifically be playing an important role. To assess what microbial changes may be occurring in *Il22ra1^{fl/fl};Villin-cre^{-/+}* mice before and after high fat diet, we revised/expanded our bacterial genera analysis. First, we included additional data examining changes in alpha and beta diversity (Supplementary Fig. 5A-B). Although no distinct differences were observed between cre- and cre+ at similar time points, we observed changes before and after exposure to high fat diet (lines 274-277).

One of such genera upregulated after high fat diet was *Oscillibacter*, which has been associated with obesity-induced metabolic phenotypes and insulin resistance (Li et al, 2022). Our results show that *Il22ra1^{fl/fl};Villin-cre⁺* mice displayed elevated levels of *Oscillibacter* when compared to cre- mice and that this difference is greatly expanded after high fat diet (Fig. 4B; Supplementary Fig. 5C-D; lines 277-281). *Oscillibacter* has been shown to induce *Mmp12* expression in WAT-associated macrophages leading to impaired glucose metabolism (Li et al, 2022). Coinciding with our data indicating that *Il22ra1^{fl/fl};Villin-cre⁺* mice possess elevated WAT inflammation and F4/80 staining (Fig. 2H-I; Supplementary Fig. 3B), we observed elevated levels of *Mmp12* in WAT from *Il22ra1^{fl/fl};Villin-cre⁺* mice (Fig. 4C; lines 281-289). Altogether, this implicates a potential role of IL-22 in downregulating *Oscillibacter* levels to protect against systemic inflammation.

To further assess whether IL-22-mediated changes in microbiota composition contribute to impaired metabolism, we performed two additional experiments. First, we performed a cecal microbiota transfer from contents collected from high fat diet (16 weeks) fed *Il22ra1^{fl/fl};Villin-cre^{-/+}* mice into 6-week germ-free mice. Mice were gavaged 3 times with contents, and GTTs were performed 1 week after the first gavage. We observed that germ-free mice that received cre+ cecal homogenate displayed impaired glucose clearance, further indicating that IL-22-mediated regulation of microbiota composition regulates systemic glucose metabolism. Second, we cohoused *Il22ra1^{fl/fl};Villin-cre^{-/+}* mice during their 16 weeks of high fat diet. Upon performing

GTTS, we still observed that *Il22ra1^{fl/fl};Villin-cre+* mice did not clear glucose as effectively as their cre- littermate controls (lines 297-307).

References:

- Li, Z. *et al.* Microbiota and adipocyte mitochondrial damage in type 2 diabetes are linked by *Mmp12* + macrophages. *Journal of Experimental Medicine* **219**, e20220017 (2022).

5. The authors claimed that colons from DSS-treated Adiponectin-Cre+ *Il22ra1f/f* mice are better defined (Line 337). However, data interpretation of Fig.5G is not standardized and sufficiently convincing. Quantification of data (such as disease severity score) or alternative analytical methods (by confocal images) need to be provided. What was the time points for the H&E images shown in Fig.5G (go with Fig 4D or Fig.5H)? In most cases, DSS colitic mice with 20% weight loss usually will cause significant damage (or even death) in the colon which could be easily identified by H&E staining.

To additionally quantify colonic damage, we measured colonic crypt/villus length of DSS-treated *Il22ra1^{fl/fl};Adiponectin-cre* mice. Correlating with their impaired colon length (Fig. 5G), we observed that *Il22ra1^{fl/fl};Adiponectin-cre+* mice also displayed shorter crypt/villus structures (Fig. 5I; lines 354-356). We also examined epithelial proliferation by staining for PCNA. We observed that *Il22ra1^{fl/fl};Adiponectin-cre+* mice generally displayed greater epithelial proliferation post DSS (Supplementary Fig. 6B-C; lines 356-358); however, this difference was not significant. In addition, we observed that *Il22ra1^{fl/fl};Adiponectin-cre+* mice displayed increased goblet cell numbers when compared to their cre- littermate controls (Fig. 5J-K; lines 358-361). Reduced goblet cell numbers typically correlate with worse intestinal inflammation (van der Post et al, 2019; Gersemann et al, 2009). In addition, we clarified the experimental conditions for the H&E images (line 354).

References:

- van der Post, S. *et al.* Structural weakening of the colonic mucus barrier is an early event in ulcerative colitis pathogenesis. *Gut* **68**, 2142–2151 (2019).
- Gersemann, M. *et al.* Differences in goblet cell differentiation between Crohn's disease and ulcerative colitis. *Differentiation* **77**, 84–94 (2009).

6. Recovery from DSS treatment greatly depends on epithelial tissue regeneration or repairing in the colon. As such, Fig.5F-G showed that regeneration/repair from DSS-induced colon injury is much better in Adiponectin-Cre+ *Il22ra1f/f* mice that likely leads to more resistant in percent weight loss (Fig.5D). However, the authors did not provide any clues (could be a WAT-to-gut feedback effect, which is a novel observation) about why adipose IL-22Ra1 signaling may affect colon regeneration which is a key factor why Adiponectin-Cre+ *Il22ra1f/f* mice are more resistant, if total epididymal fat-pad mass is not significantly increased in these mice. Data in Fig.6 appear not well-integrated (or separated from) to other data shown in Fig.2-3, and therefore it is kind of superficial analysis without any mechanistic insights.

We discuss that WAT has been shown to mediate intestinal functions under inflammatory conditions. Previous studies indicate that elevated visceral and mesenteric adipose tissue are present in patients with inflammatory bowel disease (Rowan et al, 2021; Karaskova et al, 2021). In addition, WAT function can be mediated by the microbiota after intestinal inflammation. Adipocytes express receptors that detect microbial antigens, and translocation of the gut microbiota to mesenteric adipose tissue has been shown to promote the occurrence of hyperplastic mesenteric adipose tissue (i.e., “creeping fat”), an associated phenotype of Crohn’s disease (Ha et al, 2020; Peyrin-Biroulet, 2019). This is discussed in lines 343-349.

Furthermore, we examined additional factors that may contribute to why *Il22ra1^{fl/fl};Adiponectin-cre+* mice experience less severe DSS-induced weight loss. To assess differences in epithelial proliferation between chow fed *Il22ra1^{fl/fl};Adiponectin-cre-/+* mice post DSS, we stained their colonic tissues for PCNA as described in comment 5 (Supplementary Fig. 6B-C; lines 356-358). We also observed that chow fed *Il22ra1^{fl/fl};Adiponectin-cre+* mice displayed an increased number of colonic goblet cells post DSS when compared to their *cre-* littermate controls as described in comment 5 (Fig. 5J-K; lines 358-361).

Altogether, we discuss the importance of WAT-to-gut feedback in mediating inflammation and included new observations identifying additional altered intestinal functions that may explain why *Il22ra1^{fl/fl};Adiponectin-cre+* mice display less severe DSS-induced injury. Although it is possible that IL-22 acts via visceral or creeping fat to mediate intestinal inflammation, the exact mechanisms underlying the basis of these interactions is beyond the scope of our paper.

References:

- Rowan, C. R., McManus, J., Boland, K. & O’Toole, A. Visceral adiposity and inflammatory bowel disease. *Int J Colorectal Dis* **36**, 2305–2319 (2021).
 - Karaskova, E. *et al.* Role of Adipose Tissue in Inflammatory Bowel Disease. *IJMS* **22**, 4226 (2021).
 - Ha, C. W. Y. *et al.* Translocation of Viable Gut Microbiota to Mesenteric Adipose Drives Formation of Creeping Fat in Humans. *Cell* **183**, 666-683.e17 (2020).
 - Peyrin-Biroulet, L. *et al.* Mesenteric fat in Crohn’s disease: a pathogenetic hallmark or an innocent bystander? *Gut* **56**, 577–583 (2007).
7. While Fig.6A-B show notable observations between WT vs KO, the authors unfortunately did not expand those observations to explore the underlying mechanisms, regarding how intestinal IL-22Ra1 signaling regulates those target genes related to lipid storage or peroxisome at the molecular levels. IL-22 was reported to regulate lipid metabolism in a previous study (refer to Fig.4 in *Nature* 514:237, 2014), thus, more insights is somewhat expected here not just comparing differentially expressed genes between WT/KO. For example, how does Ppara/Lipe levels contribute to weight gain, glucose clearance, lipid accumulation in the gut vs liver?

Injection of IL22.Fc by i.p. downregulates Ppara/Pparg in the ileum of WT mice (Fig.1H), so how to prove that IL-22Ra1 signaling can improve glucose clearance or reduce lipid accumulation in the respective tissues via regulating these gene programs? I'll suggest to combine observations in Fig.1H and 6B, by performing new Exp with 4 groups of mice with and without IL22.Fc (or with some inhibitors within the IL-22Ra1-Ppara/Pparg signaling pathway) under HFD, to reveal some potential IL-22Ra1-mediated mechanistic pathway.

As suggested, we performed additional experiments with four groups of mice treated with or without IL-22.Fc to provide mechanistic insight regarding how intestinal IL-22RA1 signaling regulates the expression of key lipid metabolism genes. Please refer to our response for comment #1 and Supplementary Fig. 7B-E.

Next, we assessed what pathways may be regulated by IL-22 to mediate the expression of these lipid metabolism genes in the intestine. Since IL-22 can activate Stat3 and we detected putative Stat3 binding sites upstream of *Ppara* (Supplementary Fig. 7F), we first assessed whether IL-22 acts via Stat3 to regulate the expression of *Ppara*. To do so, we treated wildtype small intestinal organoids with either control media, rIL-22, S3I-201 (Stat3 inhibitor), or both rIL-22 and S3I-201. We observed a modest reduction in *Ppara* expression after treatment of organoids with S3I-201 (Stat3 inhibitor) and a significant reduction after treatment with rIL-22. However, treatment of S3I-201 did not reverse or enhance the inhibitory effects of rIL-22 on the expression of *Ppara*, indicating that IL-22 may not be responsible for mediating *Ppara* transcription in a Stat3-dependent manner (Supplementary Fig. 7G; lines 402-409).

Since IL-22 has also been shown to induce the production of epithelial IL-18 to mediate its protective effects (Chiang et al, 2022; Muñoz et al, 2015), we evaluated whether IL-22 may induce IL-18 to mediate the expression of intestinal lipid metabolism genes. First, we observed that IL-22 results in elevated ileal *Il18* transcript levels in IL-22.Fc injected wildtype mice (Fig. 6C) which also displayed decreased levels of *Pnpla2*, *Lipe*, *Ppara*, and *Pparg* (Fig. 1G). Furthermore, upon stimulating wildtype small intestinal organoids with rIL-18, we observed significantly downregulated expression of *Lipe* and *Pnpla2* but not *Ppara* (Fig. 6D). This implies that IL-22 may mediate some of its metabolic effects in conjunction with or by inducing IL-18. These results are described in lines 410-423.

References:

- Chiang, H.-Y. *et al.* IL-22 initiates an IL-18-dependent epithelial response circuit to enforce intestinal host defence. *Nat Commun* **13**, 874 (2022).
- Muñoz, M. *et al.* Interleukin-22 Induces Interleukin-18 Expression from Epithelial Cells during Intestinal Infection. *Immunity* **42**, 321–331 (2015).

8. **As IL-22 (refer to Immunity 40:262, 2014) and intestinal IL-22Ra1 signaling (Fig.4) regulate microbiota community, care should be taken when using dataset mining from other groups (Fig.6C, 7D) where the HFD protocol or microbiota could be different due to housing conditions of the respective vivarium. The authors may need to justify whether other groups' HFD protocol is the same and appropriate. Data quality of Fig.6F is not good (refer to related data in Fig.2b-c**

in Nat Comm 13:874, 2022). Because of these concerns and being not a major finding, I suggest moving Fig.6 to supplementary data section.

We appreciate the reviewer's comment and mention these limitations regarding dataset mining (diet duration/composition, housing conditions, microbiota composition) in our manuscript (lines 676-681). Due to these concerns, we also moved this figure (originally Fig. 6C) to the supplementary data section (Supplementary Fig. 8A). We also replaced our original figure examining Ki67 staining with PCNA staining (Supplementary Fig. 8E).

- 9. The authors claimed that intestinal IL-22Ra1 signaling is important in mediating Paneth cell function and numbers during HFD (Line 430-431, based on Fig.7A-C). However, the difference already exists between WT/KO mice at the steady state as shown by a previous paper from the authors (refer to Fig.2 in *Mucosal Imm* 14:389, 2020). Whether or not the difference is larger during HFD needs to be confirmed with new Exp using 4 groups of mice. To reveal a specific function of IL-22Ra1 signaling in Paneth cells during HFD, normal chow and HFD-fed *Il22ra1f/f* vs *Defa6-Cre+ Il22ra1f/f* mice should be used (Fig.7G-J), as there are already defects in Paneth cells in *Defa6-Cre+Il22ra1f/f* mice at the steady state (refer to Fig.4 in *Mucosal Imm* 14:389, 2020).**

We agree with the reviewer that our previous paper (*Mucosal Immunology*; 14:389, 2020) shows that Paneth cell numbers are reduced in the absence of IL-22RA1 signaling. To assess whether Paneth cell-specific IL-22RA1 signaling mediates systemic glucose metabolism in a diet-dependent manner, we placed *Il22ra1^{f/f};Defa6-cre-/+* mice on a long-term (16 weeks) control diet. We observed that control diet fed *Il22ra1^{f/f};Defa6-cre+* mice displayed similar clearance of glucose as their littermate controls (Supplementary Fig. 9E-F; lines 523-527). This contrasts our findings that high fat diet fed *Il22ra1^{f/f};Defa6-cre+* mice displayed impaired glucose metabolism post-fasting (Fig. 7K; Supplementary Fig. G; lines 527-529) and indicates that both IL-22RA1 signaling and diet composition regulate Paneth cell function. Furthermore, we assessed the importance of Paneth cells in mediating systemic glucose metabolism by generating Paneth cell-specific knockout and littermate control mice (*Defa6-cre-/+*, *ROSA26^{DTA}*; Supplementary Fig. 9C). We observed that chow diet fed *Defa6-cre+*, *ROSA26^{DTA}* mice displayed impaired glucose clearance (Fig. 7G), further supporting the Paneth cell-specific role in mediating systemic glucose metabolism (lines 514-518).

- 10. The main conclusion (as noted by the title: Intestinal IL-22Ra1 upregulates Paneth cell differentiation of Lgr5+ stem and progenitor cells to ameliorate high fat diet-induced metabolic disorders) appears entirely based on the result of Fig.7E (IL-22Ra1 signaling is not triggered, therefore may not be involved, in the datasets of Fig.6C and 7D). In my opinion, Fig.7E is simply not sufficient and convincing to support the major conclusion of paper. First, while Lgr5-eGFP reporter mice were on HFD, the authors did not show the treatment with IL22.Fc indeed ameliorate metabolic disorders (with QPCR, GTT, or image data shown in Fig.1~3) under this condition, with Lgr5+ cells being sorted from the same mice. Second, QPCR data alone is not convincing evidence. Approach such as confocal images or electron microscopy should be provided to strengthen the conclusion. Third, systemic i.p. injection of IL22.Fc is not specific to**

the gut. How do we know IL22.Fc is affecting the gut (and Lgr5+ cells) and then causing alterations in the liver/WAT? An approach to address this could be using HFD-fed Lgr5-eGFP;Vil-Cre+IL22ra1f/f mice for i.p. injection as controls.

We agree that the old title may not sufficiently represent the main findings of the paper. Our new title (Intestinal IL-22RA1 signaling regulates intrinsic and systemic lipid and glucose metabolism to alleviate obesity-associated disorders) provides a more accurate description of these findings.

To further assess whether IL-22RA1 signaling increases Lgr5⁺ ISC and progenitor cell differentiation into Paneth cells, we assessed whether mice injected with 80 ug IL22.Fc display higher Paneth cell numbers than control-treated mice. We provided quantification for the percentage of Paneth cells (CD24+, UEA+) isolated from Lgr5-GFP mice injected with 0 or 80 ug IL22.Fc (Supplementary Fig. 9A-B). Our data show a trend towards increase in Paneth cell number in IL22.Fc treated mice (lines 495-497).

The use of tamoxifen-inducible *Il22ra1^{fl/fl};Lgr5-EGFP-creERT2* mice may be problematic for determining the specific effects of IL-22RA1 signaling to Lgr5⁺ ISCs in mediating HFD-mediated disorders. The rate of epithelial cell turn-over in the intestines occurs after approximately 3-5 days, and published Lgr5 lineage tracing results indicate sufficient labeling of almost all descendent cells within 7-14 days. Thus, within 1-2 weeks of tamoxifen injection, a majority of intestinal epithelial cells in *Il22ra1^{fl/fl};Lgr5-creERT2* mice will lack IL-22RA1 and appear similar to *Il22ra1^{fl/fl};Villin-cre* mice. Furthermore, it is established that a longer period (12 weeks and longer) is necessary to see HFD-induced metabolic changes. Thus, it may be difficult to provide evidence that IL-22RA1 signaling in Lgr5⁺ ISCs promotes metabolic disorder.

- 11. Conceptually, I am confused about the title and final statement (IL-22Ra1 signaling to Paneth cells is important for maintaining their antimicrobial functions and mediating systemic glucose metabolism, Line 419-420). Do the authors want to conclude that a) IL-22Ra1 signaling targets Lgr5+ stem and progenitor cells to affect Paneth cell differentiation from these cells for ameliorating metabolic disorders (by title), or b) IL-22Ra1 signaling directly targets Paneth cells to regulate lipid/glucose metabolism (by the final statement)? Apparently, conclusion (a) and (b) are different things.**

We agree that the title and final statement focus on distinct findings and have changed both to provide more accurate descriptions. In figure 7, we show that Paneth cell number/function is reduced in the absence of intestinal IL-22RA1 signaling (Fig. 7A-C). To address why this may be occurring, we examined whether IL-22 acts on Lgr5⁺ ISCs or progenitor cells to mediate their expression of Paneth cell differentiation programs in order to replenish Paneth cell numbers. We observed that Lgr5⁺ ISCs and progenitor cells express elevated transcripts of *Lyz1* and *Mmp7* after IL-22.Fc treatment (Fig. 7E). We also show a trend towards increased Paneth cell numbers in IL-22.Fc treated mice (Supplementary Fig. 9B). Altogether, these results indicate that IL-22 may act on Lgr5⁺ ISCs and progenitor cells to mediate their differentiation into Paneth cells. We then included new data showing that Paneth cell numbers directly upregulate systemic glucose metabolism by using diphtheria toxin-depleted Paneth cell knockout and littermate control mice

(Fig. 7G; lines 514-518). Finally, we show that Paneth cell-specific IL-22RA1 signaling is important for increasing Paneth cell number and potentially plays a direct role in mediating systemic glucose metabolism (Fig. 7H-I, K). We edited the title (lines 1-2) and final statement (lines 469-470) to better represent these findings.

Minor Comments

- 1. Last paragraph in the Introduction (Line 105-125) is more like a conclusion or summary of results. I suggest to simplify the statement here or move to the discussion section.**

We reduced the length of the last paragraph and followed the journal guidelines to provide a brief summary of the manuscript's major conclusions (lines 106-118).

- 2. Some result texts appear to be lengthy, containing too much background, rationale, methodology, or discussion. It is advised to keep concise and focused to the results.**

We removed aspects of our background, methodology, discussion, and references from the results section to reduce its length and direct greater focus towards our results.

- 3. To be consistent with symbols used in Fig.2B, filled and open triangle as symbols for HFD-fed WT vs KO mice should also be used in Fig.2C, 2E, 2F, 4C, and 4D.**

We agree that the symbols should be consistent among similar experiments. We changed all high fat diet fed cre- mice to filled circles and cre+ mice to open circles for figures displaying bodyweight and glucose tolerance tests. We changed all control diet fed cre- mice to filled triangles and cre+ mice to open triangles for figures displaying bodyweight.

- 4. Duplicate references (Line 1095-1097 and Line 1098-1100).**

We deleted the duplicate reference.

- 5. The authors did not explore the possibility that IL-22Ra1 signaling may target and regulate brown adipose tissues (BAT) which also govern energy or lipid metabolism like WAT. WAT is the main site for storing excess fuel while BAT is specific for energy dissipation (Sci Transl Med 12:eaaz8664, 2020). Do we know adiponectin also drive Cre expression in BAT? It is well-known that both BAT and WAT contribute to systemic energy homeostasis, both could be linked to weight gain in a macroscopic assessment.**

We acknowledge that the role of IL-22RA1 signaling to brown adipose tissue and additional metabolic organs was not examined in this paper as a limitation. We address this in our discussion (lines 701-705).

Reviewer 2:

Remarks to the Author: In this manuscript by Stephen J, et al, the authors explored IL-22 mediated signaling in different tissue including intestine, liver and adipose tissue for metabolic disorders induced by HFD using tissue specific IL-22 ko mice. Since IL-22 is an important cytokine for mediating gut homeostasis and could influence host metabolism, it will be important and interesting to define the detailed mechanisms by which IL-22 impact host metabolism, especially in HFD context. While interestingly, this manuscript was only described, it is not clear how intestinal IL-22, gut microbiota influence host metabolisms in HFD-fed mice. Major and minor comments are outlined below:

- 1. The authors observed that IL-22 signaling in intestine and liver are important to regulate systemic glucose metabolism in HFD, but not in control diet condition. Did HFD influence IL-22 expression in these tissues?**

No difference in ileal expression of *Il22* was detected in the ileum of high fat diet (16 weeks) fed *Il22ra1^{fl/fl};Villin-cre-/+* mice (Supplementary Fig. 3G). We also did not observe differences in IL-22 protein in the liver or serum of *Il22ra1^{fl/fl};Villin-cre-/+* mice placed on a long-term high fat diet (Supplementary Fig. 3H-I). These results are mentioned in lines 230-233.

- 2. In Figure 2, Although the author did not observe a difference in weight gain between control diet and HFD, it is still important to have the fat pad mass or percentage to show whether there is difference in obesity for these mice.**

We included our findings for epididymal WAT fat pad mass for high fat diet fed *Il22ra1^{fl/fl};Villin-cre-/+* (Supplementary Fig. 3C), *Il22ra1^{fl/fl};Albumin-cre-/+* (Fig. 3H), and *Il22ra1^{fl/fl};Adiponectin-cre-/+* mice (Fig. 5C). No differences were observed in WAT fat pad mass after high fat diet. These results are presented in lines 219-221, 257-258, and 337-339.

- 3. Also in Figure 2, the authors showed that Villin-cre+ mice displayed impaired glucose clearance when compared to their littermate controls by glucose tolerance test, did these mice have difference in insulin resistance?**

We examined high fat diet fed *Il22ra1^{fl/fl};Villin-cre-/+* mice for insulin resistance (Supplementary Fig. 2B) as well as serum c-peptide and insulin levels (Supplementary Fig. 2C). No differences were detected in either case (lines 197-199).

- 4. The authors mostly only compare the data in HFD condition, seems lack the control diet. For example, In Figure 3, the control diet data should be included to compare with HFD, which may be used to explain why this phenotype only be observed in HFD. Also, is Albumin-cre mice have different expression of lipid metabolism genes in Figure 3?**

We included additional data regarding control diet conditions for *Il22ra1^{fl/fl};Villin-cre* (Supplementary Fig. 2D-E; Supplementary Fig. 3A, D; lines 202-203, 206-209, 216-217, 221-224) and *Il22ra1^{fl/fl};Albumin-cre* mice (Fig. 3B; Supplementary Fig. 4A; lines 244-246). No differences in systemic glucose metabolism were observed during control diet in either case (Supplementary

Fig. 2A; Supplementary Fig. 4A; lines 186-188, 244-246). Furthermore, we extended our analysis of high fat diet fed *Il22ra1^{fl/fl};Albumin-cre* mice by performing H&E staining and RT-PCR analysis of the lipid metabolism genes in the liver and adipose tissue (Fig. 3D-H; Supplementary Fig. 4B; lines 248-253). We only observed that liver-specific IL-22RA1 signaling was important in mediating the gene expression of *G6pc* (lines 249-251).

- 5. In Figure 3F, the authors showed that Villin-cre+ mice displayed significantly greater accumulation of crown-like structures than their littermate controls by HE staining of fat, while accumulation of crown-like structures are often evaluated by immunostaining of macrophages, not just HE staining.**

To evaluate macrophage accumulation, we performed immunofluorescence staining of F4/80 on tissues from long-term high fat diet fed *Il22ra1^{fl/fl};Villin-cre* mice (Supplementary Fig. 3B). We observed greater staining in the *Il22ra1^{fl/fl};Villin-cre+* mice, correlating with their increased inflammation observed via H&E. These results are presented in lines 218-219. We also detected elevated expression of *Mmp12*, a marker for insulin-resistant macrophages, in epididymal WAT from high fat diet fed *Il22ra1^{fl/fl};Villin-cre+* mice, further supporting our observations that these mice possess elevated levels of inflammatory cells (Fig. 4C). These results are discussed in lines 281-289.

- 6. It was interestingly that IL-22 upregulates hepatic lipid metabolism while downregulating intestinal lipid metabolism, could the authors discuss it?**

IL-22 has been shown to induce the production of epithelial IL-18 to provide intestinal protection (Chiang et al, 2022; Muñoz et al, 2015). We similarly observed that IL-22 induces expression of *Il18* in the intestines. However, we also observed that treatment of wildtype small intestinal organoids with rIL-18 resulted in significantly decreased expression of *Lipe* and *Pnpla2* as well as a modest decrease in *Ppara*. This supports the possibility that IL-22 may mediate some of its metabolic effects in conjunction with or by inducing IL-18. When we assessed whether liver- or WAT-specific expression of *Il18* was influenced after systemic injection with/without IL22.Fc, we did not observe any significant differences. This potentially indicates that intestinal IL-18 responses may be more strongly regulated by IL-22. These results are discussed in lines 410-423 and 571-579.

References:

- Chiang, H.-Y. *et al.* IL-22 initiates an IL-18-dependent epithelial response circuit to enforce intestinal host defence. *Nat Commun* **13**, 874 (2022).
- Muñoz, M. *et al.* Interleukin-22 Induces Interleukin-18 Expression from Epithelial Cells during Intestinal Infection. *Immunity* **42**, 321–331 (2015).

- 7. According to composition of control diet, it seems the control diet is compositional defined diet rather than grain-based chow diet, but the authors mentioned several times of chow diet in the manuscript. Please clarify.**

We reviewed our use of these terms. We utilized control diet for all experiments serving as a control for our long-term high fat diet mice. We also placed all germ-free mice on a control diet when performing the microbiota transfer experiments (Supplementary Fig. 5E). Chow diet was used for all DSS-associated experiments, RT-PCR analysis in Supplementary Fig. 7A, and the publicly available sequencing data we examined (Supplementary Fig. 8A).

8. For microbiota analysis by 16Ssequencing, it will be helpful to include more data about alpha diversity and beta diversity.

To greater assess what microbial changes may be occurring in *Il22ra1^{fl/fl};Villin-cre-/+* mice before and after high fat diet, we included additional data examining changes in alpha and beta diversity (Supplementary Fig. 5A-B). Although no distinct differences were observed between cre- and cre+ at similar time points, we observed changes before and after exposure to high fat diet. These results are discussed in lines 274-277.

9. For antibiotic treatment, did the mice with and without antibiotics have difference about GTT at week 12 before antibiotic treatment? Fecal microbiota transplantation may be needed to confirm the conclusion that intestinal IL-22RA1 signaling mediates extra-intestinal metabolism in a microbiota Intestinal.

We did not perform GTTs on *Il22ra1^{fl/fl};Villin-cre-/+* mice before placement on antibiotics (week 12). However, we performed cecal microbiota transplantation to additionally assess whether differences in microbiota composition between high fat diet fed *Il22ra1^{fl/fl};Villin-cre-/+* mice mediate systemic glucose metabolism. To do so, cecal contents from 16 week high fat diet fed *Il22ra1^{fl/fl};Villin-cre-/+* mice were homogenized and gavaged into 6-week germ-free mice (3 gavages every other day). GTTs were performed 1 week after the first gavage. We observed that germ-free mice that received contents from *Il22ra1^{fl/fl};Villin-cre+* mice displayed impaired glucose clearance (Supplementary Fig. 5E). These results support the notion that IL-22-induced microbiota changes are important for mediating extra-intestinal metabolism. These findings are mentioned in lines 297-301.

10. In Figure 5, the data about *Il22ra1^{fl/fl};Adiponectin-cre+* for DSS was confused to include in the manuscript. Did DSS or HFD induce IL-22 expression in fat?

We did not observe differences in WAT *Il22* expression from *Il22ra1^{fl/fl};Adiponectin-cre-/+* mice fed a long-term high fat diet (Supplementary Fig. 6A; lines 340-341). We also performed new experiments to assess the effects of WAT-specific IL-22RA1 on intestinal function after DSS to better incorporate these results in our manuscript. *Il22ra1^{fl/fl};Adiponectin-cre+* mice displayed greater crypt/villus length (Fig. 5I; lines 354-356) and numbers of alcian blue+ cells (Fig. 5 J-K; lines 358-361). These mice also displayed a trend for greater PCNA staining, a marker for epithelial proliferation (Supplementary Fig. 6B-C; lines 356-358).

11. In Figure 1, is any rationale that the dose of IL-22-Fc injection is 80ug?

The dose used was based on data from our previous publication (*Mucosal Immunology*; PMID: 33060802). To do so, a dose response curve was generated to determine the optimal dose of IL-22 in inducing ileal *Reg3g*. Our rationale for using this dose is referenced in lines 144-145.

12. Did Albumin-cre+ mice have difference in body weight and glucose tolerance test when fed control diet?

No differences in body weight or glucose tolerance were observed between *Il22ra1^{fl/fl};Albumin-cre* mice fed a long-term control diet (Fig. 3B; Supplementary Fig. 4A). These results are mentioned in lines 241-246.

13. The methodology is mostly sound, yet some important information is missing, e.g., the mice age at which initial HFD.

We indicate the age at which mice were placed on high fat diet or control diet in the methods (lines 732-7734, 749).

14. Some statement miss references, such as, lines 63-72.

We removed these lines from our manuscript to make our introduction more focused.

15. In Figure 5G, the colonic pathological score needs to be calculated.

As discussed in comment 10, we expanded our analysis regarding differences between DSS-treated *Il22ra1^{fl/fl};Adiponectin-cre-/+* mice. To quantify differences in colonic structure between DSS-treated *Il22ra1^{fl/fl};Adiponectin-cre-/+* mice, we utilized ImageJ to measure the lengths of colonic crypt/villus structures. We observed that *Il22ra1^{fl/fl};Adiponectin-cre+* mice displayed significantly greater crypts/villi (Fig. 5I; lines 354-356). Also discussed in comment 10, we performed staining for epithelial proliferation (PCNA+ cells) and goblet cells, both of which have been shown to correlate with intestinal inflammation (Supplementary Fig. 6B-C; Fig. 5J-K; lines 358-361, 356-358).

16. In Figure 7J, the whole GTT data should presented rather glucose level at 0 min.

We included the whole GTT data for high fat diet fed *Il22ra1^{fl/fl};Defa6-cre-/+* mice in Supplementary Fig. 9G. This data is referred to in 527-529.

17. In line 510, the author think that weight gain may be more clearly observed under stronger inflammatory conditions (such as DSS-induced colitis), this statement may be not correct, since DSS induced colitis will cause loss of body weight.

We agree that the term “weight gain” was misused here and have edited the text (lines 595-597).

18. In line 536, is one “reduced” should be “increased”?

We edited the text (line 639)

Reviewer 3:

Remarks to the Author: The systematic effects of IL-22 on metabolism have been well documented. However, how IL-22/IL-22RA signaling in different organs contribute to glucose and lipid metabolism is not clear. In this study, the authors used several lines of IL-22RA conditional KO mice to characterize the role of IL-22/IL-22RA signaling on metabolism in the liver, intestine and adipose tissue. Here are some concerns:

- 1. In Fig.1E, the authors observed IL-22Fc increased the expression of lipolysis related genes *Lipe* and *Pnpla2*, which was contradict with the results in Fig.1F which indicated that IL-22Fc increased lipid accumulation in HepG2 cells? Also, in Fig.1H, IL-22Fc reduced the expression of *Lipe* and *Pnpla2* in ileum, but inhibits lipid accumulation in intestinal organoids? For the impact of IL-22 on WAT, the authors only checked several lipid metabolism related gene expression, but how IL-22 directly influences WAT lipid metabolism is not clear. The genes selected mainly focused on lipolysis and beta oxidation, genes related to lipid synthesis and transport should be included in this study.**

Elevated expression of hepatic lipolysis genes (*Pnpla2*, *Lipe*) may occur because IL-22 promotes greater accumulation of intracellular lipids which thereby induces elevated expression of lipid metabolism genes. Similarly, decreased expression of ileal lipid metabolism genes (*Pnpla2*, *Lipe*, *Ppara*, *Pparg*) may occur because IL-22 downregulates intestinal lipid metabolism via induction of IL-18.

We expanded our analysis of lipid transport/metabolism genes (*Abca1*, *Apoa1*, *Fabp1*, *Fasn*, *Npc1l1*, *Stard4*) in the liver of high fat diet fed *Il22ra1^{fl/fl};Villin-cre* (Supplementary Fig. 2F; lines 206-209) and *Il22ra1^{fl/fl};Albumin-cre* mice (Supplementary Fig. 4B; line 251). We also observed that epididymal WAT from high fat diet fed *Il22ra1^{fl/fl};Villin-cre+* (but not fed *Il22ra1^{fl/fl};Albumin-cre+* or *Il22ra1^{fl/fl};Defa6-cre+*) displayed decreased levels of *Fasn* (Fig. 2J; Fig. 3I; Supplementary Fig. 9I). These results are mentioned in lines 221-224, 257-259, and 529-530. We also included gas chromatography analysis of triglyceride levels from high fat diet fed *Il22ra1^{fl/fl};Villin-cre-/+* mice (Supplementary Fig. 3F; lines 228-229).

Upon analysis of the microbiota from long-term high fat diet fed *Il22ra1^{fl/fl};Villin-cre-/+* mice, we detected elevated levels of *Oscillibacter* in *Il22ra1^{fl/fl};Villin-cre+* mice. *Oscillibacter* has been associated with the activation of insulin-resistant WAT macrophages (Li et al, 2022). We also observed that elevated inflammation and presence of macrophages in WAT from *Il22ra1^{fl/fl};Villin-cre+* mice corresponded with increased expression of *Mmp12*, an indicator for WAT-infiltrating metabolically active macrophages. Therefore, we believe it is possible that elevated abundance of *Oscillibacter* and potentially a dysbiosis of other microbes is driving this altered WAT

inflammation and, in turn, lipid metabolism. These results are mentioned in lines 278-289 and 649-658.

References:

- Li, Z. *et al.* Microbiota and adipocyte mitochondrial damage in type 2 diabetes are linked by *Mmp12* + macrophages. *Journal of Experimental Medicine* **219**, e20220017 (2022).

2. **In Figure 2, the authors did not see the difference in bodyweight between control and two of the IL-22RA conditional KO mice, however, in Figure3, more lipid accumulation were observed in the liver in *Il22ra1^{fl/fl};Villin-cre* mice. More biochemical markers such as serum TG, CHOL levels need to be evaluated.**

We were unable to expand our analysis for additional biochemical serum markers. However, we performed additional RT-PCR analysis of lipid metabolic genes in the liver and WAT as described in comment 1. We also included gas chromatography analysis of triglyceride levels from high fat diet fed *Il22ra1^{fl/fl};Villin-cre-/+* mice (Supplementary Fig. 3F; lines 228-229).

3. **In Fig.3A, Cre- mice with HFD should have significant steatosis in H&E staining. Fig.3C, more genes related to lipid metabolism should be included to evaluated the impact of IL-22RA signaling on lipid metabolism.**

We replaced the prior liver H&E image for high fat diet fed *Il22ra1^{fl/fl};Villin-cre-* mice with a more accurate representative image (Fig. 2D). We also increased our RT-PCR analysis for metabolic genes. We performed RT-PCR analysis on liver and WAT from high fat diet fed *Il22ra1^{fl/fl};Albumin-cre* mice to match our data from high fat diet fed *Il22ra1^{fl/fl};Villin-cre* mice (Fig. 3E, I), but no differences in these genes were observed. As mentioned in comment 1, additional genes associated with lipid transport/metabolism (*Abca1*, *Apoa1*, *Fabp1*, *Fasn*, *Npc1l1*, *Stard4*) were assessed in the liver of high fat diet fed *Il22ra1^{fl/fl};Villin-cre* and *Il22ra1^{fl/fl};Albumin-cre* mice (Supplementary Fig. 2F; Supplementary Fig. 4B). We also expanded our analysis of liver metabolism genes for *Il22ra1^{fl/fl};Defa6-cre* mice (Supplementary Fig. 9H), but did not detect any differences in the expression of these genes. Also as mentioned in comment 1, we observed that epididymal WAT from high fat diet fed *Il22ra1^{fl/fl};Villin-cre+* (but not fed *Il22ra1^{fl/fl};Albumin-cre+* or *Il22ra1^{fl/fl};Defa6-cre+*) displayed decreased levels of *Fasn* (Fig. 2J; Fig. 3I; Supplementary Fig. 9I).

4. **Will lack of IL-22RA in WAT contribute to the size and weight change in WAT when fed on HFD?**

We did not observe a difference in epididymal WAT fat pad mass between long-term high fat diet fed *Il22ra1^{fl/fl};Adiponectin-cre-/+* mice (Fig. 5C; lines 337-339).

5. **Did the author observed any difference in lipid metabolism in the liver and WAT in *Il22ra1fl/fl;Defa6-cre+* mice with HFD?**

No differences were detected in the expression of specific glucose (*Foxo1*, *G6pc*) or lipid (*Acc*, *Ppara*) metabolism genes in liver tissues of high fat diet fed *Il22ra1^{fl/fl};Defa6-cre-/+* mice (Supplementary Fig. 9H). In addition, no differences in lipid metabolism genes (*Acox1*, *Lipe*, *Fasn*) were observed in epididymal WAT from high fat diet fed *Il22ra1^{fl/fl};Defa6-cre-/+* mice (Supplementary Fig. 9I). These results are mentioned in lines 529-530.

REVIEWER COMMENTS

Reviewer #1 (Remarks to the Author):

Comments for Author

In this revised version of manuscript, the authors provide some new data to address previous concerns. While not fully address all questions, I do appreciate the authors' efforts to make revised manuscript more impactful and complete. As the authors made substantial changes in data presentation, interpretation, and conclusion, as such, I have to raise some additional concerns accordingly, based on the new story presented. Overall, the condition of HFD is an important novelty of this study, which may help to distinguish this study from several prior publications (J Hepatol 53:339, 2010, Nature 514:237, 2014, Nature 531:53, 2016). Thus, a major issue, as also raised by other reviewer, with the revised manuscript is still the Exp setting if HFD condition is included in the conclusion statement. That being said, the authors need to include 4 groups of mice (control-diet vs HFD in WT vs KO), whenever applicable, to draw the effect of HFD on those biological difference, that is the reason why two-way ANOVA is used in such analysis. Furthermore, because of revision and more data included, now the manuscript becomes even lengthy in terms of data interpretation. As mentioned in my previous comments, it is advised to keep concise and focused to the main findings (>5000 words only in the result section).

(01) The abstract is not well-written. Basically it covers many individual and descriptive findings without an ending statement (or take-home message).

(02) Based on many comparable measurements among Fig.2,3,5 (after Vil-Cre data being moved to Fig.2, all Albumin-Cre data being moved to Fig.3, all Adiponectin-Cre data being moved to Fig.5), the contribution of liver- and WAT-IL-22ra1 signaling on lipid/glucose metabolism appears very minimal, likely due to very low IL-22ra1 expression in liver and WAT compared to the ileum (Fig.1C). This result, unfortunately, somewhat decreases the impact of study and instead distracts the readers to DSS or HFD-DSS model used in Fig.5E-5L (as was also confused by reviewer-2 in his comment #10), which I found are not well-integrated studies.

(03) Unfortunately in this revised manuscript, the underlying mechanism of how IL-22ra1 signaling contributes to weight gain during HFD (Fig.1A is the starting point) is still unclear. I agree that "Intestinal epithelium- and liver-specific IL-22ra1 signaling upregulated systemic glucose metabolism independent of weight gain" (Line 46-47). However, the conclusion "weight loss was increased by WAT-specific IL-22ra1 signaling during intestinal inflammation" (Line 47-48) is less convincing, due to those results based on an inappropriate model of HFD-DSS, as pointed out by concerns described below (point #06-08).

(04) Line 174, “the tissue-specific effects of IL-22ra1 signaling remain unclear”. Indeed, opposing results (Fig.1D vs 1G, Fig.2G vs 3F) and similar results (Fig.2B-C vs 3B-C, Fig.2F vs 3E), are not equivocally supporting the outcome of liver steatosis (Fig.2D vs 3D) and WAT inflammation (Fig.2H vs 3G). If IL-18 is a key, the authors should try, for example, some IL-18 blockade exp in HFD-fed KO exp. Actually, all IL-18 data presented is nothing to do with the HFD condition.

(05) While data in Fig.4E-H are clear and convincing (although control-diet groups were not included), data quality in Fig.4G-I was not good, as high variations were presented. One can notice that the Y-scale is very large in Fig.4G (vs Fig.2F) and in Fig 4I (vs Fig 2J), which makes them to be statistically insignificant, leading to the conclusion that microbiota did play a role in intestinal epithelium IL-22ra1-mediated lipid metabolism.

(06) Data in Fig.5, overall, is weak and not sufficient to support the main conclusion “WAT-specific IL-22RA1 signaling mediates weight gain after exposure to HFD and intestinal inflammation” (Line 324-325). Actually only Fig.5L is related to the above statement. Fig.5E-K (somewhat irrelevant) are to test DSS phenotypes in Adiponectin-Cre;Il22ra1 f/f mice but Fig.5L is the key data related to the conclusion, which the authors unfortunately did not perform more analyses to strengthen the statement.

(07) Are Fig.5E (weight change%) and 5L (percent difference) the same in definition? How do we know the phenotype in Fig.5L was due to HFD-DSS but not DSS only? Is it possible to perform Exp with 4 groups of mice (WT with control diet+DSS, WT with HFD+DSS, KO with control diet+DSS, KO with HFD+DSS)? This is crucial as DSS treatment also causes weight loss (as was also confused by reviewer-2 in his comment #17) and that Adiponectin-Cre;Il22ra1 f/f mice are indeed more resistant to DSS already.

(08) Line 341-343, “Although we did not observe changes in weight gain after HFD, it may be possible that WAT-specific IL-22RA1 signaling is important for mediating weight gain during intestinal inflammation”. This is a weak rationale, but if it were the case, the same hypothesis should be valid (or tested) in intestinal epithelium-specific and liver-specific IL-22ra1 signaling in mediating weight gain during intestinal inflammation.

(09) If inflammation is the key factor for WAT to control weight gain, then the authors should see body weight difference in HFD-fed Villin-Cre;Il22ra1 f/f mice in Fig.2B, since WAT inflammation was detected in these mice (Fig.2H-I) without DSS. Or maybe body weight difference could be detected if HFD-DSS is applied to Villin-Cre;Il22ra1 f/f mice. It is not clear whether WAT inflammation or intestinal inflammation triggers weight gain mediated by WAT-specific IL-22ra1 signaling? I am confused about this point.

(10) The authors should provide those similar data (Fig.2D-E HFD/liver, Fig.2H-I HFD/WAT) in Adiponectin-Cre;Il22ra1 f/f mice in Fig.5.

(11) In Fig.5E and in Supplementary Fig.2A, filled and open triangle should be used (control diet fed mice).

(12) Data quality needs improvement. Fig.2D (has difference) vs Fig.3D (has no difference) is not convincing. Oil red staining of liver in HFD-fed Albumin-Cre;Il22ra1 f/f mice should be provided in Fig.3. WAT inflammation in Fig.3G should be quantified as in Fig.2H-I.

(13) While Fig.6A (and some genes confirmed by QPCR in Fig.6B) revealed an overview of potential lipid metabolism genes in gut epithelium that could be affected by intestinal IL-22ra1 signaling (or potentially in conjunction with an IL-22 inducible effect of IL-18) during HFD, unfortunately this is a correlation study which cannot address those biological outcomes (increased body weight gain, aberrant glucose metabolism, steatosis, WAT inflammation) in HFD-fed Villin-Cre;Il22ra1 f/f mice, in a manner similar to “an association of Oscillibacter/Mmp12 to microbiota dysbiosis (Fig.4A-C and Line 320-321)”. To make it impactful, the authors may consider to perform some blockade Exp (such as to inhibit potential target lipid metabolism genes) in HFD-fed Villin-Cre;Il22ra1 f/f mice to rescue the aberrant lipid metabolism.

(14) Line 484-486 “Since increased Paneth cell gene expression and numbers were observed in Il22ra1 fl/fl;Villin-cre- mice after HFD (Fig. 7A-C)”. The statement is not completely accurate, depending on what mice you are comparing. a) after HFD, Paneth cell numbers are decreased (control diet-WT vs HFD-WT), b) after HFD, KO Paneth cell numbers, compared to WT, are decreased (HFD-fed WT vs HFD-fed KO). In either case, I would not conclude “an increase in WT after HFD” because the dataset shown in Fig.7D. Instead, I would describe as Paneth cell program can not be maintained in KO after HFD, compared to that in WT.

(15) Line 491-497 (linked to Fig.7E, Supplementary Fig.9A-B). UEA1 is a lectin for glycans that is known to stain Goblet cell strongly. While Goblet cells (but how about other epithelial cells?) could be CD24-, CD24 high, but not CD24+, is a better way to gate Paneth cells (indicated in Fig.1D in Nat Cell Biol 14:1099, 2012, Fig.2 in Nature 469:415, 2011). To justify the sorting protocol (Supplementary Fig.9A) to minimize the above concerns, it is crucial to run a Paneth cell purity check on sorted CD24+ UEA+ population, either by QPCR or by flow cytometry of lysozyme. As this could potentially affect the conclusion, upregulation of Paneth cells by IL-22 Fc injection should be demonstrated at protein or cellular level by Flow (Supplementary Fig.9B) or IF (such as Fig.7B), not only by QPCR (Fig.7E, not convincing).

(16) The data in Supplementary Fig.9E vs 9G (no difference) and in Supplementary Fig.9F vs Fig.7K (marginal difference) is not convincing to draw the conclusion “IL-22ra1 signaling acts on Paneth cells to maintain a baseline glucose metabolism (Line 530-531)”. I think the authors would like to add “during HFD” to this statement. As mentioned in my previous comments, a defect (reduced numbers) of Paneth

cells already exists between WT/KO mice at the steady state (revised comment-9). So the novelty here is the condition of HFD (vs control diet) which the authors should use the approach of two-way ANOVA to make comparisons (control diet vs HFD in WT vs KO) whenever applicable. In brief, 4 groups of mice need to be conducted together to justify the true difference between WT/KO in the absence (control diet) and presence of HFD (this was also pointed out by reviewer-2 in couple exp settings). Supporting this, this is why the dataset in Fig.7D (done by others) was conducted in control and HFD conditions. In a similar way, in Line 486-488, an effect of HFD to alter Paneth cell differentiation/transcriptional program of Lgr5+ ISCs and progenitor cells (linked to Fig.7E), needs to be conducted in control-diet and HFD-fed mice, without (PBS) and with IL-22 Fc injection.

Reviewer #2 (Remarks to the Author):

The authors have taken significant steps to improve this manuscript, minor remaining comments are outlined below:

1. The macrophages stained by F4/80 in supplementary Fig. 3B should be quantified.
2. The representative images for MMP12 staining should be included in figure 4.
3. In line 354. "Correlating with their data from figures E-G" is missing the number of figure.
4. In line 360, it may be more accurate to say *Il22ra1*^{fl/fl}; Adiponectin-cre⁺ mice also displayed an "higher" number of alcian rather than "increased"

Reviewer #3 (Remarks to the Author):

I am satisfied with the revised version

Reviewer 1:

Comments for Author:

In this revised version of manuscript, the authors provide some new data to address previous concerns. While not fully address all questions, I do appreciate the authors' efforts to make revised manuscript more impactful and complete. As the authors made substantial changes in data presentation, interpretation, and conclusion, as such, I have to raise some additional concerns accordingly, based on the new story presented. Overall, the condition of HFD is an important novelty of this study, which may help to distinguish this study from several prior publications (J Hepatol 53:339, 2010, Nature 514:237, 2014, Nature 531:53, 2016). Thus, a major issue, as also raised by other reviewer, with the revised manuscript is still the Exp setting if HFD condition is included in the conclusion statement. That being said, the authors need to include 4 groups of mice (control-diet vs HFD in WT vs KO), whenever applicable, to draw the effect of HFD on those biological difference, that is the reason why two-way ANOVA is used in such analysis. Furthermore, because of revision and more data included, now the manuscript becomes even lengthy in terms of data interpretation. As mentioned in my previous comments, it is advised to keep concise and focused to the main findings (>5000 words only in the result section).

- 1. The abstract is not well-written. Basically it covers many individual and descriptive findings without an ending statement (or take-home message).**

We revised our abstract to provide an overview of the manuscript and an ending statement.

- 2. Based on many comparable measurements among Fig.2,3,5 (after Vil-Cre data being moved to Fig.2, all Albumin-Cre data being moved to Fig.3, all Adiponectin-Cre data being moved to Fig.5), the contribution of liver- and WAT-IL-22ra1 signaling on lipid/glucose metabolism appears very minimal, likely due to very low IL-22ra1 expression in liver and WAT compared to the ileum (Fig.1C). This result, unfortunately, somewhat decreases the impact of study and instead distracts the readers to DSS or HFD-DSS model used in Fig.5E-5L (as was also confused by reviewer-2 in his comment #10), which I found are not well-integrated studies.**

We removed all *Il22ra1^{fl/fl};Adiponectin-cre* DSS data due to concerns that Reviewer 1 currently has regarding its integration in the manuscript as well as former concerns held by Reviewer 2 during our first round of revisions. Our manuscript now expands on our findings from *Il22ra1^{fl/fl};Albumin-cre* and *Il22ra1^{fl/fl};Adiponectin-cre* after HFD. These new findings are included in Figure 5 and lines 299 – 313.

- 3. Unfortunately in this revised manuscript, the underlying mechanism of how IL-22ra1 signaling contributes to weight gain during HFD (Fig.1A is the starting point) is still unclear. I agree that “Intestinal epithelium- and liver-specific IL-22ra1 signaling upregulated systemic glucose metabolism independent of weight gain” (Line 46-47). However, the conclusion “weight loss was increased by WAT-specific IL-22ra1 signaling during intestinal inflammation” (Line 47-48) is less convincing, due to those results based on an inappropriate model of HFD-DSS, as pointed out by concerns described below (point #06-08).**

We removed all *Il22ra1^{fl/fl};Adiponectin-cre* DSS data due to concerns that Reviewer 1 currently has regarding its integration in the manuscript. While we do not see differences in weight gain under normal conditions, we observe that HFD fed *Il22ra1^{fl/fl};Villin-cre+* mice injected with IL-22.Fc do not lose as much weight as their littermate cre- controls. This suggests that intestinal IL-22RA1 signaling may mediate weight gain when elevated levels of IL-22 are present or after exogenous treatment with IL-22. These findings and discussion are presented in lines 342 – 348, 509 – 510. Furthermore, we do not solely rely on this phenotype to support our claim that tissue-specific IL-22RA1 signaling is important for regulating metabolism after HFD.

- 4. Line 174, “the tissue-specific effects of IL-22ra1 signaling remain unclear”. Indeed, opposing results (Fig.1D vs 1G, Fig.2G vs 3F) and similar results (Fig.2B-C vs 3B-C, Fig.2F vs 3E), are not equivocally supporting the outcome of liver steatosis (Fig.2D vs 3D) and WAT inflammation (Fig.2H vs 3G). If IL-18 is a key, the authors should try, for example, some IL-18 blockade exp in HFD-fed KO exp. Actually, all IL-18 data presented is nothing to do with the HFD condition.**

We would like to clarify the notion of similar and opposing results seen with IL-22.Fc injection versus intestine/liver-specific IL-22RA1 knockout mice. Supplementation of IL-22 and lack of IL-22RA1 signaling are two separate events. IL-22 supplementation will induce additional proteins/pathways such as IL-18, whereas IL-22RA1 deficiency solely reflects a lack of IL-20/IL-22/IL-24-mediated effects. Intestinal epithelial cells, but not hepatocytes or white adipose cells, respond to IL-22 to express IL-18. Thus, IL-22 supplementation mediates opposing events in the intestine and liver. Prior studies show that IL-22 mediates cytoprotective effects in the liver; likewise, we observe that intestinal IL-22RA1 deficiency results in impaired liver function. We discuss this in lines 361 - 378. We show that IL-18 may be responsible for additionally regulating the expression of metabolic genes in the intestine. However, determining the effects of IL-18 on metabolism is not the major focus of our manuscript. While our organoid experiments using rIL-18 were conducted under control conditions, similar trends in metabolic gene expression were observed when compared to our organoid experiments using rIL-22 under homeostatic conditions.

5. While data in Fig.4E-H are clear and convincing (although control-diet groups were not included), data quality in Fig.4G-I was not good, as high variations were presented. One can notice that the Y-scale is very large in Fig.4G (vs Fig.2F) and in Fig 4I (vs Fig 2J), which makes them to be statistically insignificant, leading to the conclusion that microbiota did play a role in intestinal epithelium IL-22ra1-mediated lipid metabolism.

We appreciate the reviewer for pointing this out. The variation observed may be due to a technical reason such as impaired RNA quality or sample preparation. We reisolated RNA from backup tissues that displayed higher variation in gene expression. Although some variation was still present, we observed that these newly isolated samples displayed an overall decrease in variation. In addition, the majority of data points for cre- and cre+ samples overlap, and no significant differences between these samples were observed. Shown below are the old (top) and new (bottom) RT-PCR analyses for these samples. We replaced the old images with the new ones.

Figure 1. Old and new RT-PCR analyses from liver and WAT of antibiotics-treated *Il22ra1^{fl/fl};Villin-cre* mice. (A – B) Old RT-PCR analyses, respectively, for liver and WAT. (C – D) New RT-PCR analyses, respectively, for liver and WAT.

6. Data in Fig.5, overall, is weak and not sufficient to support the main conclusion “WAT-specific IL-22RA1 signaling mediates weight gain after exposure to HFD and intestinal inflammation” (Line 324-325). Actually only Fig.5L is related to the above statement. Fig.5E-K (somewhat irrelevant) are to test DSS phenotypes in Adiponectin-Cre;Il22ra1 f/f mice but Fig.5L is the key data related to the conclusion, which the authors unfortunately did not perform more analyses to strengthen the statement.

We removed all *Il22ra1^{f/f};Adiponectin-cre* DSS data due to several concerns that Reviewer 1 raised regarding its integration in the manuscript.

- 7. Are Fig.5E (weight change%) and 5L (percent difference) the same in definition? How do we know the phenotype in Fig.5L was due to HFD-DSS but not DSS only? Is it possible to perform Exp with 4 groups of mice (WT with control diet+DSS, WT with HFD+DSS, KO with control diet+DSS, KO with HFD+DSS)? This is crucial as DSS treatment also causes weight loss (as was also confused by reviewer-2 in his comment #17) and that Adiponectin-Cre;Il22ra1 f/f mice are indeed more resistant to DSS already.**

We removed all *Il22ra1^{f/f};Adiponectin-cre* DSS data due to several concerns that Reviewer 1 raised regarding its integration in the manuscript.

- 8. Line 341-343, “Although we did not observe changes in weight gain after HFD, it may be possible that WAT-specific IL-22RA1 signaling is important for mediating weight gain during intestinal inflammation”. This is a weak rationale, but if it were the case, the same hypothesis should be valid (or tested) in intestinal epithelium-specific and liver-specific IL-22ra1 signaling in mediating weight gain during intestinal inflammation.**

We removed all *Il22ra1^{f/f};Adiponectin-cre* DSS data due to several concerns that Reviewer 1 raised regarding its integration in the manuscript.

- 9. If inflammation is the key factor for WAT to control weight gain, then the authors should see body weight difference in HFD-fed Villin-Cre;Il22ra1 f/f mice in Fig.2B, since WAT inflammation was detected in these mice (Fig.2H-I) without DSS. Or maybe body weight difference could be detected if HFD-DSS is applied to Villin-Cre;Il22ra1 f/f mice. It is not clear whether WAT inflammation or intestinal inflammation triggers weight gain mediated by WAT-specific IL-22ra1 signaling? I am confused about this point.**

We removed all *Il22ra1^{f/f};Adiponectin-cre* DSS data due to several concerns that Reviewer 1 raised regarding its integration in the manuscript.

- 10. The authors should provide those similar data (Fig.2D-E HFD/liver, Fig.2H-I HFD/WAT) in Adiponectin-Cre;Il22ra1 f/f mice in Fig.5.**

We included additional data examining the systemic metabolic effects of HFD on *Il22ra1^{f/f};Adiponectin-cre* mice. These include liver and WAT H&E staining (Fig. 5D and 5G), liver Oil Red O staining and quantification (Fig. 5E), and liver and WAT RT-PCR analysis (Fig. 5F and 5I). These results are presented in lines 307 – 313.

- 11. In Fig.5E and in Supplementary Fig.2A, filled and open triangle should be used (control diet fed mice).**

We thank the reviewer for bringing this to our attention. While Figure 5E was removed from our revised manuscript, we edited Supplementary Figure 2A accordingly.

- 12. Data quality needs improvement. Fig.2D (has difference) vs Fig.3D (has no difference) is not convincing. Oil red staining of liver in HFD-fed Albumin-Cre;Il22ra1 f/f mice should be provided in Fig.3. WAT inflammation in Fig.3G should be quantified as in Fig.2H-I.**

Oil red O staining and quantification are included for HFD fed *Il22ra1^{fl/fl};Albumin-cre* mice (Fig. 3E). Pathologist evaluation was performed to assess WAT inflammation in *Il22ra1^{fl/fl};Villin-cre* mice because a difference in inflammation was observed among cre- and cre+ slides. WAT from *Il22ra1^{fl/fl};Albumin-cre* mice were not scored by a pathologist because no notable differences were observed upon visual inspection of cre- and cre+ slides.

- 13. While Fig.6A (and some genes confirmed by QPCR in Fig.6B) revealed an overview of potential lipid metabolism genes in gut epithelium that could be affected by intestinal IL-22ra1 signaling (or potentially in conjunction with an IL-22 inducible effect of IL-18) during HFD, unfortunately this is a correlation study which cannot address those biological outcomes (increased body weight gain, aberrant glucose metabolism, steatosis, WAT inflammation) in HFD-fed Villin-Cre;Il22ra1 f/f mice, in a manner similar to “an association of Oscillibacter/Mmp12 to microbiota dysbiosis (Fig.4A-C and Line 320-321)”. To make it impactful, the authors may consider to perform some blockade Exp (such as to inhibit potential target lipid metabolism genes) in HFD-fed Villin-Cre;Il22ra1 f/f mice to rescue the aberrant lipid metabolism.**

We appreciate the reviewer’s comment. Indeed, we observed that IL-22 negatively regulates the expression of several lipid and peroxisome pathway related genes in the intestine. Liver and WAT also expresses these genes, and their associated pathways are well characterized in these tissues. It may be difficult to block lipid and peroxisome metabolism pathway only in the intestines. Any given inhibitor in *Il22ra1^{fl/fl};Villin-cre* mice will also impact liver/WAT functions. An ideal approach would be to use intestinal epithelium-specific Ppara knockout mice but this is beyond the scope of our present study. We discuss this limitation in lines 506 – 509.

- 14. Line 484-486 “Since increased Paneth cell gene expression and numbers were observed in Il22ra1 fl/fl;Villin-cre- mice after HFD (Fig. 7A-C)”. The statement is not completely accurate, depending on what mice you are comparing. a) after HFD, Paneth cell numbers are decreased (control diet-WT vs HFD-WT), b) after HFD, KO Paneth cell numbers, compared to WT, are decreased (HFD-fed WT vs HFD-fed KO). In either case, I would not conclude “an increase in WT after HFD” because the dataset shown in Fig.7D. Instead, I would describe as Paneth cell program can not be maintained in KO after HFD, compared to that in WT.**

We appreciate the comment and edited the text as suggested (Lines 433 – 434).

- 15. Line 491-497 (linked to Fig.7E, Supplementary Fig.9A-B). UEA1 is a lectin for glycans that is known to stain Goblet cell strongly. While Goblet cells (but how about other epithelial cells?) could be CD24-, CD24 high, but not CD24+, is a better way to gate Paneth cells (indicated in**

Fig.1D in *Nat Cell Biol* 14:1099, 2012, Fig.2 in *Nature* 469:415, 2011). To justify the sorting protocol (Supplementary Fig.9A) to minimize the above concerns, it is crucial to run a Paneth cell purity check on sorted CD24⁺ UEA⁺ population, either by QPCR or by flow cytometry of lysozyme. As this could potentially affect the conclusion, upregulation of Paneth cells by IL-22 Fc injection should be demonstrated at protein or cellular level by Flow (Supplementary Fig.9B) or IF (such as Fig.7B), not only by QPCR (Fig.7E, not convincing).

We thank the reviewer for bringing this to our attention. The previous gating strategy shown was missing one additional step. However, we modified Supplementary Figure 7A to include the new gating strategy used in our manuscript. This gating strategy was similar to that which we previously published in our paper (*Mucosal Immunology*, PMID: 33060802) and was based on work done by Sato et al (*Nature*, PMID: 21113151). To assess the purity of this sorted cell population, we performed RT-PCR analysis of different epithelial markers for cells that display CD24 after feeding mice a control diet and HFD (Supplementary Figure 9B). These include markers for Paneth cells (*Lyz1*), Goblet cells (*Muc2*, *Retnlb*), and enteroendocrine cells (*Chga*). We observed that transcript levels of *Lyz1* were greatly expressed when compared to the levels of *Muc2*, *Retnlb*, and *Chga*. Please note that Paneth cells stain positive for both LYZ1 and MUC2 (*Am J Physiol Gastrointest Liver Physiol*, PMID: 29698056); thus, *Retnlb* represent a true goblet cell marker. This supports our claim that this sorted population was highly enriched for Paneth cells. These results are discussed in lines 444 – 446.

16. The data in Supplementary Fig.9E vs 9G (no difference) and in Supplementary Fig.9F vs Fig.7K (marginal difference) is not convincing to draw the conclusion “IL-22ra1 signaling acts on Paneth cells to maintain a baseline glucose metabolism (Line 530-531)”. I think the authors would like to add “during HFD” to this statement. As mentioned in my previous comments, a defect (reduced numbers) of Paneth cells already exists between WT/KO mice at the steady state (revised comment-9). So the novelty here is the condition of HFD (vs control diet) which the authors should use the approach of two-way ANOVA to make comparisons (control diet vs HFD in WT vs KO) whenever applicable. In brief, 4 groups of mice need to be conducted together to justify the true difference between WT/KO in the absence (control diet) and presence of HFD (this was also pointed out by reviewer-2 in couple exp settings). Supporting this, this is why the dataset in Fig.7D (done by others) was conducted in control and HFD conditions. In a similar way, in Line 486-488, an effect of HFD to alter Paneth cell differentiation/transcriptional program of Lgr5⁺ ISCs and progenitor cells (linked to Fig.7E), needs to be conducted in control-diet and HFD-fed mice, without (PBS) and with IL-22 Fc injection.

We added “during HFD” to our conclusion as suggested (Line 481) as well as the section title (Line 425). We also included control diet data for our sorted Lgr5⁺ ISC and progenitor cell analysis (Lines 441 – 442; Supplementary Fig. 9B). After feeding control diet, we observed that levels of *Mmp7* but not *Lyz1* appeared to increase after IL-22.Fc injection. This data was plotted independently of our data obtained under HFD conditions due to a lower sample size. Since we performed our sorting experiments under control diet conditions (Supplementary Fig. 9B), we removed Figure 7D (heatmap of publicly available RNA-seq data in previous manuscript) from our current manuscript.

Reviewer 2:

Comments for Author:

The authors have taken significant steps to improve this manuscript, minor remaining comments are outlined below:

1. The macrophages stained by F4/80 in supplementary Fig. 3B should be quantified.

F4/80 staining was quantified. This data is referenced in line 194 and presented in Supplementary Fig. 3C.

2. The representative images for MMP12 staining should be included in figure 4.

WAT from HFD fed *Il22ra1^{fl/fl}; Villin-cre* mice were stained for MMP12. Representative images were included. These results are presented in line 263 and Fig. 4D.

3. In line 354. “Correlating with their data from figures E-G” is missing the number of figure.

This data was removed from the revised manuscript.

4. In line 360, it may be more accurate to say *Il22ra1fl/fl*; Adiponectin-cre+ mice also displayed an “higher” number of alcian rather than “increased”

This data was removed from the revised manuscript.

REVIEWER COMMENTS

Reviewer #1 (Remarks to the Author):

Comments for Author

(01) The authors did not fully address my previous concerns, regarding a role of IL-22RA signaling in Paneth cells during HFD. Fig.7E (organoid study) is a good example for in vivo experimental setup, where control (i.e. normal chow) and PA (i.e. HFD condition), with or without rIL-22 treatment (i.e. with/without IL-22 Fc injection), need to be included and assayed at the time, in order to fully justify the contribution of PA (i.e. HFD) to the regulatory function of IL-22. As only WT organoids were used in Fig.7E, one-way ANOVA was thus used for statistical analysis.

If based on this concept, when WT/KO were to be included, essentially at least 4 groups of mice (WT/KO in control diet vs HFD, or more with/without IL-22 Fc injection) need to be conducted when setting up such exp (Sup Fig.7E is a good example for this). I understand it could be challenging to have big groups of mice (i.e. to have gender and age-matched animals for all genotypes but metabolic disorder such as HFD is an age-dependent process), however; some assays (such as IF or QPCR, using saved tissue samples collected from different exp, but weight change or glucose test cannot be done this way) can be done at the same time, to a) minimize technical variations between batches and b) justify the true difference between groups and the effect of HFD. With this, one could plot 4 groups of data, with two-way ANOVA, on one scale to clearly show the HFD effect like Fig.7E and Sup Fig.7E. In my opinion, using independent control data (such as dataset below) on different scale to support the HFD effect is not a valid approach.

Fig.2F vs Sup Fig.2E

Fig.2J vs Sup Fig.3E

Fig.4C-D vs missing controls

Fig.6B vs Sup Fig.7A

Fig.7A vs missing controls (or Fig.2c in author's Mucoal Imm paper)

Fig.7B vs missing controls (or Fig.2d in author's Mucoal Imm paper)

Fig.7D vs Sup Fig.9B

Fig.7G vs missing controls (or Fig.4e in author's Mucoal Imm paper)

Sup Fig.8D-E vs missing controls

(02) All organoid QPCR data (Fig.1I, Fig.6D, Fig.7E, Sup Fig.7G, Sup Fig.8C) in the control groups is always set to 1, is this true? If the authors were using a house-keeping gene for normalization, one could still see a good distribution of control dataset, which is a correct way to analyze and compare the data between groups. Please explain this.

(03) Fig.6E-F appears on a different topic from Fig.6A-D. Move to supplementary or can be integrated into Paneth cell part in Fig.7.

(04) Line 407, 30 μ M, not 30 μ m.

Line 384, may be affected “by” IL-22RA1 signaling...

(05) Line 431-432 “This indicates that intestinal IL-22RA1 signaling mediates Paneth cell antimicrobial programs and number after HFD”, is misleading. Without 4 groups data on one scale, one cannot draw this conclusion as WT/KO at the steady state already have difference without HFD.

(06) It is uncommon to see the new “Discussion” section was greatly reduced from previous 6.5 pages to 0.5 page in the revision. It is suggested for authors to elaborate more biological impact, new findings, controversial points compared to previous publications, or molecular insights here to gain the visibility of this study.

Reviewer #2 (Remarks to the Author):

The authors have addressed all my concerns.

Reviewer 1:

Comments for Author:

1. The authors did not fully address my previous concerns, regarding a role of IL-22RA signaling in Paneth cells during HFD. Fig.7E (organoid study) is a good example for in vivo experimental setup, where control (i.e. normal chow) and PA (i.e. HFD condition), with or without rIL-22 treatment (i.e. with/without IL-22 Fc injection), need to be included and assayed at the time, in order to fully justify the contribution of PA (i.e. HFD) to the regulatory function of IL-22. As only WT organoids were used in Fig.7E, one-way ANOVA was thus used for statistical analysis.

If based on this concept, when WT/KO were to be included, essentially at least 4 groups of mice (WT/KO in control diet vs HFD, or more with/without IL-22 Fc injection) need to be conducted when setting up such exp (Sup Fig.7E is a good example for this). I understand it could be challenging to have big groups of mice (i.e. to have gender and age-matched animals for all genotypes but metabolic disorder such as HFD is an age-dependent process), however; some assays (such as IF or QPCR, using saved tissue samples collected from different exp, but weight change or glucose test cannot be done this way) can be done at the same time, to a) minimize technical variations between batches and b) justify the true difference between groups and the effect of HFD. With this, one could plot 4 groups of data, with two-way ANOVA, on one scale to clearly show the HFD effect like Fig.7E and Sup Fig.7E. In my opinion, using independent control data (such as dataset below) on different scale to support the HFD effect is not a valid approach.

Fig.2F vs Sup Fig.2E

Fig.2J vs Sup Fig.3E

Fig.4C-D vs missing controls

Fig.6B vs Sup Fig.7A

Fig.7A vs missing controls (or Fig.2c in author's Mucoal Imm paper)

Fig.7B vs missing controls (or Fig.2d in author's Mucoal Imm paper)

Fig.7D vs Sup Fig.9B

Fig.7G vs missing controls (or Fig.4e in author's Mucoal Imm paper)

Sup Fig.8D-E vs missing controls

We revised our paper to combine control and HFD data for the following figures mentioned above: Fig.2F vs Sup Fig.2E, Fig.2J vs Sup Fig.3E, Fig.6B vs Sup Fig.7A, and Fig.7D vs Sup Fig.9B. We also included control data for Fig.4C-D, Fig.7A, Fig.7B, Fig.7G, and Sup Fig. 8D-E.

2. All organoid QPCR data (Fig.1I, Fig.6D, Fig.7E, Sup Fig.7G, Sup Fig.8C) in the control groups is always set to 1, is this true? If the authors were using a house-keeping gene for normalization, one could still see a good distribution of control dataset, which is a correct way to analyze and compare the data between groups. Please explain this.

Yes, housekeeping genes were used to determine relative expression and a distribution of data points can be observed for our organoid data. However, the control data appears as 1.0 because we are displaying fold change values (derived from $\Delta\Delta C_T$ analysis) that were calculated separately for each biological sample. This method was performed to observe relative changes in gene expression among treatments more

precisely by setting the control group value as a baseline. We provide an explanation for this analysis in the methods (lines 592-594).

3. Fig.6E-F appears on a different topic from Fig.6A-D. Move to supplementary or can be integrated into Paneth cell part in Fig.7.

We included Fig. 6E-F with the supplementary data (Sup Fig.8 D-E).

4. Line 407, 30 μ M, not 30 μ m. Line 384, may be affected “by” IL-22RA1 signaling

This correction has been made (line 369).

5. Line 431-432 “This indicates that intestinal IL-22RA1 signaling mediates Paneth cell antimicrobial programs and number after HFD”, is misleading. Without 4 groups data on one scale, one cannot draw this conclusion as WT/KO at the steady state already have difference without HFD

We included control data for these groups as suggested. A decrease in LYZ1+ cells/crypt was observed after HFD but not chow diet (Figure 7B – C; lines 416 – 418).

6. It is uncommon to see the new “Discussion” section was greatly reduced from previous 6.5 pages to 0.5 page in the revision. It is suggested for authors to elaborate more biological impact, new findings, controversial points compared to previous publications, or molecular insights here to gain the visibility of this study.

We expanded our discussion to highlight new findings of our manuscript, place our work within the context of prior literature, and suggest future avenues of research.

REVIEWERS' COMMENTS

Reviewer #4 (Remarks to the Author):

The authors have addressed most of the questions raised by the reviewers. However, some issues still need to be checked and corrected, especially regarding the changes in the Figures and the corresponding explanations in the text. The discussion should be improved substantially and completed.

Specific comments are below.

Line 115. Indicate if mice were fed a HFD or a standard diet when analyzed the effects of intraperitoneally injected IL22.Fc. This is also the case in Materials and Methods section (line 554). If studies were performed under control diet feeding only, the reasons for doing so should be provided as IL22 signaling does not seem to be affected under controls diet feeding in the next section (starting at line 148).

Q1. The authors have compared results of genetic function loss under both control diet and HFD feeding but there are still inconsistencies between the text and the figures. It also seems that all parameters have not been analyzed in the four experimental groups and this can generate confusion (for example, it seems that triglycerides, ceramides and cholesterol have only been quantified in liver tissues under HFD feeding but not under control diet). It will be helpful if the authors include at the top of each graph the type of diet used to facilitate the interpretation of the results. This applies to all figures.

Line 175. The text is not aligned with the Figures. The Figure 2F shows effects of gene expression of Foxo1, G6PC and Acc, Ppara in both control diet and HDD fed mice while Supplementary Fig 3E shows absence of effect on other genes only under HFD feeding.

Line 177. When referring to Figure 2F and Supp 2F it should be stated that significant effects on individual lipids analysed were not detected (this is shown in Supplementary Figure 2F).

Q2. The transformations used for data normalization (set at 1 the value of the housekeeping gene) are not necessary and could change the final results. Fold-change analysis is usually based on the mean differences between the expression of the housekeeping gene and the tested gene in the different replicates.

Q3-4. Addressed.

Lines 462-464 and Figure 7. The presentation of results of the role of Paneth cells and IL22ra1 in Paneth cells in glucose metabolism is confusing. The evaluation of the role of Paneth cells knock out in glucose metabolism has been done only under chow diet (Fig 7F), but not under HFD feeding, this should be clarified in line 454. Then, the authors evaluated the specific role of IL22ra1 in Paneth cells in glucose metabolism showing no effect on GTT and fasting glucose under chow diet (Supp 9F and 9G) and not in GTT under HFD (Sup 9H). The authors indicate in the text that the IL22ra1 gene loss affects fasting glucose levels after HFD but refer also to Fig Sup 9H, while Fig Sup 9H does show no effects on GTT under HFD feeding. Better state that there were no effects on GTT under HFD when referring to Sup Fig 9H directly to avoid confusion.

Q6. The first statements are those included in the Abstract. It is recommended that the authors directly focus on their new findings, for example, stating that their study provides new insights into the tissue specific role of IL22 in metabolic disease, etc.

Line 489. Change remains to “remained” since authors provide new info in this regard.

Line 489. Change “implicate” to “show”.

Line 503. It seems that “also” is not necessary in the sentence.

Line 505. Add an explanation on the way enterocytes are involved in systemic glucose metabolism, whether authors refer to their role in glucose uptake or endocrine functions, etc.

Line 506. Provide specific examples of inhibitory cross talks; are the author referring to an opposite regulation of antimicrobial peptide expression and systemic glucose levels? This does not seem to be the case, they seem to be regulated in the same direction.

Line 515. It should be specified that you are referring to RNA-seq data of the small intestine.

REVIEWERS' COMMENTS

Reviewer #4 (Remarks to the Author):

The authors have addressed most of the questions raised by the reviewers. However, some issues still need to be checked and corrected, especially regarding the changes in the Figures and the corresponding explanations in the text. The discussion should be improved substantially and completed.

Specific comments are below.

- 1. Line 115. Indicate if mice were fed a HFD or a standard diet when analyzed the effects of intraperitoneally injected IL22.Fc. This is also the case in Materials and Methods section (line 554). If studies were performed under control diet feeding only, the reasons for doing so should be provided as IL22 signaling does not seem to be affected under controls diet feeding in the next section (starting at line 148).**

Mice were fed a chow diet. This has been noted in the main text (lines 116 – 118 and 578) and corresponding figure legend (lines 965 and 968). We were interested in determining the effects only of exogenous IL-22 in regulating metabolic gene expression and, therefore, performed experiments on chow diet fed mice. This is clarified in lines 115 – 118. While *Il22ra1^{fl/fl}*; *Villin-cre-* and *cre+* mice do not display differences in glucose clearance or metabolic gene expression after chow diet, this serves as a different model to evaluate the effects of endogenous IL-22RA1 signaling than experiments using intraperitoneal injection of IL-22.Fc. When administered at a higher dose, IL-22.Fc will act on all systemic tissues to more noticeably and broadly induce its effects. We have added this in revised manuscript (lines 119 – 121).

- 2. Q1. The authors have compared results of genetic function loss under both control diet and HFD feeding but there are still inconsistencies between the text and the figures. It also seems that all parameters have not been analyzed in the four experimental groups and this can generate confusion (for example, it seems that triglycerides, ceramides and cholesterol have only been quantified in liver tissues under HFD feeding but not under control diet). It will be helpful if the authors include at the top of each graph the type of diet used to facilitate the interpretation of the results. This applies to all figures.**

We included diet type at the top of all relevant figures and in the figure legends.

- 3. Line 175. The text is not aligned with the Figures. The Figure 2F shows effects of gene expression of Foxo1, G6PC and Acc, Ppara in both control diet and HDD fed**

mice while Supplementary Fig 3E shows absence of effect on other genes only under HFD feeding.

We appreciate the reviewer pointing this out and have corrected the text (lines 178-180).

- 4. Line 177. When referring to Figure 2F and Supp 2F it should be stated that significant effects on individual lipids analyzed were not detected (this is shown in Supplementary Figure 2F).**

We have modified the text for Supplementary Figures 2F and 3F where relevant (lines 181 – 183 and lines 196 – 198).

- 5. Q2. The transformations used for data normalization (set at 1 the value of the housekeeping gene) are not necessary and could change the final results. Fold-change analysis is usually based on the mean differences between the expression of the housekeeping gene and the tested gene in the different replicates.**

Fold change analysis was only performed this way for organoid experiments with paired biological samples. Organoids derived from a single mouse were exposed to different treatments (i.e., control media vs IL-22 supplementation). Control groups for individual biological samples are set to a baseline expression of 1.0 to clearly assess the change induced by each treatment on that biological sample. A brief description of why we conducted this analysis was included in our methods (lines 615 – 618).

- 6. Q3-4. Addressed.**

We thank the reviewer for their comment.

- 7. Lines 462-464 and Figure 7. The presentation of results of the role of Paneth cells and IL22ra1 in Paneth cells in glucose metabolism is confusing. The evaluation of the role of Paneth cells knock out in glucose metabolism has been done only under chow diet (Fig 7F), but not under HFD feeding, this should be clarified in line 454. Then, the authors evaluated the specific role of IL22ra1 in Paneth cells in glucose metabolism showing no effect on GTT and fasting glucose under chow diet (Supp 9F and 9G) and not in GTT under HFD (Sup 9H). The authors indicate in the text that the IL22ra1 gene loss affects fasting glucose levels after HFD but refer also to Fig Sup 9H, while Fig Sup 9H does show no effects on GTT under HFD feeding. Better state that there were no effects on GTT under HFD when referring to Sup Fig 9H directly to avoid confusion.**

We clarified the diet conditions in line 460. We also edited the text to reflect that no changes in glucose clearance were observed post-injection with glucose (lines 471 – 472).

- 8. Q6. The first statements are those included in the Abstract. It is recommended that the authors directly focus on their new findings, for example, stating that their study provides new insights into the tissue specific role of IL22 in metabolic disease, etc.**

To attract greater focus to our study's findings, we restructured the first paragraph of the Discussion to avoid restating background information that was previously presented in the Abstract and Introduction. We also focused on highlighting key findings from our manuscript that are further expanded on in the subsequent paragraphs. These changes are present in lines 478 – 482.

- 9. Line 489. Change remains to “remained” since authors provide new info in this regard.**

We changed this wording (line 494).

- 10. Line 489. Change “implicate” to “show”.**

We changed this wording (line 494).

- 11. Line 503. It seems that “also” is not necessary in the sentence.**

This word was deleted (line 509).

- 12. Line 505. Add an explanation on the way enterocytes are involved in systemic glucose metabolism, whether authors refer to their role in glucose uptake or endocrine functions, etc.**

We added a discussion regarding how IL-22 may act on other cells, such as enteroendocrine cells, to mediate systemic glucose metabolism (lines 510 – 516).

- 13. Line 506. Provide specific examples of inhibitory cross talks; are the author referring to an opposite regulation of antimicrobial peptide expression and systemic glucose levels? This does not seem to be the case, they seem to be regulated in the same direction.**

We agree with the reviewer that our original wording was confusing. We were referring to our observations that IL-22RA1 signaling results in decreased systemic glucose metabolism and increased expression of intestinal lipid metabolism genes. We modified the manuscript to remove this phrasing from our text (line 529). The examples we

previously included in lines 530 – 532 refer to studies where immune and metabolic functions are linked together and either regulated in similar or opposing manners.

14. Line 515. It should be specified that you are referring to RNA-seq data of the small intestine.

We changed this wording (lines 517 – 518).